# The induction of pyrenoid synthesis by hyperoxia and its implications for the natural diversity of photosynthetic responses in *Chlamydomonas*

Peter Neofotis[1], Joshua Temple[1,2], Oliver L Tessmer[1], Jacob Bibik[1], Nicole Norris[1], Eric Pollner[1], Ben Lucker[1], Sarathi M Weraduwage[1,3], Alecia Withrow[4], Barbara Sears[1], Greg Mogos[1], Melinda Frame[4], David Hall[1], Joseph Weissman[5], David M Kramer[1]*

[1]MSU-DOE Plant Research Laboratory, Michigan State University, East Lansing, United States; [2]Department of Plant Biology, Michigan State University, East Lansing, United States; [3]Great Lakes Bioenergy Research Center, Michigan State University, East Lansing, United States; [4]Center for Advanced Microscopy, Michigan State University, East Lansing, United States; [5]Corporate Strategic Research, ExxonMobil, Annandale, United States

**ABSTRACT** In algae, it is well established that the pyrenoid, a component of the carbon-concentrating mechanism (CCM), is essential for efficient photosynthesis at low $CO_2$. However, the signal that triggers the formation of the pyrenoid has remained elusive. Here, we show that, in *Chlamydomonas reinhardtii*, the pyrenoid is strongly induced by hyperoxia, even at high $CO_2$ or bicarbonate levels. These results suggest that the pyrenoid can be induced by a common product of photosynthesis specific to low $CO_2$ or hyperoxia. Consistent with this view, the photorespiratory by-product, $H_2O_2$, induced the pyrenoid, suggesting that it acts as a signal. Finally, we show evidence for linkages between genetic variations in hyperoxia tolerance, $H_2O_2$ signaling, and pyrenoid morphologies.

*For correspondence:
kramerd8@msu.edu

## Introduction

The maximal primary productivity of algae is often determined by the efficiency of photosynthesis, which is strongly impacted by environmental factors. In turn, the products of photosynthesis can also impact local environmental conditions, leading to feedback- (or self-) limitations (*Livansky, 1996*; *Pulz, 2001*; *Raso et al., 2012*; *Torzillo et al., 1998*; *Vonshak et al., 1996*; *Weissman et al., 1988*). One important, but relatively little studied feedback factor is hyperoxia, which results when $O_2$ is emitted as a by-product of photosynthesis more rapidly than it diffuses away or is consumed by respiration. Microalgae cultures are often observed with dissolved oxygen levels of up to 100–400% of air – or even higher (*Peng et al., 2013*), especially when the local supply of inorganic carbon ($C_i$) is high but consumption or diffusion of $O_2$ slow. In some species, hyperoxia constitutes a major hurdle in achieving low cost, highly productive micro-algae farms (*Peng et al., 2013*). Hyperoxia has been directly associated with loss of productivity in a wide range of algal and cyanobacterial species, including *Nannochropsis* (*Raso et al., 2012*), *Chlamydomonas reinhardtii* (*Kliphuis et al., 2011*), *Neochloris oleabundans* (*Peng et al., 2016a*; *Sousa et al., 2012*), *Chlorella sorokiniana* (*Ugwu et al., 2007*), and *Spirulina* (*Vonshak et al., 1996*).

Despite being recognized as a problem, how hyperoxia interferes with photosynthetic growth is not fully understood, and various mechanisms have been proposed, including reactive oxygen (ROS)-induced damage to the photosynthetic machinery, membrane structure, DNA, and other cellular components (*Marquez et al., 1995*; *Santabarbara et al., 2002*; *Ugwu et al., 2007*). Another mechanism by which high $O_2$ has been proposed to decrease productivity is photorespiration, a process initiated when the ribulose bisphosphate carboxylase/oxygenase (rubisco) enzyme fixes $O_2$ rather than $CO_2$, resulting in the production of the toxic side product phosphoglycolate, which is detoxified through the photorespiratory pathway, at the cost of lost energy and the release of fixed carbon (*Bauwe et al., 2010*). Oxygenation of rubisco can also result in the formation of rubisco inhibitors (*Kim and Portis, 2004*) that can further slow photosynthesis. If rubisco becomes inactivated, then ROS accumulation can lead to chlorosis and cell death, particularly in high light (*Spreitzer and Mets, 1981*).

Because photorespiration depends on competition between $CO_2$ and $O_2$ at rubisco, it can also contribute to loss of productivity under low inorganic carbon (*Wang et al., 2015*). The current atmospheric $CO_2$ concentration is well below the saturated concentration for rubisco's carboxylase activity (*Raven et al., 2008*) and $CO_2$ diffuses through aquatic environments 10,000 times slower than in air (*Moroney and Ynalvez, 2007*). Thus, many aquatic phototrophs, including the chlorophyte *Chlamydomonas reinhardtii*, possess carbon concentrating mechanisms (CCMs) that concentrate $CO_2$ above its $K_M$ at rubisco, which has been reported to be 29 μM (*Jordan and Ogren, 1981*) to 57 μM (*Berry et al., 1976*), in order to increase the relative rates of carboxylation relative to oxygenation (*Aizawa and Miyachi, 1986*; *Badger et al., 1980*).

The expression and function of green algal CCMs in eukaryotic algae is highly regulated; cells grown on or below air levels of $CO_2$ (0.04%) develop active CCMs (*Aizawa and Miyachi, 1986*; *Badger et al., 1980*), whereas those grown with high $CO_2$ levels lack them, and thus show low apparent affinities for $CO_2$. Cells grown at high $CO_2$ and rapidly transferred to low $CO_2$ show strong inhibition of photosynthesis (*Badger et al., 1980*; *Spalding et al., 1983*) until the CCM is induced and activated (*Aizawa and Miyachi, 1986*; *Badger et al., 1980*; *Manuel and Moroney, 1988*). Induction can then result in about 25 % of all genes being affected (*Fang et al., 2012*). It is thought that this acclimation is mediated by some mechanism in the cell to sense $CO_2$ availability, although CCM1 (also known as CIA5), the regulatory gene and protein thought to control the induction of the CCM in *Chlamydomonas*, is expressed at both high and low $CO_2$ conditions (*Fukuzawa et al., 2001*). Analysis of the dark-to-light transition in synchronized *Chlamydomonas* cells reveals that mechanisms, independent of gene transcription of known CCM components, are likely to play a role in the CCM's induction (*Mitchell et al., 2014*). The CCM in Chlorophytes involves a large number of components, including proteins that serve enzymatic and structural functions as well as a starch sheath that surrounds the pyrenoid, forming a subcellular compartment which acts as a trap to concentrate pumped inorganic carbon near localized rubisco (*Mackinder et al., 2017*; *Ramazanov et al., 1994*; *Wang et al., 2015*).

Pyrenoids are thought to have evolved multiple times (*Barrett et al., 2021*; *Mackinder et al., 2016*; *Meyer et al., 2020a*). The vast majority of data on pyrenoid formation is based on *Chlamydomonas*, where the pyrenoid forms by liquid-liquid phase separation (*Banani et al., 2016*; *Barrett et al., 2021*; *Wunder et al., 2018*), and several lines of evidence suggest that across the diverse lineages liquid-liquid phase separation is an integral part of pyrenoid formation (*Barrett et al., 2021*). Although across algal lineages it appears that rubisco catalytic properties are CCM dependent, it remains difficult to differentiate limitations in carbon uptake versus the leakiness of $CO_2$ as the selective pressure operating on rubisco; more detailed physiological experiments are needed to deduce these and other competing processes (*Goudet et al., 2020*). In *Chlamydomonas*, recent studies have indicated that a correctly formed starch sheath is required for normal pyrenoid operation of the CCM (*Itakura et al., 2019*; *Toyokawa et al., 2020*), although one study (*Villarejo et al., 1996*) had called that into question. Besides starch, the sheath contains several proteins which appear to be distributed over or in close proximity to the starch plates (*Mackinder et al., 2017*). The functional implication of these proteins, and their distribution patterns, remains unclear (*Toyokawa et al., 2020*). The starch plates are penetrated by tubule-like extensions of the thylakoid membranes, which are thought to supply $CO_2$ to the trapped rubisco by dehydration of luminal $HCO_3^-$ (*Engel et al., 2015*; *Mitra et al., 2004*; *Moroney and Ynalvez, 2007*). How the organelle's subcompartments of membrane tubules, surrounding phase separated rubisco matrix, and peripheral starch sheath are all held together is

unknown (*Meyer et al., 2020b*). Although, it has recently been found that some pyrenoid proteins share a sequence motif that is necessary to target proteins to the pyrenoid and bind to rubisco (*Meyer et al., 2020b*).

Extensive genetic and biochemical studies have identified a large number of components essential for CCM function (*Goodenough and Levine, 1970*; *Henk et al., 1995*; *Itakura et al., 2019*; *Spalding et al., 1983*; *Toyokawa et al., 2020*). Of particular interest to the current work are factors that contribute to the pyrenoid compartment itself, especially those that affect the localization of rubisco within its starch sheath or those that modify the structure of the starch sheath. A range of mutants in diverse genetic components fail to form pyrenoids (*Goodenough and Levine, 1970*; *Henk et al., 1995*; *Spreitzer et al., 1985*), or have altered pyrenoid ultrastructure with disorganized or missing starch sheaths (*Henk et al., 1995*; *Itakura et al., 2019*; *Toyokawa et al., 2020*). These mutants tend to require high $CO_2$ for growth, emphasizing the importance of the pyrenoid structure for the function of the CCM. However, the pyrenoid is not necessary in all cases for survival under low $CO_2$, as some species of *Chloromonas*, despite lacking pyrenoids, have functioning CCMs (*Morita et al., 1998*).

In this work, we explore the importance of an aspect of the CCM, in particular the pyrenoid, in responses to hyperoxia, rather than low $C_i$ availability. Since both low $CO_2$ and hyperoxia involve a lowering of the $CO_2$:$O_2$ ratio, we hypothesized that (1) the pyrenoid is induced by hyperoxia; (2) that differences in its induction and/or formation can be related to hyperoxia tolerance. Furthermore, since both low $CO_2$ and hyperoxia result in increased photorespiration, we hypothesized that (3) the signal for pyrenoid formation might be a by-product of photorespiration, $H_2O_2$. In order to address these hypotheses, we examined two natural isolates of *Chlamydomonas* with varying tolerances to hyperoxia, and their progeny, with the goal of better understanding the physiological mechanisms that underly responses to hyperoxia. Understanding such traits can give insights into the mechanisms and tradeoffs of adaptations for specific environmental niches. By extension, such traits and tradeoffs have strong relevance to applications ranging from algae cultivation to bioengineering crops for increase productivity (*Long et al., 2015*). Engineering the algal CCM into land plants is seen as a key route to improving crop photosynthesis (*Fei et al., 2021*; *Hennacy and Jonikas, 2020*; *Mackinder, 2018*; *Meyer et al., 2016*; *Rae et al., 2017*). If the algal pyrenoid $CO_2$ concentration system were engineered into crops such as rice, wheat, or soya yields could increase by up to 60 % (*Long et al., 2019*); yet these improvements will likely only occur if a complete algal-like CCM is assembled in angiosperms (*Atkinson et al., 2020*; *Barrett et al., 2021*). Such ambitions necessitate an understanding of the signals and trade-offs of pyrenoid formation, for which *Chlamydomonas* is an excellent model system.

## Materials and methods
### *Chlamydomonas* strains and mating

Strain CC-2343 (mt+), in a search for strains resistant to heavy metals $CdCl_2$ and $HgCl_2$, was isolated from soil in Melbourne, Florida in 1988 (*Spanier et al., 1992*). Strain CC-1009 (mt-) is a wild type strain tracing back to the 1945 collection of G.M. Smith, isolated in Amherst MA, but has been a separate line from the sequenced and widely regarded reference strain c137 (CC-124 and CC-125) and Sagar (CC-1690) since about 1950 (*Pröschold et al., 2005*). CC-5357 was generated by Luke MacKinder in the laboratory of Martin Jonikas (*Mackinder et al., 2016*). These (*Appendix 1—figure 1*) and other strains were obtained from the Chlamydomonas Resource Center (https://www.chlamycollection.org). CC-2343 and CC-1009 were mated using an established protocol (*Jiang and Stern, 2009*).

**Key resources table**

| Reagent type (species) or resource | Designation | Source or reference | Identifiers | Additional information |
|---|---|---|---|---|
| Strain, strain background (*Chlamydomonas reinhardtii*) | CC-2343 | https://www.chlamycollection.org | CC-2343 wild type mt+ [Jarvik #224, Melbourne, FL] | Wild type isolated from Florida in 1988 *Spanier et al., 1992* |
| Strain, strain background (*Chlamydomonas reinhardtii*) | CC-1009 | https://www.chlamycollection.org | CC-1009 wild type mt- [UTEX 89] | A descendant of a wild type collected in Amherst, MA in 1945 *Pröschold et al., 2005* |

*Continued on next page*

*Continued*

| Reagent type (species) or resource | Designation | Source or reference | Identifiers | Additional information |
| --- | --- | --- | --- | --- |
| Strain, strain background (*Chlamydomonas reinhardtii*) | CC-5357 | https://www.chlamycollection.org | CC-5357 RbcS1-Venus mt- | Contains a YFP tagged rubisco *Mackinder et al., 2016* |
| Strain, strain background (*Chlamydomonas reinhardtii*) | CC-2702 | https://www.chlamycollection.org | CC-2702 mt+ | Cia5 mutant, lacks carbon concentrating mechanism *Xiang et al., 2001* |
| Strain, strain background (*Chlamydomonas reinhardtii*) | c1_1 | This study | c1_1 mt+ | Tolerant to hyperoxia |
| Strain, strain background (*Chlamydomonas reinhardtii*) | c1_2 | This study | c1_2 mt+ | Tolerant to hyperoxia |
| Strain, strain background (*Chlamydomonas reinhardtii*) | c1_3 | This study | c1_3 mt- | Intolerant to hyperoxia |
| Strain, strain background (*Chlamydomonas reinhardtii*) | c1_4 | This study | c1_4 mt- | Intolerant to hyperoxia |

## Growth and biomass

Cultures (i.e. CC-2343, CC-1009, and progeny) were grown autotrophically in environmental photo-bioreactors (ePBRs) (*Lucker et al., 2014*), or in some cases in 125 mL Erlenmeyer flasks, in either a medium called 2NBH (*Davey et al., 2012*), which is a modified Bristol's medium (*Supplementary file 1A*), or (i.e. CC-5357 and other strains descendant from CC-4533) in Sueoka's high-salt medium (HS) (*Sueoka, 1960*) because of their requirement for ammonium rather than nitrate as a nitrogen source. When grown in ePBRs, culture density was maintained by turbidostat-controlled automatic dilution, adjusted to give chlorophyll concentrations of approximately 3 µg/mL. The media filled the columns to 15 cm in height, bringing the total volume to 330 mL. Following inoculation, all cultures were maintained for at least three days at constant chlorophyll prior the measuring of productivity. Standard illumination was provided on a 14:10 hr (light:dark) sinusoidal diurnal cycle, with the peak light intensity of 2000 µmol m$^{-2}$ s$^{-1}$ PAR. Gas was filtered with using a HEPA-Cap disposable air filtration capsule (Whatman, #67023600), and bubbled through a 5 mm gas dispersion stone with a porosity of 10–20 µm at a flow rate of 350 ml/min.

In our ePBRs, we used a series of sparging protocols to establish a range of $CO_2$ and $O_2$ levels as well as to simulate fluctuations in $CO_2$ that might occur during production culturing, including: (1) rapid sparges (one min on and one min off) during illumination; (2) 'raceway sparges', one min sparge each hour during illumination. For normoxic conditions, the sparge gas was 5 % $CO_2$, 21 % $O_2$, balance $N_2$. For 'hyperoxia' treatments, the sparge gas was 5 % $CO_2$ and 95 % $O_2$.

Biomass productivity in units of g · m$^{-2}$ · day$^{-1}$ was estimated by multiplying the volume of eluted culture as a result of turbidostatic dilutions by the measured Ash Free Dry Weight (AFDW) per unit volume, then normalizing to m$^2$ by dividing by the surface area of the ePBR water column. The column height was 15 cm and the surface area is 26.6 cm$^2$. Unless noted otherwise, all experiments were done in biological triplicate, each separate bioreactor or flask representing a different biological replicate.

When grown in the Erlenmeyer flasks, the cultures were grown in batch mode (*Anderson, 2005*) under ~80 µmol m$^{-2}$ s$^{-1}$ PAR of light and bubbled continuously via a glass Pasteur pipette with 5 % $CO_2$.

In our aerophilic, mixotrophic assays (i.e. *Appendix 1—figure 7*), cells were grown at steady state in 2NBH media in photobioreactors and then counted using a Beckman Coulter Z2 Coulter Counter at sizes between 3–10 microns. 50000, 5000, 500, 50 cells were then spotted onto Tris Acetate Phosphate (TAP) plates (*Gorman and Levine, 1965*) and grown under 80 µmol m$^{-2}$ s$^{-1}$ of PAR.

## Estimation of culture bicarbonate concentrations

Dissolved bicarbonate levels were estimated using an approach based on the release of $CO_2$ upon acidification of the media (*Hawkes et al., 1993*) using an in-house built instrument consisting of a 250 mL sealed glass reactor (a standard canning jar, Mason, USA) that houses a small but sensitive atmospheric $CO_2$ sensor (S8, https://www.senseair.com), a 3 cm long Teflon-coated magnetic stir bar and a small septum for introducing reagents. During experiments, the output of the $CO_2$ sensor was continuously collected at a rate of 1 Hz using a microcontroller (Teensy 3.2, PJCR, Sherwood, OR, USA)

and analyzed with a Python Jupyter (https://www.jupyter.org) notebook. The experiments started with the addition of 10 mL of sample and continuous stirring (approximately 30 Hz rotation frequency). Similar results were obtained when samples were drawn directly from the ePBR or passed through a 4 micron filter to remove cells. After introducing the sample, the system was allowed to equilibrate for 3 min, at which time the sample was acidified to pH <4 by addition of 200 µL of 1 N HCl. The acidification leads to hydrolysis of $HCO_3^-$ to $CO_2$ + $H_2O$, resulting in release (outgassing) of $CO_2$ into the chamber. To account for differential partitioning of $CO_2$ into the medium and atmosphere, responses were then calibrated by spiking the samples with a known concentrations of sodium bicarbonate.

## Estimation of $O_2$ and $C_i$ compensation points

The $C_i$ compensation point was estimated by measuring the extent to which steady-state photosynthesis in a suspension of cells could draw down $CO_2$ levels above the samples. Note that this approach will monitor the overall competition between $CO_2$ uptake by assimilation and $CO_2$ release by photorespiration and respiration. In the absence of the CCM, cells will directly fix $CO_2$ that diffuses into the chloroplast, but when the CCM is active, the cell will actively transport of $HCO_3^-$ to the chloroplast. In both cases, we expect equilibration between $HCO_3^-$ and $CO_2$ (in the medium and atmosphere), so that at a constant pH, the atmospheric $CO_2$ level should provide a measure of the ability of cells to draw down $C_i$. Freshly harvested cells were centrifuged (800 x $g$ for 5 min) and then placed into 2 mL of well buffered medium (HS +20 mM HEPES, pH 7.0) within a sealed, 25 mL plastic cuvette (Coulter, cat. no. A35471) and stirred with a 0.5 cm diameter magnetic stir bar rotating at approximately 3 Hz. Changes in $CO_2$ levels were monitored with a small $CO_2$ sensor (Senseair, cat. no. 004-0-0013), placed in the headspace above the sample. The suspension was sparged with nitrogen gas for 5 min to deplete the medium of $CO_2$ and $O_2$, then illuminated for 20 min to allow photorespiration, mitochondrial respiration and photosynthetic assimilation to achieve the steady state atmospheric $CO_2$ level, which was taken as the $C_i$ compensation point.

## Microscopy

At each time point of interest, 1 mL of culture (at ~3 µg/mL chlorophyll) were removed from our bioreactors, placed in Eppendorf tubes and mixed with 2 µl of Lugol's Solution (Sigma-Aldrich, cat. no. L6146) before being viewed in a Leica DMi90 inverted light microscope.

Transmission Electron Microscopy (TEM) was performed using a JEOL 1400 Flash instrument, and images were photographed with a Metattaki Flash cMOS camera. To prepare cells for microscopy, samples were resuspended in 2.5 % glyceraldehyde in cultures of 2NBH media, and then treated as previously described (*Du et al., 2018*). To quantify the relative size of the starch sheaths, using ImageJ (*Schindelin et al., 2012*) the area around the inner parameter of the pyrenoid was subtracted from the area around the outer parameter of the starch sheath. The remaining area was then divided by the total area of the cells to give the relative pyrenoid sheath size. To determine the percent exposure of the pyrenoid matrix, using ImageJ lines were drawn across the length of the starch plates or matrix holding the plates together, and total length of these lines was then assessed. Similarly, lines were drawn across gaps in the pyrenoid structures and the total length of the gap was also assessed. The total gap length was then divided by combined length of the gap and plates to give the 'percent exposure.'

Subcellular localization of Rubisco labeled with Venus fluorescence protein was imaged using an Olympus FluoView 1000 Confocal Laser Scanning Microscope, configured on an Olympus IX81 inverted microscope using either a 60 x PlanApo (NA 1.42) oil objective or a 100 x UplanApo (NA 1.40) oil objective. Venus fluorescence protein was excited using the 515 nm Argon laser emission line, and fluorescence emission was detected using a 530–620 nm band pass filter. We also repeated the analysis using a Nikon A1 Confocal Laser Scanning Microscope, configured on a Nikon Ti Eclipse inverted microscope using a 100 x Apo TIRF (NA 1.49) oil objective. Venus fluorescence protein was excited using the 515 nm Argon laser emission line, and fluorescence emission was detected using a 530–600 nm band pass filter. Transmitted laser light was simultaneously collected using brightfield optics. Confocal Z-series through the thickness of the algal cells were collected in 0.5 µm increments, typically through a 5 µm thickness, and the Z-stacked images were compressed into a 2D image, displayed as a Maximum Intensity Projection.

Confocal work to probe cellular reactive oxygen species (ROS) production was performed on the Olympus confocal microscope setup described above, using methods previously described (*Du et al., 2018*).

## H$_2$O$_2$ measurements

For H$_2$O$_2$ measurements, cells were treated with reagents of an Amplex Red Hydrogen Peroxide/Peroxidase Assay Kit (Molecular Probes/Invitrogen, Carlsbad, CA, USA), as has been used by previous researchers (*Lin et al., 2013*). In brief, 5 mL of the culture was collected by centrifugation, and the pellet was flash frozen in liquid nitrogen. The cells were then broken in 1 mL of 1 X reaction buffer from the assay kit, ground with glass beads, and briefly sonicated. The mixture was then centrifuged and the supernatant was then used to measure the cellular H$_2$O$_2$ concentrations after incubation with horseradish peroxidase at 25 °C for 30 min. The H$_2$O$_2$ concentrations were determined by a standard curve developed using 0.25–2.5 µM and normalized by calculating the amount of protein in the extract using a standard Bradford Assay (*Bradford, 1976*) with Bradford Reagent (Sigma-Aldrich, cat no. B6916).

## Rubisco activity assay

Rubisco enzymatic activity was assayed using an established protocol (*Li et al., 2019*; *Roeske and O'Leary, 1985*; *Sharkey et al., 1986*), with slight adjustments to make the protocol suitable specifically for *Chlamydomonas*. Briefly, cultures were harvested from the bioreactors, flash frozen in liquid nitrogen, and stored at –80 °C. Just prior to assay, samples were suspended in extraction buffer [50 mM 4-2(2-hydroxyethyl)–1-piperazine propane sulfonic acid (EPPS), pH 8, 30 mM NaCl, 10 mM mannitol, 5 mM MgCl$_2$, 2 mM EDTA, 5 mM DTT, 0.5 % (v/v) Triton X-100, 1 % polyvinylpolypyrrolidone (PVPP), 0.5 % casein, and 1 % protease inhibitor cocktail (P9599; Sigma-Aldrich)], sonicated, and vortexed to extract proteins. Aliquots of 20 µL of the extract was added to 80 µL of assay buffer [50 mM 4-2(2-hydroxyethyl)–1-piperazine propane sulfonic acid, pH 8, 5 mM MgCl$_2$, 0.2 mM EDTA, 0.5 mM Ribulose bisphosphate, and 15 mM NaH$^{12}$CO$_3$, and 0.3 mM H$^{14}$CO$_3^-$]. The suspensions were vortexed for three seconds, incubated for one minute, and then the reaction was halted by adding 100 µL of 1 M formic acid. The resulting acidification liberates unfixed inorganic C by converting HCO$_3^-$ to CO$_2$, which escapes from the buffer. The mixtures were vortexed again for 3 s and then dried on a hotplate at 75 °C. For measurements of total rubisco activity, the extracts were pre-incubated with activation solution (to give final concentrations of 20 mM MgCl$_2$, 15 mM H$^{12}$CO$_3$, and 61 mM 6-phosphogluconate) for ten minutes before being mixed with the assay buffer. The amount of fixed radioactivity was determined using a liquid scintillation counter (TriCarb 2800TR, Perkin Elmer). Each day, radioactivity in 10 µL of the assay buffer was counted to determine specific activity. Based on 1mCi = 2.22 x 10$^9$ disintegrations min$^{-1}$, initial and total rubisco activity was calculated as expressed as µmol m$^{-2}$ s$^{-1}$. The rates were divided by 0.943 to account for the discrimination against $^{14}$C (*Li et al., 2019*; *Roeske and O'Leary, 1985*). Three algal samples, each constituting a biological replicate, were run per treatment or condition, and three technical replicates were run for each biological replicate.

## Chlorophyll measurements

Chlorophyll content was measured using the method of Porra (*Balcerzak et al., 2021*; *Porra, 2002*), but with the modifications that the extraction solution contained 60 % acetone and 40 % DMSO, instead of 80 % acetone.

## Oxygen evolution and quantum yield of photosystem II ($\Phi_{II}$)

Cell suspensions growing at steady state were removed from the bioreactors and concentrated in fresh media to 50 µg/mL of chlorophyll in a cuvette. To drive out the oxygen, the cultures were then sparged with 1 % CO$_2$ and 99% N$_2$ gas. Subsequently also supplemented with 6.25 mM sodium bicarbonate, the cultures were illuminated with approximately 750 µmol photons m$^{-2}$ s$^{-1}$ of photosynthetically active radiation (PAR), measured using a submersible spherical micro quantum sensor (US-SQS/L, Walz) attached to a light meter (Li-250A, LiCor) from two red LEDs (emission at approximately 630 nm) aimed at opposing sides of the cuvette. Oxygen evolution was then measured using a Neofox oxygen sensor via a fiber optic fluorescent probe by Ocean Optics (Dunedin, Florida). The $\Phi_{II}$ measurements of the TAP plates were made in our dynamic environmental photosynthesis imager

(DEPI), using methods described previously in detail (*Cruz et al., 2016*) but, to avoid direct reflection of the measuring light into the camera, the plates were tilted by approximately 5° from the horizontal position.

## Results

### Hyperoxia differentially affects rubisco activity in the tolerant and sensitive lines

In an initial screen of sequenced *Chlamydomonas* isolates (*Jang and Ehrenreich, 2012*), we found two with contrasting tolerances to hyperoxia, with strain CC-1009 relatively tolerant to hyperoxia, continuing to grow, albeit at a suppressed rate, when exposed to 95 % oxygen and 5 % $CO_2$, while CC-2343 showed severely suppressed growth and eventual chlorosis or photobleaching in our ePBRs (*Hall, 2017*). Qualitatively, this varying tolerance was also observed when cultures were continuously sparged in batch culture (*Appendix 1—figure 2*), when the cultures were $CO_2$ saturated, indicating that the differential sensitivity was caused by hyperoxia rather than depletion of inorganic carbon sources (see also results on rapid sparging in the ePBR system, below).

*Spreitzer and Mets, 1981* found that rubisco activity-deficient mutants exhibited chlorotic phenotypes similar to those observed with CC-2343 under hyperoxia. We conjectured that rubisco inhibition may be playing a role in CC-2343's intolerance to hyperoxia. To be clear, while *Spreitzer and*

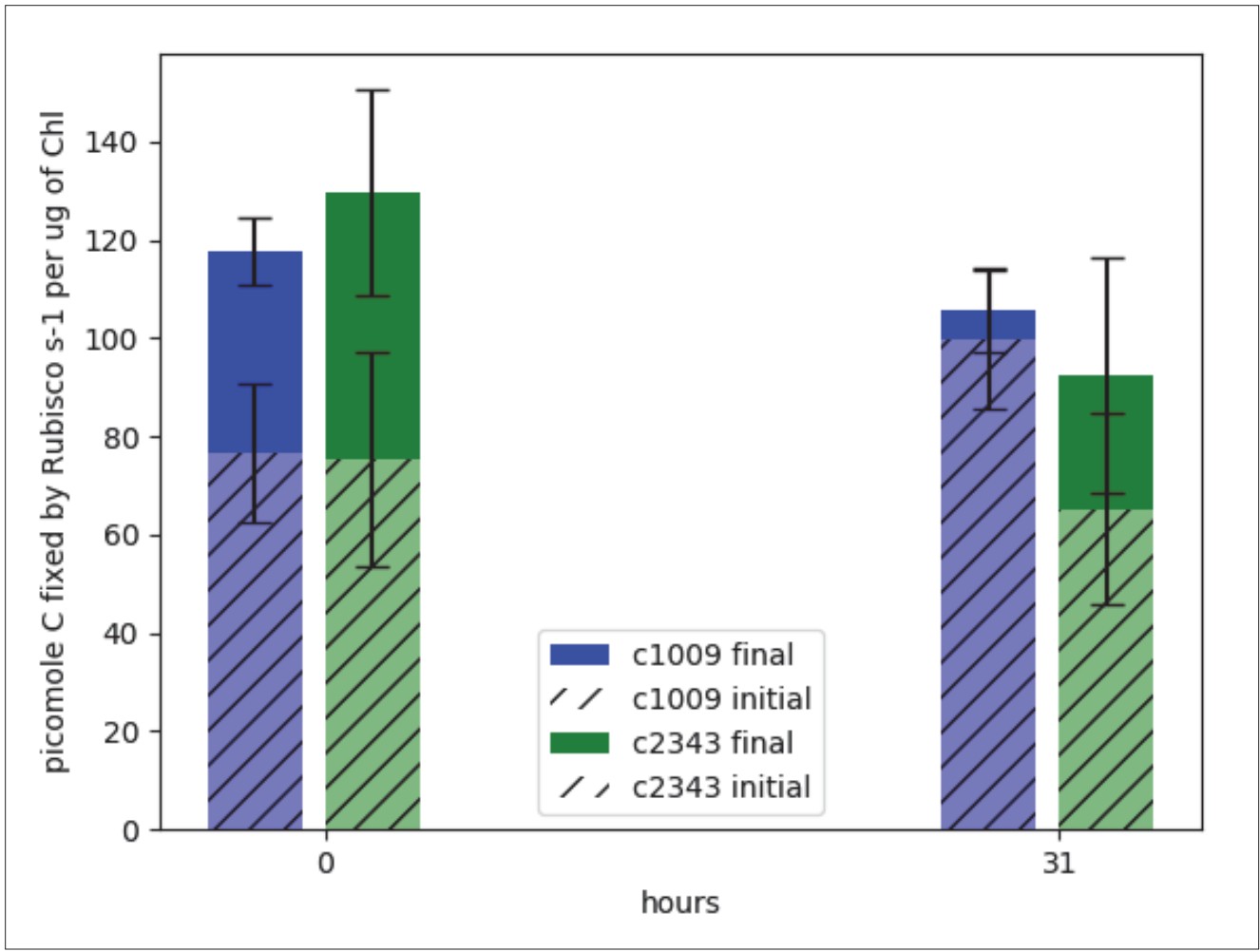

**Figure 1.** Graph of effects of hyperoxia on activity of rubisco in CC-1009 and CC-2343. Raw extracts of the cells prior to (zero hours) and after exposure to hyperoxia (31 hr, see Materials and methods) were assayed rapidly (hatched bars), reflecting the native activation state, or after pre-incubation for 10 min in the presence of $MgCl_2$, $H^{12}CO_3^-$, and 6-phosphogluconate, which promotes reactivation of inhibited enzyme (solid bars). For the table of values see *Supplementary file 1B*. Error bars represent the standard deviation of the three biological replicates, each with three technical replicates.

*Mets, 1981* were screening for mutants that were highly sensitive to even low light (~90 µmoles m$^{-2}$ s$^{-1}$ PAR), we grew our wild-type strains of *Chlamydomonas* under hyperoxia with diurnal sinusoidal light with peak light intensities of 2000 µmoles m$^{-2}$ s$^{-1}$ PAR. We found that, when sparged with 5 % $CO_2$, CC-1009 and CC-2343 grow very well at such light intensities. We measured rubisco activity of both strains prior to and after 31 hr exposure to hyperoxia (*Figure 1*). Rubisco activity was measured immediately after isolation to estimate steady-state activity at the time point of interest, which is controlled by both the total enzyme content and the fraction of the enzyme in the inactive state related to carbamylation state or the presence of inhibitors (*Li et al., 2019*; *Roeske and O'Leary, 1985*). Pre-incubating for ten minutes in the presence of $MgCl_2$, $HCO_3^-$, and 6-phosphogluconate (6 PG) promotes activation of the enzyme by stabilizing the Enzyme-$CO_2$-Mg-Complex of rubisco, allowing for the estimation of the maximal rubisco activity (*Badger and Lorimer, 1981*; *Chu and Bassham, 1973*; *Matsumura et al., 2012*). Using this method, we estimate that, under atmospheric levels of $O_2$, approximately 60 % of the enzyme was in its active form for both CC-1009 and CC-2343. After 31 hrs of exposure to hyperoxia, the total (maximal) activity of rubisco decreased in both lines, by about 10% and 28% in CC-1009 and CC-2343, respectively. However, in the case of CC-1009, the loss in total activity was compensated for by a large increase (to about 95%) in the fraction of active enzyme, leading to an overall increase of about 23 % in steady-state activity. By contrast, the fraction of activated rubisco was unchanged in CC-2343, leading to an overall decrease of about 13 % in steady-state activity.

Although we cannot ascribe the differences in photosynthetic phenotypes solely to rubisco deactivation, these results do suggest that the $CO_2/O_2$ concentrations or metabolic environments near rubisco are different under hyperoxia in the two lines. Apart from the metabolic environment, the activation state of rubisco can also be affected by the levels or activity of the rubisco activase (*Pollock et al., 2003*). But we found no consistent differences in the cellular contents of the rubisco activase protein between the cell lines. Another important factor that could affect the activity of rubisco and its metabolic environment in *Chlamydomonas* is its pyrenoid, a distinct, well-structured starch sheath surrounding localized rubisco that, under low $CO_2$, plays a key role in trapping $CO_2$ in the CCM (*del Campo et al., 1995*; *Harris, 1989*; *Harris, 2009*; *Ramazanov et al., 1994*). It has also been proposed to shield rubisco from high $O_2$ levels generated by PSII (*McKay and Gibbs, 1991*; *Toyokawa et al., 2020*). We thus initially hypothesized that: (1) the pyrenoid could be important for responses to hyperoxia and (2) differences in pyrenoid structure or regulation may then contribute to the distinct photosynthetic responses in the two lines. Consistent with these hypotheses, under saturating $CO_2$ conditions (*Appendix 1—figure 3*), the pyrenoid starch sheaths are not clearly discernable, in agreement with previous research showing that the pyrenoid starch sheath is not expressed under $C_i$ replete conditions (*Borkhsenious et al., 1998*; *Ramazanov et al., 1994*). Exposure to hyperoxia (95 % $O_2$ and 5 % $CO_2$), both when sparged rapidly (a square wave cycle one minute sparge and one minute rest, see Materials and methods) (*Appendix 1—figure 4 and 5*) or under high light with our raceway sparging regime (one minute sparge every hour) strongly induced starch sheath formation in our strains, but with genotype-dependent morphologies (*Figure 2*). The tolerant line, CC-1009, exhibited clearly defined, continuous starch sheath rings around its pyrenoid compartment, punctuated only in places where thylakoid tubules enter the pyrenoid matrix (*Figure 2A*). By contrast, CC-2343 showed more fragmented and porous structures, with gaps that were not clearly association with tubules (*Figure 2B*).

To test if decreased $CO_2$ or inorganic carbon levels could account for induction of pyrenoid synthesis, we directly assayed levels at various times during the sparge cycle, using the method described in Materials and methods. For the rapid sparging protocol, the estimated [$HCO_3^-$] under both normoxia and hyperoxia remained between 2 and 3 mM, and for the 'raceway' sparging protocol, between 1.3 and 1.7 mM a few minutes after sparging and 1.0–1.4 mM just prior to the following sparge. In all cases, the pH of the medium remained below 7.4. Thus, based on the known p$K_a$ values for the $CO_2$/ bicarbonate system, we estimate the lowest $CO_2$ levels experienced by the cultures, which occurred under raceway sparging conditions, remained above 100 µM, above the $K_m$ of rubisco (~29 µM – 57 µM) in *Chlamydomonas* (*Berry et al., 1976*; *Jordan and Ogren, 1981*). This is also in excess of the concentration found by *Toyokawa et al., 2020* to induce the formation of the pyrenoid starch sheath (2.1–3.1 µM). The CCM in *C. reinhardtii* is typically induced when the concentration of $CO_2$ in the air bubbled through the culture is decreased to around 0.5 % or lower (*Vance and Spalding, 2005*).

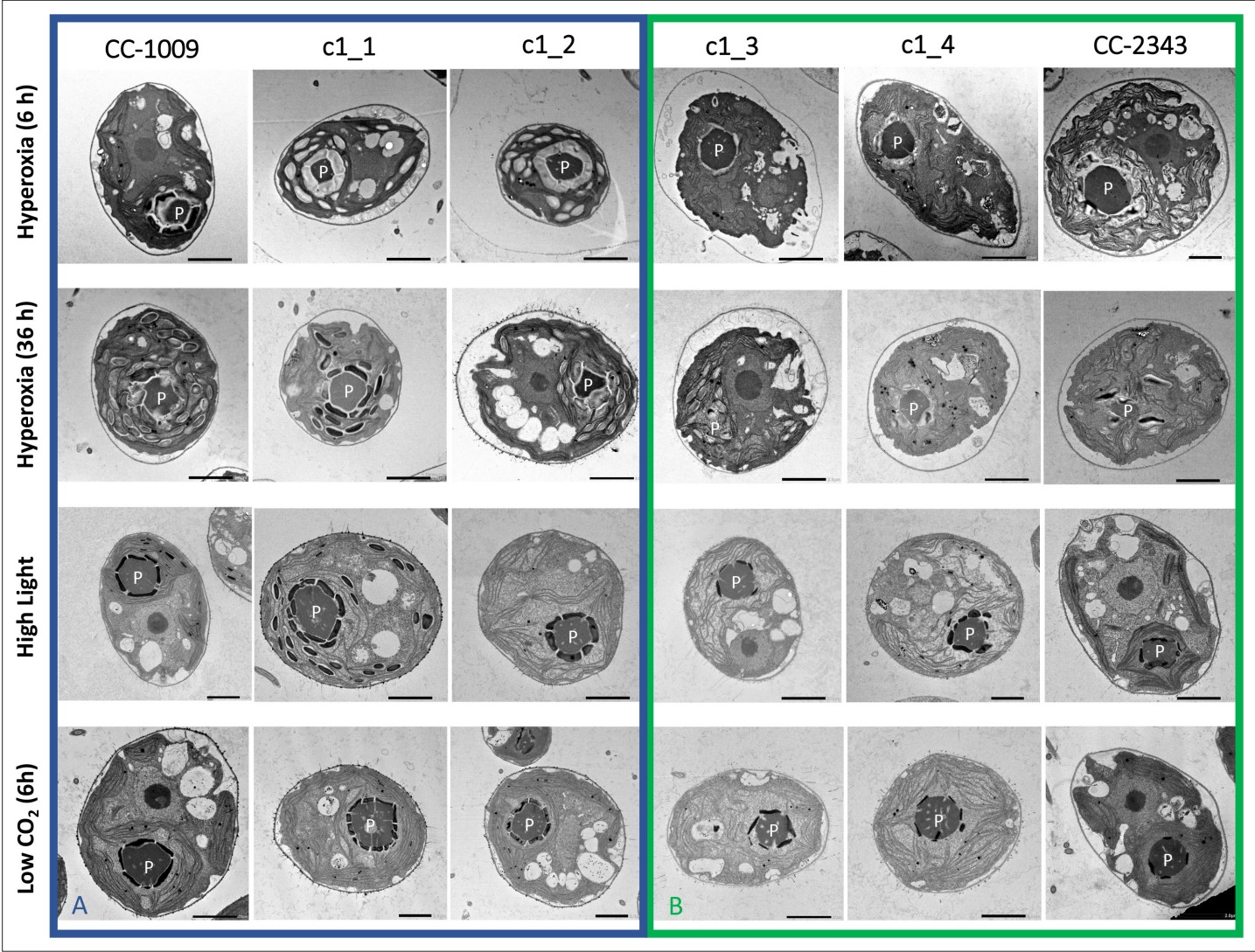

**Figure 2.** Representative TEM images of *Chlamydomonas* strains, the parents (CC-1009 & CC-2343), as well as their progeny c1_1, c1_2, c1_3, c1_4. Panel **A** shows strains with clearly defined, continuous starch sheath rings around the pyrenoid compartments, while the strains in Panel **B** have fragmented and porous pyrenoids, particularly under hyperoxia. Under steady state conditions, cells are grown with 5 % $CO_2$ with 14:10 hr (light:dark) sinusoidal illumination with peak light intensity of 2000 µmol m$^{-2}$ s$^{-1}$ PAR, in minimal 2NBH media. Here we show cells growing under hyperoxia (i.e. 95 % $O_2$ and 5 % $CO_2$) for 6 and 36 hr, near peak high light intensity under steady state, and low $CO_2$ (6 hr). Cells were fixed at 11:00 am, at 1945 µmoles m$^{-2}$ s$^{-1}$ PAR. Pyrenoids are labeled with 'P'. Scale bar = 2 µm.

Similar genotype-dependent pyrenoid morphologies were also observed, in a 2:2 segregation pattern, in four daughter cells dissected from a single tetrad (*Figure 2* & *Appendix 1—figure 6*). Two of the progenies, designated c1_1 and c1_2, when exposed to hyperoxia, developed completely sealed and robust rings, like CC-1009, while two others, designated c1_3 and c1_4, showed fragmented, porous structures, like CC-2343 (*Figure 2*). These differences were even more apparent after 31 hr of hyperoxia (*Figure 2*). Strains with fragmented pyrenoids (CC-2343, c1_3 ad c1_4) showed an abrupt inhibition of growth after one day of exposure to hyperoxia, whereas those with sealed pyrenoids (CC-1009, c1_1 and c1_2) continued to grow rapidly and produce biomass (*Figure 3*). Both progeny with fragmented pyrenoid sheaths grew even more slowly than the sensitive parent, CC-2343. On the other hand, those progeny with sealed pyrenoids (c1_1 and c1_2) initially grew more slowly than CC-1009, but maintained steady growth even on the fourth day of hyperoxia (*Figure 3*). These results suggest that the ultrastructural differences in the pyrenoid starch sheath (*Figure 2*) are related to the observed tolerances of growth to hyperoxia in both parent and progeny lines (*Figure 3*). However, the differences among the tolerant and sensitive lines, particularly the observation that the progeny

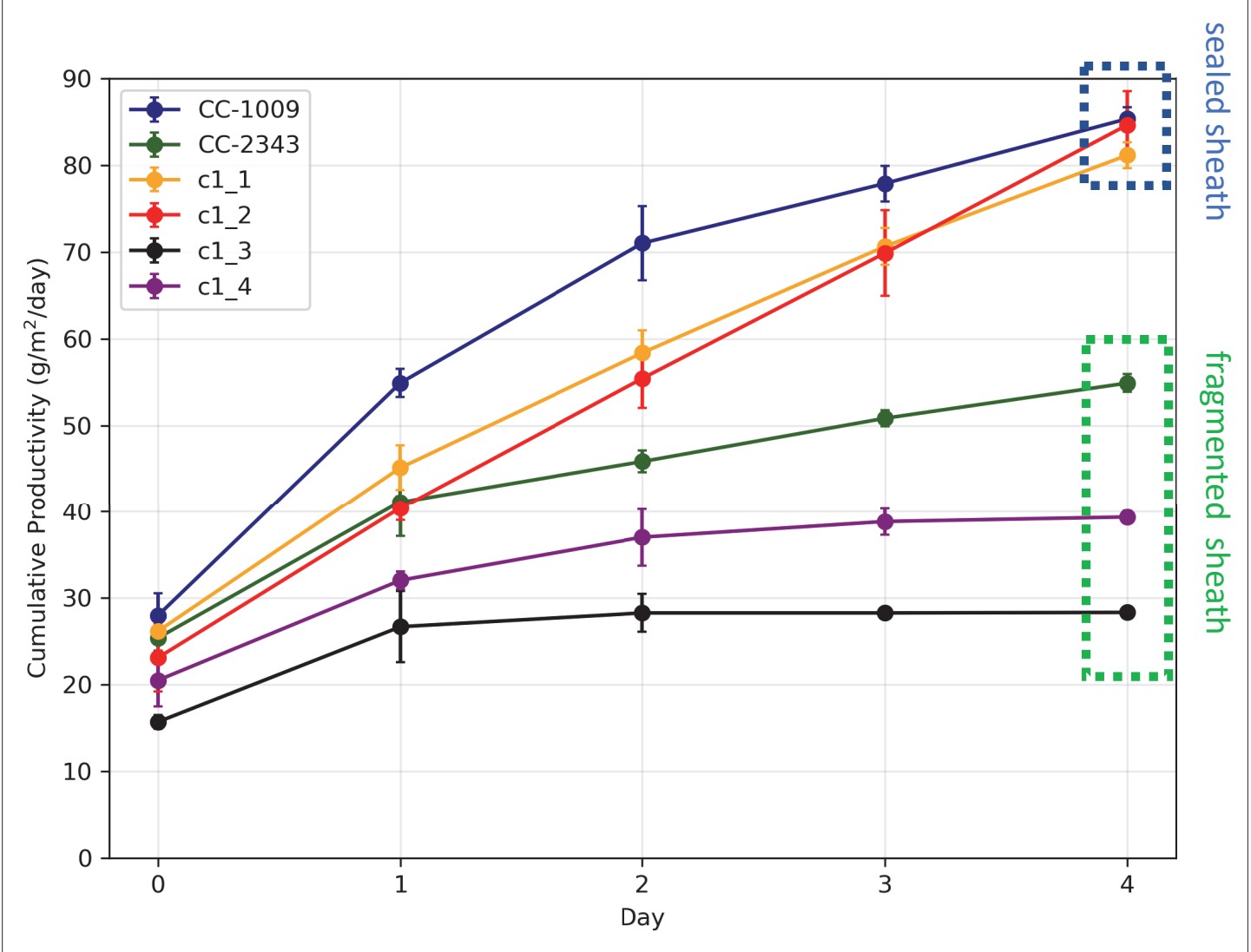

**Figure 3.** Cumulative biomass productivity following switching the bioreactors over to hyperoxia at dawn on day 0. Strains CC-1009, c1_1, and c1_2, which all showed continuous, sealed pyrenoids at 6 hours (see **Figure 2A**) all continued to accumulate biomass after three days of hyperoxia, while CC-2343, c1_3, and c1_4, which had fragmented, porous pyrenoid (**Figure 2B**) structures, did not, with daily productivities hovering at zero. Visual inspection (via light microscopy) also revealed that the cultures of the intolerant lines by day 3 consisted of severely stressed or dead cells, while the tolerant lines showed cells with continued viability. Prior to exposure to hyperoxia, cultures were grown at steady state (with 5 % $CO_2$ with 14:10 hr (light:dark) sinusoidal illumination with peak light intensity of 2000 µmoles $m^{-2}$ $s^{-1}$, in minimal 2NBH media) and at Day 0, the gas was switched to hyperoxia not at midnight but at dawn. Even at steady state, c1_3 had lower growth than the other strains, although this was not true when grown at other conditions (i.e. see **Appendix 1—figure 7**). Error bars represent standard deviation for three separate reactor experiments. By day 3, c1_3 always ceased growth. Even though productivities had just begun to decline at 6 hr, the pyrenoid structure (i.e. sealed vs. porous **Figure 2**) paralleled the eventual tolerances.

have phenotypes more extreme than those of the parent lines, imply that additional genetic factors (beyond those that control pyrenoid morphology) likely contribute to productivity under hyperoxia.

We also plated cultures of the CC-2343 and CC-1009 and the four progenies on TAP agar plates. Interestingly, growing the cells under aerophilic, mixotrophic conditions, we found that CC-2343 and the progeny that were intolerant to hyperoxia (c1_3 and c1_4) grew more rapidly than the hyperoxia tolerant lines which had exhibited the sealed, continuous pyrenoid starch sheaths (CC-1009, c1_1, c1_2) (**Appendix 1—figure 7**), despite exhibiting similar $\Phi_{II}$ values (**Appendix 1—figure 8**). In addition, when we grew the parent cells under steady state supplemented with 5 % $CO_2$, CC-2343 synthesized more starch (**Appendix 1—figure 9**).

Consistent with studies which have shown the pyrenoid is light dependent (*Kuchitsu et al., 1988*; *Lin and Carpenter, 1997*), CC-1009, CC-2343, and the F1 tetrad offspring also lost visible pyrenoid structures after dark exposure during the night (sparging once every hour with 5 % $CO_2$ in air), and the pyrenoid starch sheaths did not appear fully formed during the morning when PAR was low (*Appendix 1—figure 10*). As the light levels increased over 6 hr, though, the pyrenoid structures still formed under raceway sparging. Rather than a specific light level, this could be because it takes several hours to form pyrenoid structures and that photosynthesis is likely required. Under these conditions, CC-1009, c1_1, and c1_2 exhibited more tightly structured pyrenoids (*Figure 2*), although the differences were not as great as those exhibited under hyperoxia (*Figure 2*).

Consistent with previous work (*Borkhsenious et al., 1998*), pyrenoid formation was observed in all lines when cells were grown at high light and low $CO_2$, but with some differences in morphology among the lines. After exposure to low $CO_2$ (i.e. ambient air) for 6 hours, c1_1, c1_2 showed tightly closed sheath morphology similar to CC-1009 (*Figure 2A*). However, after 31 hr of exposure to low $CO_2$, the genotype differences in morphology became less apparent as all lines made starch sheaths of some integrity (*Appendix 1—figure 11*). All lines also grew similarly under ambient $CO_2$ in flasks under approximately 85 μmol photons m$^{-2}$ s$^{-1}$ (*Appendix 1—figure 12*). Taken together, these results suggest that low inorganic carbon, high $O_2$ and high light can all promote synthesis of the starch sheath, and that genetic variations modulate these responses.

## Pyrenoid formation is induced by exogenous and endogenously produced $H_2O_2$, and inhibited by the ROS scavenger, ascorbic acid

The above results suggest that a product of photosynthesis common to high light, low $CO_2$ and high $O_2$ may trigger pyrenoid formation. As discussed below, one possible signal is $H_2O_2$. *Figure 4* shows the effects of exogenous addition of $H_2O_2$ on the pyrenoid ultrastructure of *Chlamydomonas* parent lines. Cultures were harvested from photobioreactors in the morning (2 hr after the start of illumination) and diluted by half with fresh minimal 2NBH media with 5 mM bicarbonate – without (control) or with addition of 100 μM of $H_2O_2$. After 6 hr in low light (~85 μmol photons m$^{-2}$ s$^{-1}$), cells were fixed for EM as described in Materials and methods. Strikingly, treatment with $H_2O_2$ resulted in the appearance of thick, well-sealed starch sheaths, for both CC-1009 (*Figure 4A and B*) and CC-2343 (*Figure 4C and D*). Image J Analysis of the cells confirmed that there was a clear change in the size of the starch sheath (*Figure 5*). It is evident that hydrogen peroxide leads to significant increases in the prevalence of the starch sheath; which likely also coincides with a greater appearance of the pyrenoid periphery mesh – that is perhaps specifically related to LCI9 (*Mackinder et al., 2017*) - which appears to cement the starch plates together.

We also found that pyrenoids could also be induced in the presence of high bicarbonate via treatment with low concentrations of methyl viologen (*Appendix 1—figure 13*) or metronidazole (*Appendix 1—figure 14*), compounds known to induce internal hydrogen peroxide production by accepting electrons from PSI and passing them to $O_2$, forming superoxide, which is converted to $H_2O_2$ by superoxide dismutase (*Aksmann et al., 2016*; *Chang et al., 2013*; *Schmidt et al., 1977*). The concentrations of these compounds did not inhibit growth or motility over the time scale of the experiment (~6 hr) and thus their effects are likely to be caused by ROS production or altered metabolic status rather than severe cell damage. Complementing these findings, treatment with two known $H_2O_2$ scavengers, ascorbic acid (*Kuo et al., 2020*; *Nagy et al., 2015*; *Appendix 1—figure 15*) or dimethylthiourea (*Chang et al., 2013*; *Appendix 1—figure 16*) prevented the formation of the pyrenoid starch sheath under low $CO_2$ conditions. Overall, these results are consistent with the role of $H_2O_2$ in triggering the formation of the pyrenoid, though it remains to be determined whether such effects are direct or indirect, for example resulting of altered metabolic status.

Hydrogen peroxide treatment was found also to affect the localization of rubisco (*Figure 6*), which is sequestered in the pyrenoid at low $CO_2$ (*Borkhsenious et al., 1998*). We assessed changes in localization using a modified *Chlamydomonas reinhardtii* strain, CC-5357, expressing rubisco small subunit (RbcS1) tagged with the Venus fluorescent protein (*Mackinder et al., 2016*). Under control conditions (5 mM bicarbonate, no $H_2O_2$ treatment), labelled rubisco was present throughout the chloroplast, with some localization in a pyrenoid matrix-like structure (*Figure 6A*). However, approximately six hours after treatment with 100 μM $H_2O_2$, rubisco became strongly localized to the pyrenoid matrix (*Figure 6B*; *Appendix 1—figure 17*; Transparent Reporting Image 11), with very little fluorescent

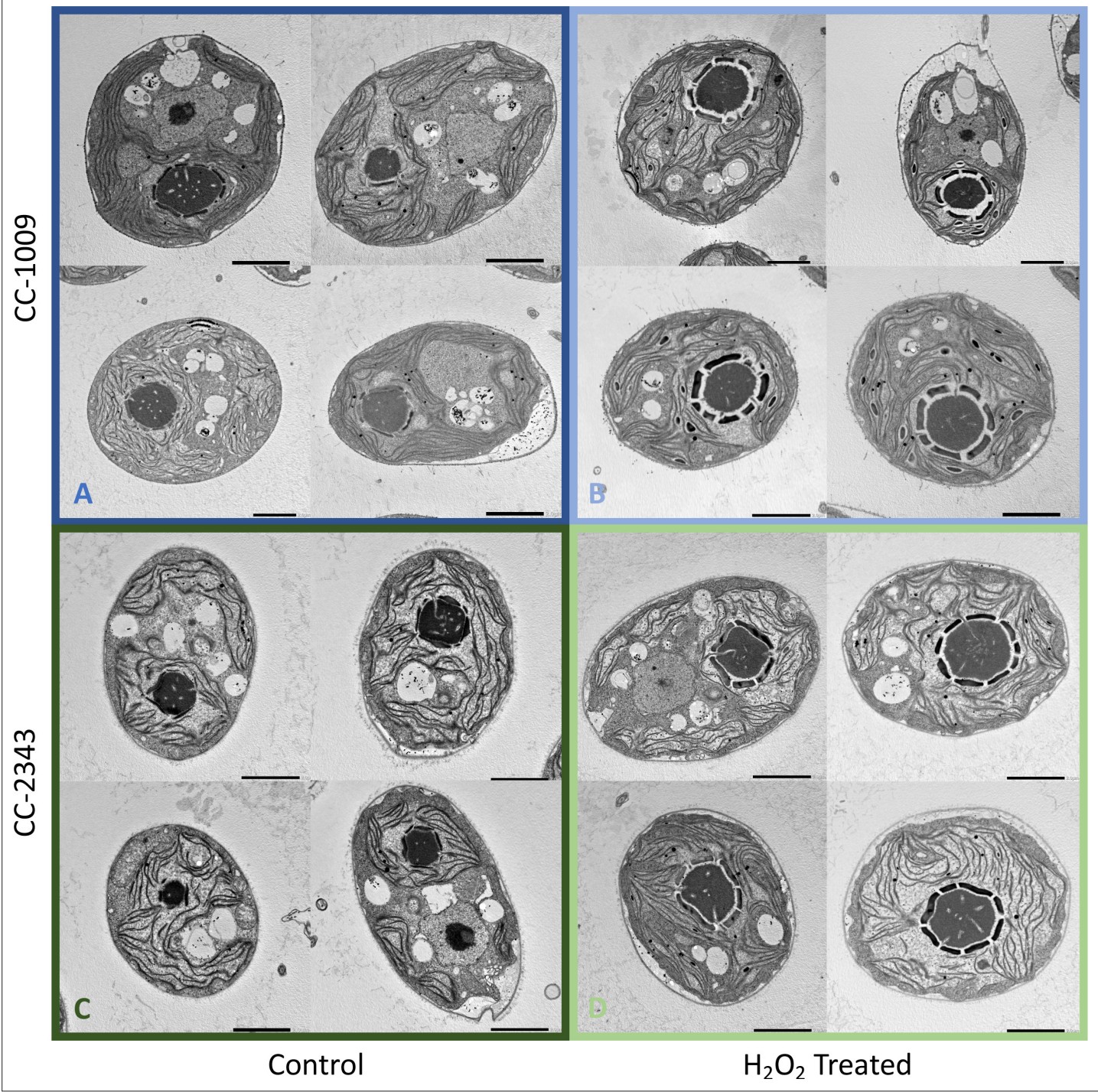

**Figure 4.** Representative TEM images of CC-1009 (Panels **A** and **B**) and CC-2343 (Panels **C** and **D**) control and cells treated, at 7:00 am in the morning, 2 hr after our sinusoidal light had turned on, with 100 µM of $H_2O_2$, and then exposed to 6 hr of low light (~85 µmol m$^{-2}$ s$^{-1}$ PAR) with saturating 5 mM bicarbonate in minimal 2NBH media. Scale bar = 2 µm.

signal outside this structure (see quantification of fluorescence signal, *Figure 6C*). Similar results were also found with the addition of methyl viologen and metronidazole (*Appendix 1—figure 18*), although the confocal laser in combination with the inhibitors appeared to make the samples unstable and not allow for multiple observations of the same slide. We also found evidence that even when sparged with saturating $CO_2$, hyperoxia may result in an apparent increase in the aggregation of rubisco in the pyrenoids (*Appendix 1—figure 19*), indicating that oxygen has some control of this aggregation.

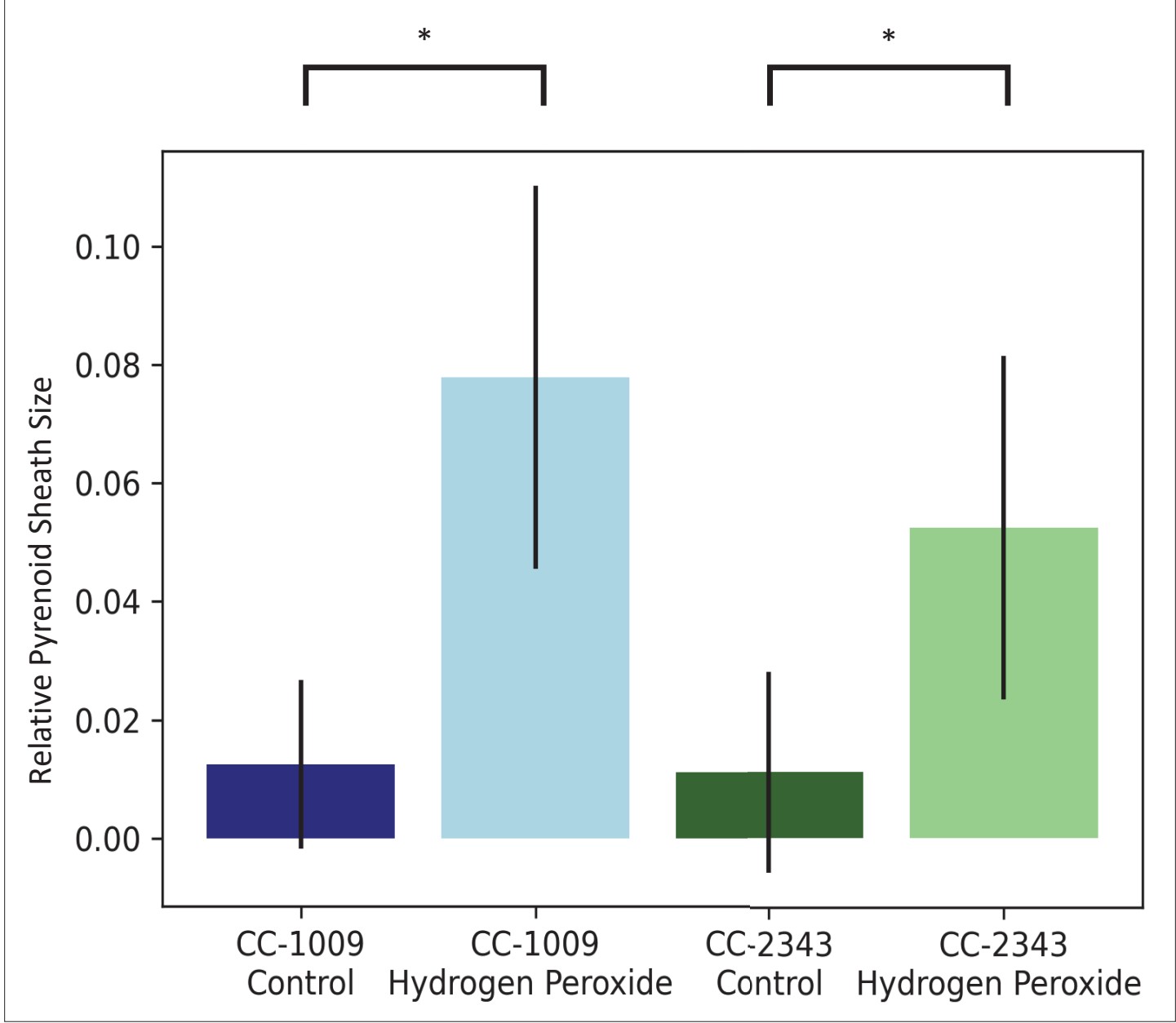

**Figure 5.** Image J Analysis of **CC-2343** and **CC-1009** cells, control (**green**, **blue**) and exposed to $H_2O_2$ (**light green**, **light blue**), normalized to cell size. In response to pre-treatment with $H_2O_2$, pyrenoid sheath size increased. Relative pyrenoid sizes were estimated using the ImageJ program by measuring the visible projected areas of starch sheath in TEM images and compared to that of the projected areas of the cells. Error bars represent the standard deviation between the approximately 30 cells analyzed (See *Figure 4* and Transparent Reporting Image 7-10). * p < 0.001.

Previously, the aggregation of rubisco into the pyrenoid has been associated with the CCM (*Freeman Rosenzweig et al., 2017*; *Mitchell et al., 2014*).

By contrast, no significant changes in rubisco localization were observed when upon addition of 100 µM $H_2O_2$ to TAP-grown cells (*Appendix 1—figure 20*), the media used in another study testing the effects of hydrogen peroxide on *Chlamydomonas* (*Blaby et al., 2015*), implying that the effect was dependent on the photosynthetic state of the cells and/or suppressed in the presence of this organic carbon source. Consistent with this interpretation, cells grown on TAP plates showed no observable pyrenoid starch sheath by light microscopy or starch staining (*Appendix 1—figure 21*) in contrast with what we observed with cells grown in liquid minimal media. Furthermore, when CC-5357 was grown

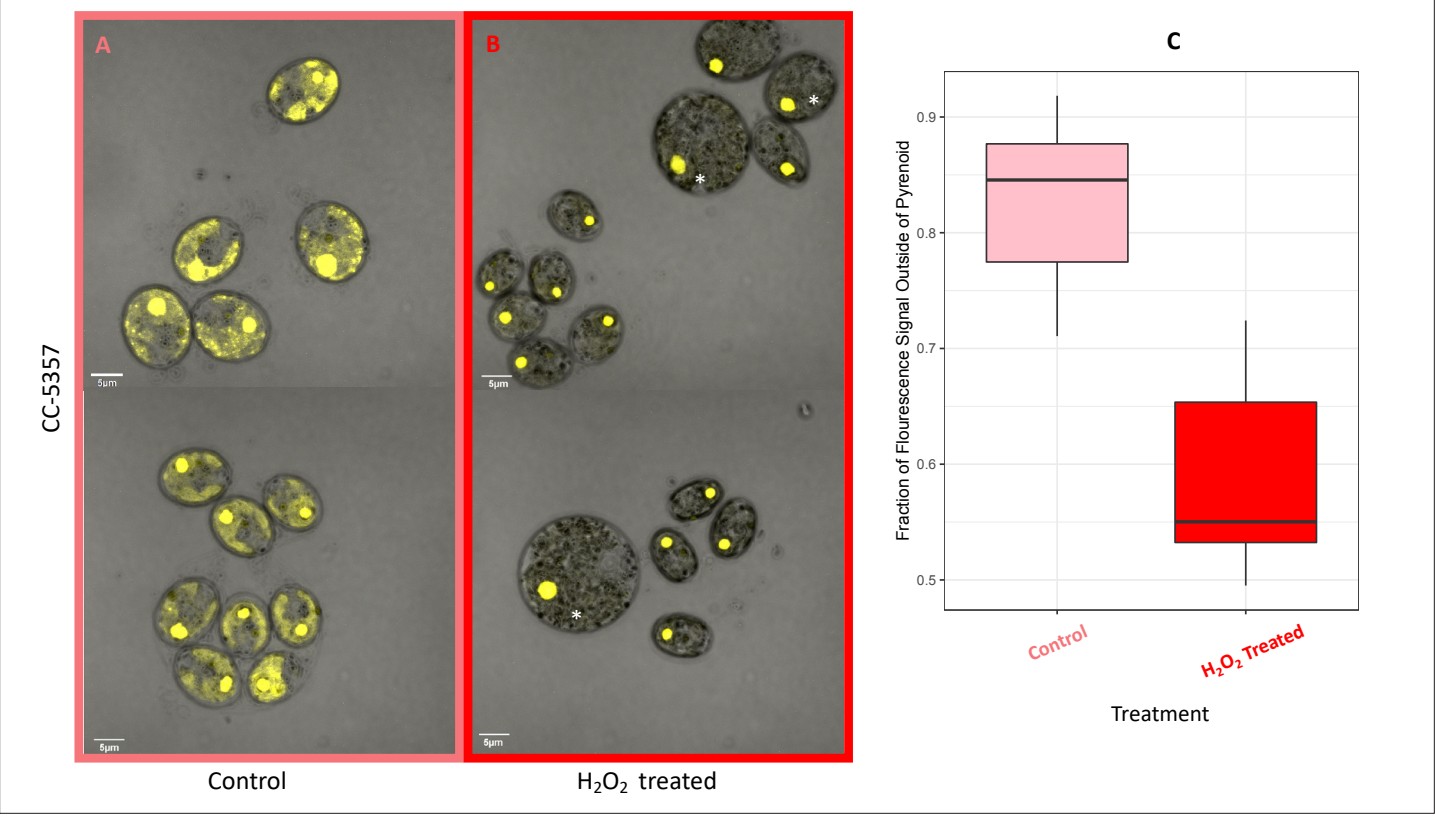

**Figure 6.** (Photos) Localization of rubisco determined by confocal microscopy of strain CC-5357, containing a RbcS1-Venus (Bar Graph, Panel **C**). Average intensity of fluorescent signal within a cell, outside of the pyrenoid region, without (**A**, minus) and with (**B**, plus) the addition of hydrogen peroxide. The average fluorescence intensity of the delocalized Venus Fluorescent Protein-labeled rubisco within *Chlamydomonas* cells was measured using the Olympus FluoView 1,000 Advanced Software. For each cell measurement, a region encircling the *Chlamydomonas* cell membrane but excluding the pyrenoid was delineated and the average fluorescence intensity within the designated region was calculated. For each treatment, measurements were performed on approximately 20 cells from three separate areas, although more areas of cells were viewed to verify the consistency of the phenotype. Cells with * were excluded from analysis to allay concerns that they may be bloated and could bias results. Fluorescence was excited using 3 % Argon gas laser intensity. Fluorescence emission was recorded through 530–630 nm band pass filter using a photomultiplier detector with a high voltage of 831. Differences were statistically significant (p < 0.001). Scale bar = 5 μm.

on TAP plates, rubisco became completely dispersed throughout the stroma, with no evidence of a pyrenoid matrix-like structure (*Figure 7*).

We next tested for differences in $H_2O_2$ production under hyperoxia in CC-2343 and CC-1009. We found that 6 and 31 hr of exposure to hyperoxia resulted in a ~ 3 fold increase in $H_2O_2$ in CC-1009, but no significant changes in CC-2343 (*Figure 8*), though CC-2343 showed a somewhat higher basal level of $H_2O_2$ on a per cell basis. *Figure 9* shows confocal laser-scanning microscope images of cells taken at steady state (*Figure 9A and B*) and at 31 hr hyperoxia (*Figure 9C and D*) and stained with 2',7'-dichlorodihydrofluorescein diacetate ($H_2$DCFDA), a general stain for reactive oxygen, sensitive to $H_2O_2$, singlet oxygen, superoxide, hydroxyl radical and various peroxide and hydroperoxides. Both cell lines accumulated ROS in response to hyperoxia. However, cells of CC-1009 showed accumulation of ROS that was highly localized in small structures (*Figure 9D*) consistent with peroxisomal microbodies (*Lauersen et al., 2016*). By contrast, CC-2343 cells showed weaker, more diffuse, staining throughout the cell, seeming to accumulate ROS throughout the thylakoids, which may be a result of rubisco inhibition, chloro-respiration, or superoxide formation (*Figure 9C*). We also observed, in CC-2343, cells uniformly stained with the $H_2$DCFDA (*Figure 9C*), reflecting severe ROS accumulation/stress in CC-2343 in subpopulations of cells, stress that did not appear to occur as much in CC-1009.

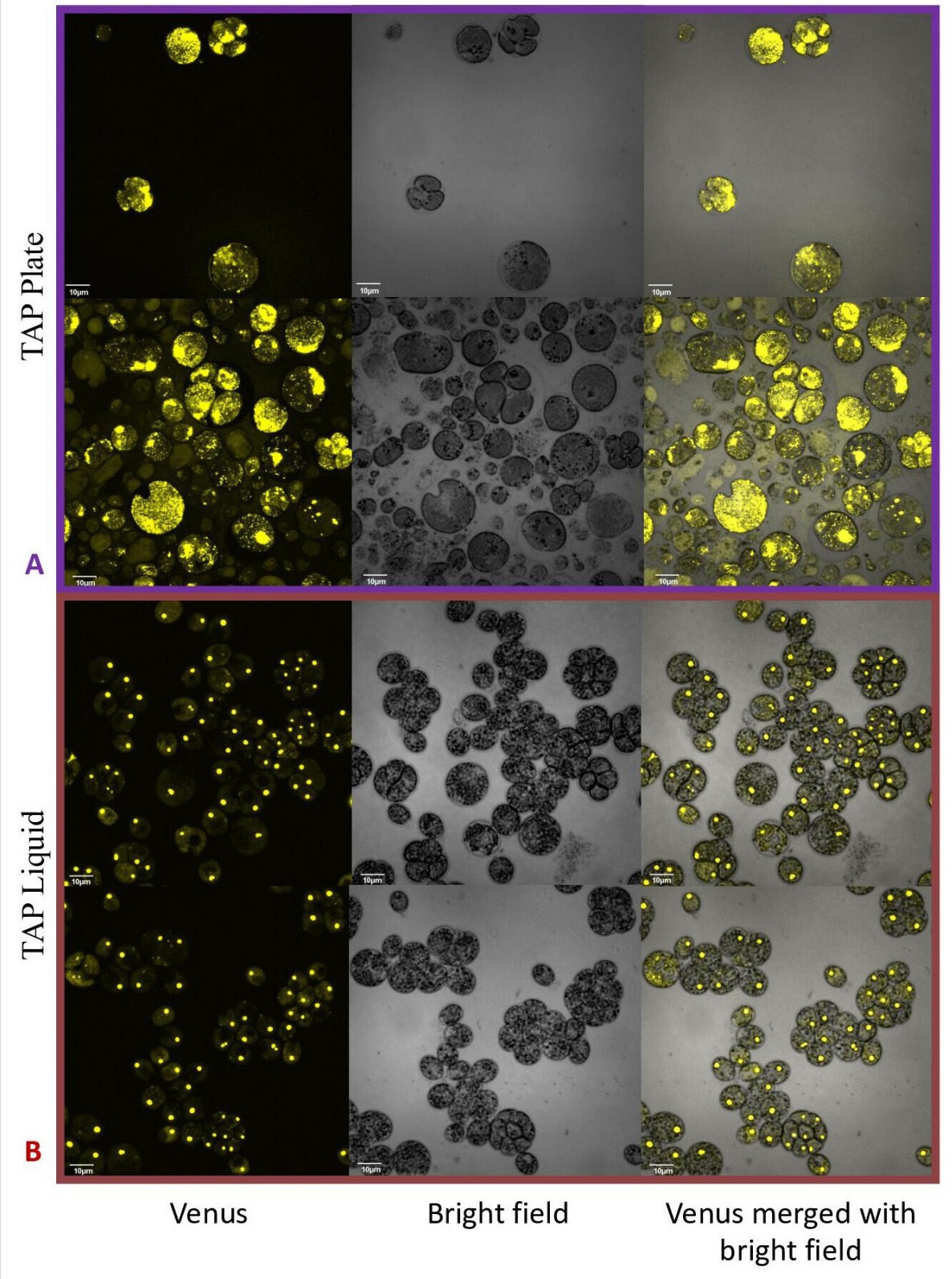

**Figure 7.** Confocal microscopy of CC-5357, which has a Venus labeled RbcS1, after being grown in on a TAP plate (Top, Panel **A**) showing rubisco completely de-localized and liquid TAP (bottom, Panel **B**), showing that rubisco has de-localized to some extent, but remains largely localized. Scale bar = 10 µm.

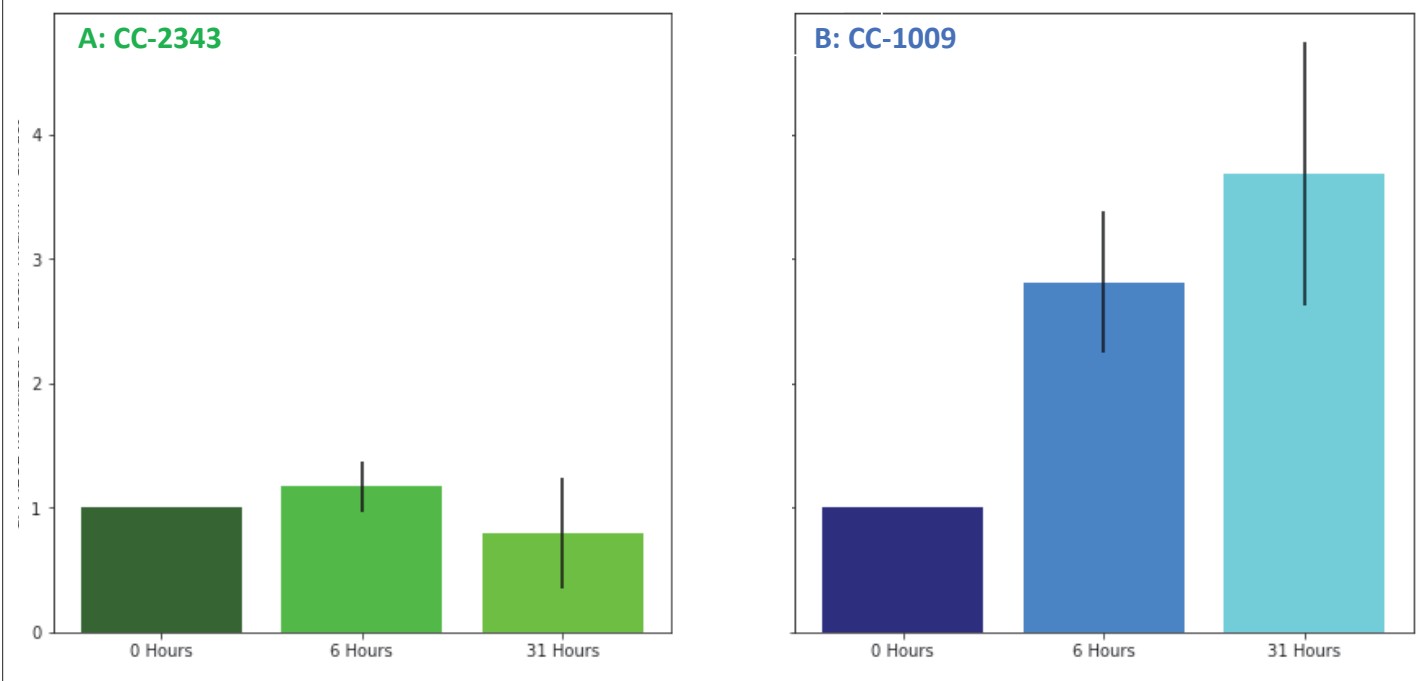

**Figure 8.** Changes in $H_2O_2$ in cellular extracts upon exposure to hyperoxia. Cells of CC-2343 (Panel **A**) and CC-1009 (Panel **B**) were rapidly broken and extracts assayed using the Amplex Red method just prior to (0 hr) and at 6 and 31 hr exposure to 95 % $O_2$, 5 % $CO_2$, as described in Materials and methods. Values shown are normalized to those taken at 0 hr, when the values normalized to the extract's protein contents were 3.37 µM for CC-2343 (Panel **A**) and 0.456 µM for CC-1009 (Panel **B**). Error bars represent the standard deviation among three biological replicates.

## Cells pre-treated with exogenous $H_2O_2$ display higher oxygen compensation points and lower $CO_2$ compensation points

*Figure 10* shows the effects of $H_2O_2$ pre-treatment on $O_2$ levels in cell suspensions of CC-1009 and CC-2343 under saturating actinic illumination. In these experiments, we tested whether $H_2O_2$-induced formation of pyrenoids with tight sheaths allowed photosynthesis to occur at higher levels of $O_2$. Prior to the traces, suspensions with 5 mM $NaHCO_3$ were sparged with air to establish low dissolved $O_2$ levels. At time zero, sparging was stopped and changes in dissolved $O_2$ were monitored with a luminescence-based $O_2$ sensor (see Materials and methods). The initial rise in $O_2$ reflects when the rate of net assimilation was maximal, under conditions when inorganic C supply was replete (5 mM $HCO_3^-$) and $O_2$ levels were low. These slopes were within 15 % of one another for both control and $H_2O_2$-treated CC-1009 (128 and 143 µM $O_2$ min$^{-1}$) and CC-2343 (104 and 110 µM $O_2$ min$^{-1}$) suspensions. After about 20 min, the rise in $O_2$ levels slowed, eventually reaching quasi-steady-state levels that represented the 'oxygen compensation point' where $O_2$ evolution from PSII was counterbalanced by $O_2$ uptake. Switching off the actinic light at ~57 min led to $O_2$ uptake, the initial rate of which likely represents the gross $O_2$ uptake, which is counterbalanced by $O_2$ evolution. Nearly equal during steady-state illumination, the two canceled each other out during the periods of light exposure. For control cells, the $O_2$ compensation points (the $O_2$ levels when the rate of $O_2$ uptake balanced that of evolution) for CC-2343 and CC-1009 were approximately 1,070 and 1230 µM $O_2$ (p < 0.05), respectively, implying, because it reaches the compensation point at a higher $O_2$ level, that CC-1009 was able to more effectively select for $CO_2$ uptake over $O_2$ reduction. The rates of uptake of $O_2$ after illumination were slightly slower in CC-2343 (36 µM $O_2$ min$^{-1}$) than CC-1009 (44 µM $O_2$ min$^{-1}$) indicating that the lower compensation point was caused by a combination of decreased linear electron flow and increased $O_2$ uptake. Strikingly, pre-treatment with $H_2O_2$ led to significant (p < 0.05) increases in the $O_2$ compensation points for both CC-2343 and CC-1009, to about 1,233 and 1356 µM $O_2$ min$^{-1}$ respectively, confirming that treated cells were better able to discriminate between uptakes of $CO_2$ and $O_2$. After the cells reached (essentially) steady-state levels of $O_2$, the actinic light was switched off. The initial slopes of $O_2$ uptake were then measured (by fitting the decay to pseudo-first

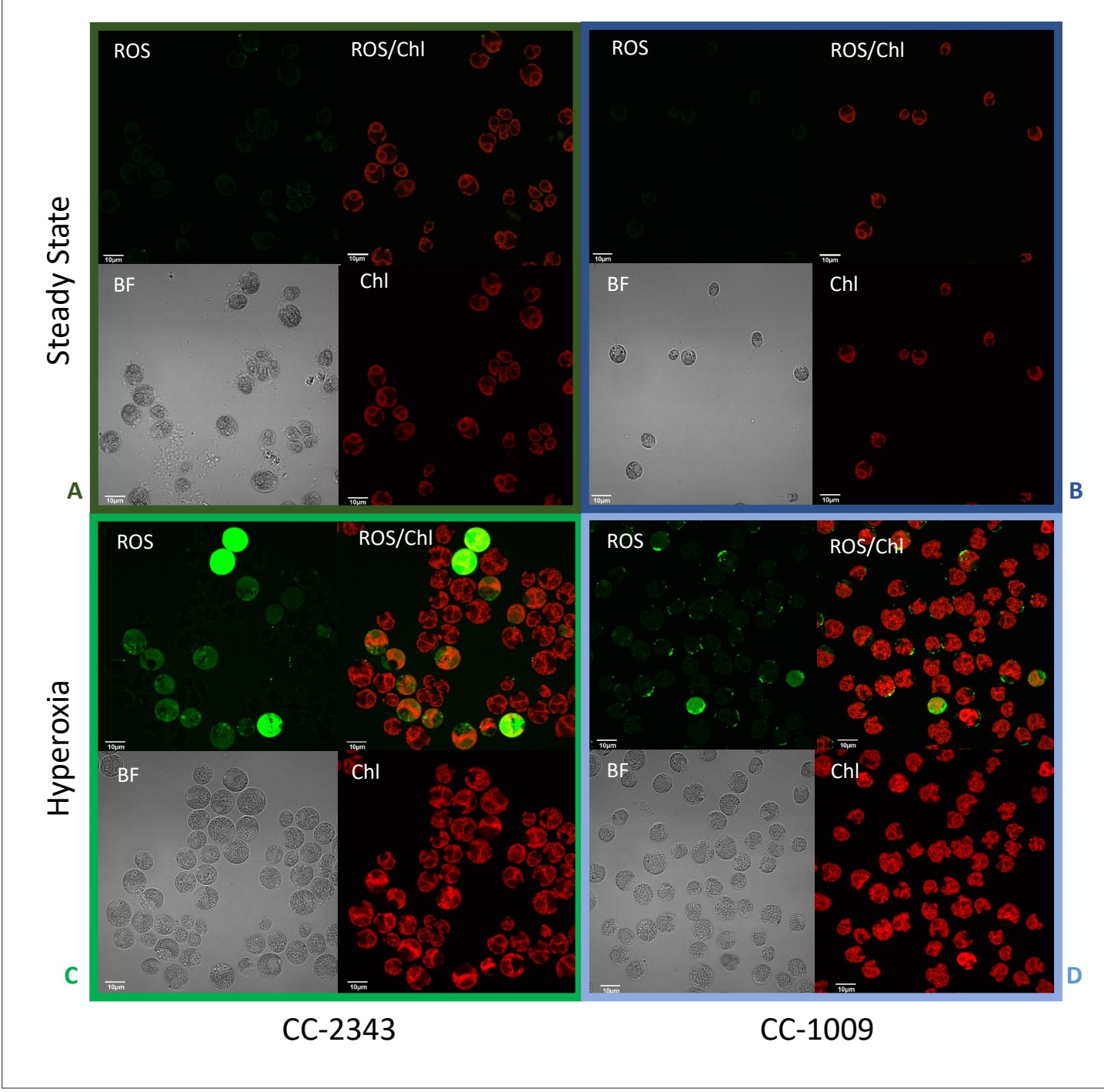

**Figure 9.** Confocal microscopy images of CC-2343 and CC-1009 showing ROS in cells growing at steady state (Panels **A** and **B**) and following exposure to approximately 31 hr of hyperoxia (Panels **C** and **D**) of CC-2343 and CC-1009. $H_2DCFDA$, a nonfluorescent probe that is converted into fluorescent dichlorofluorescein (DCF) by ROS, was used to detect the ROS. The ROS is indicated by the green fluorescence, while the auto-fluorescence of the chloroplasts is displayed in red. ROS, reactive oxygen species; Chl, chlorophyll; BF, bright field. Scale bars = 10 μm.

order decay kinetics and taking the initial rate), to give an estimate of the rates of $O_2$ evolution and uptake. CC-1009 cells showed similar $O_2$ uptake slopes in treated $H_2O_2$-treated (46 μM $O_2$ min$^{-1}$) and untreated (44 μM $O_2$ min$^{-1}$) suspensions, likely indicating that the rates of electron flow were also similar, but that the preferential fluxes of electron into assimilation allowed for a greater accumulation of $O_2$ in the treated cells. By contrast, CC-2343 cells showed a significant increase in the initial slopes

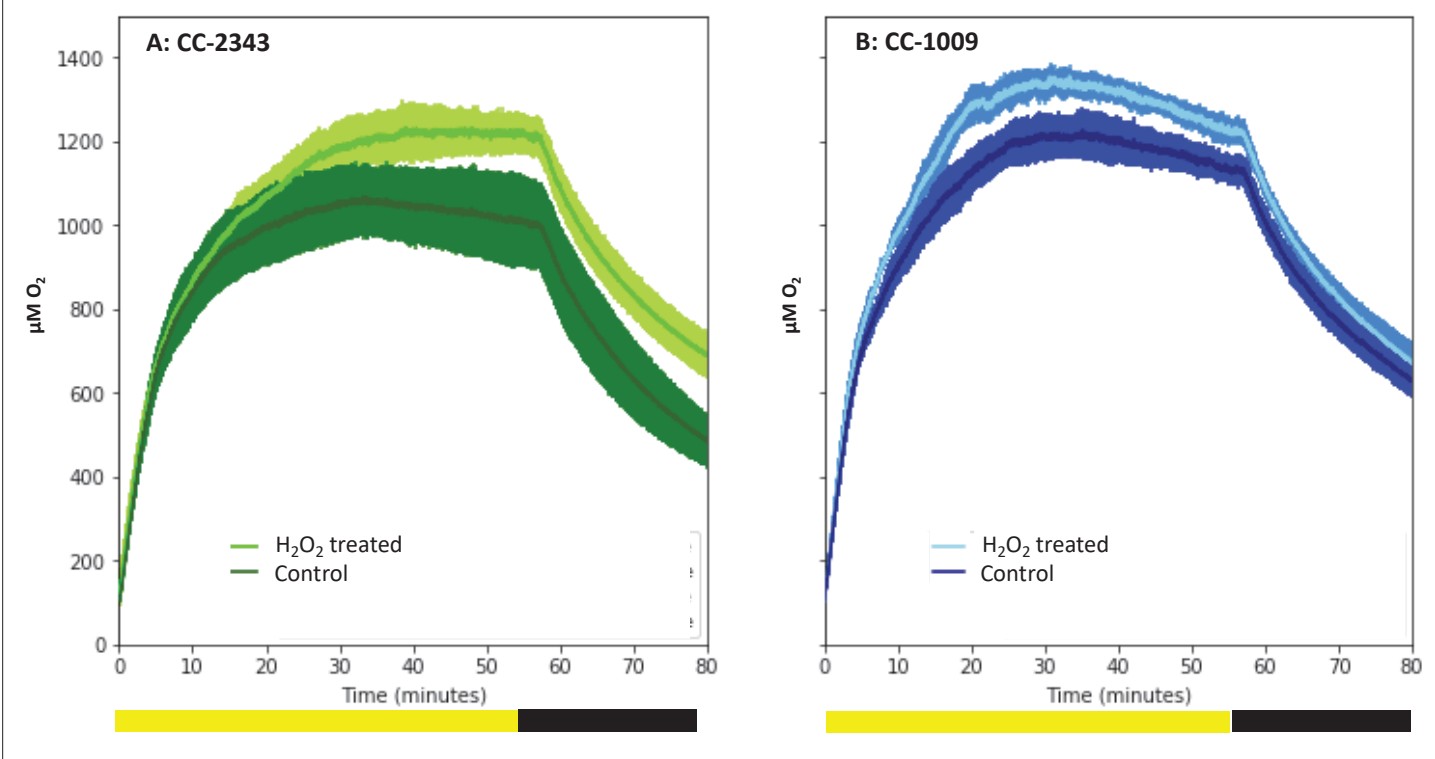

**Figure 10.** Oxygen evolution of strain CC-2343 (Panel **A**) and CC-1009 (Panel **B**) with and without the pre-treatment of 100 µM $H_2O_2$. Shading represents 95 % confidence intervals between the three biological replicates for each treatment. At approximately 3500 seconds (denoted by yellow bar on x axis), the light was turned off denoted by black bar on x axis. All measurements were done on centrifugation-concentrated cells resuspended to the same chlorophyll concentration (40 µg/ml).

of $O_2$ consumption in the treated (42 µM $O_2$ min⁻¹) compared to control (36 µM $O_2$ min⁻¹) suspensions, suggesting that the high $O_2$ levels suppressed overall rates of linear electron flow (LEF) in the untreated cells (***Supplementary file 1C***).

The above results imply that the formation of the pyrenoid after $H_2O_2$ treatment increased the ability of the cells to evolve $O_2$, either by excluding $O_2$ from or by concentrating $CO_2$ within the pyrenoid. We also measured the levels to which $CO_2$ above a cell suspension can be decreased by photosynthesis. We call this parameter the $C_i$ compensation point because it reflects the competition between assimilatory $CO_2$ uptake and the sum of photorespiration and respiration, regardless of whether the uptake occurs through diffusion and direct fixation of $CO_2$, or the active pumping of $HCO_3^-$ by the CCM (see Methods for details). As shown in ***Appendix 1—figure 22***, the apparent Ci compensation point was highest (46±4.4 ppm) for the *cia5* mutant (***Xiang et al., 2001***), which is defective in the CCM, and somewhat lower (34±4.2 ppm) in CC-5357 grown under high $CO_2$ and low light, where we expect low CCM activity, and lowest (23±4.7 ppm) in CC-5357 grown at low $CO_2$ and high light, where we expect the CCM to be fully activated. Importantly, CC-5357 cells grown under low light and high $CO_2$, but treated with $H_2O_2$, also showed a low Ci compensation point (25±4.5 ppm), indicating that the $H_2O_2$ induced pyrenoid can act effectively in the CCM. We thus conclude that the $H_2O_2$-induced pyrenoid can act to exclude $O_2$ and/or trap $CO_2$ pumped by the CCM.

## Discussion

### The induction of pyrenoid biosynthesis under hyperoxia and the role of $H_2O_2$

Photosynthesis can be inhibited by one of its major products, molecular oxygen. This is known to occur in certain aqueous environments, such as when algae ponds are enriched with $CO_2$ and photosynthesis can proceed rapidly, but diffusion of $O_2$ is slow, leading to super-saturated oxygen levels,

which can feedback limit productivity (*Livansky, 1996*; *Pulz, 2001*; *Raso et al., 2012*; *Torzillo et al., 1998*; *Vonshak et al., 1996*; *Weissman et al., 1988*). Little is known about the physiological impact of hyperoxia or the mechanisms by which some species algae are able to ameliorate its effects. Here, we took advantage of an observation that two strains of *C. reinhardtii* – and their meiotic progeny – showed distinct tolerances to hyperoxia to probe such adaptations.

Most previous work on the pyrenoid has focused on its activation by low $CO_2$ and its role in the CCM (*Borkhsenious et al., 1998*; *Freeman Rosenzweig et al., 2017*; *Mackinder et al., 2017*; *Ramazanov et al., 1994*). Both parent lines in our study and all progeny showed low $CO_2$-dependent pyrenoid formation. When cells were sparged with 5 % $CO_2$ for one minute every hour and under normoxia (~21%), the pyrenoid starch sheaths dissociated at night. This result was consistent with previous observations which demonstrated that starch formation around the pyrenoid is correlated with light and the state of the CCM (*Borkhsenious et al., 1998*; *Kuchitsu et al., 1988*; *Lin and Carpenter, 1997*; *Ramazanov et al., 1994*). Also consistent with the cited previous work, low-light with mixotrophic conditions resulted in rubisco delocalization. We further show that the most complete degree of rubisco delocalization occurs when algae is grown on a TAP agar plate exposed to air, rather than aquatically in liquid TAP (*Figure 7*). At high light, pyrenoids were formed even when $CO_2$ or bicarbonate levels were maintained at high levels (*Figure 2*), agreeing with previous assertions that light plays a role in pyrenoid biosynthesis (*Kuchitsu et al., 1988*; *Lin and Carpenter, 1997*).

Strikingly, we also observed strong pyrenoid formation, with especially tight, thick and well-sealed starch sheaths in CC-1009, c1_1, and c1_2 under hyperoxia (95 % $O_2$), despite the high $CO_2$ levels (*Figure 2*). One possible explanation for this observation is that an in-common by-product of photosynthesis and photorespiration under low $CO_2$ and hyperoxia acts to induce pyrenoid formation. Hydrogen peroxide is an obvious candidate for such a role because it is well-documented to act as a signal molecule (*Foyer et al., 2009*) and its production is increased under high light (*Roach et al., 2015*), high $O_2$ as a misfired product of oxygenation (*Kim and Portis, 2004*) or low $CO_2$ (*Foyer et al., 2009*). $H_2O_2$ is also a product of the light reactions, through an alternative electron acceptor pathway such as the Mehler peroxidase reaction (MPR) or the water-water cycle, which is expected to be more active under conditions when light input exceeds the capacity of assimilation (*Osmond et al., 2000*; *Mehler, 1951*; *Strizh, 2008*). $H_2O_2$ can also be produced as a by-product of photorespiration (*Janssen et al., 2014*). *Chlamydomonas* possesses two pathways for oxidation of glycolate during photorespiration, one involving glycolate oxidase in the peroxisome, which uses $O_2$ as an electron acceptor and produces $H_2O_2$, and another involving glycolate dehydrogenase (GLYDH) in the mitochondrion, which uses ubiquinone as an electron acceptor and presumably does not produce $H_2O_2$ (*Janssen et al., 2014*). A reasonable explanation is that, under conditions of high light, low $CO_2$ or high $O_2$ production of $H_2O_2$ by the glycolate pathway can act as a signal to induce pyrenoid formation.

We found that in autotrophically grown cells, exogenous addition of $H_2O_2$ in the presence of light strongly induced within approximately 6 hr the formation pyrenoid starch sheaths (*Figure 4*; *Figure 5*; *Appendix 1—figure 23*), and caused rubisco to localize into the pyrenoid (*Figure 6*). The $H_2O_2$ did not induce the pyrenoid when the cells were kept in the dark (*Appendix 1—figure 24*). The starch sheaths formed after addition of $H_2O_2$ had tight, thick structures in both parent lines, though CC-1009 seemed to still display slightly more robust starch plates (*Figure 4*). These structural changes were accompanied by increased $O_2$ compensation points (*Figure 10*), indicating an increased ability to discriminate between $O_2$ and $CO_2$ as $O_2$ levels increased. Our working hypothesis is that $H_2O_2$-induced formation of pyrenoids with tight sheaths allowed the accumulation of higher concentrations of $CO_2$ at the active site of rubisco, outcompeting or shielding out higher levels of $O_2$. Further, the formation of these pyrenoids enhance the discrimination of $CO_2/O_2$, implying that $H_2O_2$ induction of pyrenoids could convey performance advantages under hyperoxia. Consistent with this hypothesis, the induction of the CCM has been found to be coordinated with that of genes for photorespiratory enzymes, although the specific metabolic control of this co-regulation had remained unknown (*Tirumani et al., 2019*). Interestingly, a separate RNA expression study (*Blaby et al., 2015*) did not show strong induction of pyrenoid components by $H_2O_2$, but, importantly, was conducted on TAP-grown cells, which we found do not show $H_2O_2$-induced formation of the pyrenoid (*Appendix 1—figure 20*).

Debate remains about the signal that induces the CCM (*Spalding, 2009*; *Spalding et al., 2002*; *Vance and Spalding, 2005*), which consists not only of the pyrenoid but also the inorganic carbon transporters (*Spalding, 2008*) and carbonic anhydrases (*Moroney et al., 2011*); some have argued

that the signal is $CO_2$ itself, while others have proposed that the signal is a metabolite produced under low $CO_2$ during photosynthesis or photorespiration. The later, termed the 'metabolic signal hypothesis,' (*Spalding, 2009*) proposed that photorespiratory intermediates could serve as a trigger for CCM induction (*Marcus et al., 1983*; *Suzuki et al., 1990*). The hypothesis was rooted in observations that, unlike wild type cells, various photosynthetic mutants did not exhibit CCM activity under low $CO_2$, and that CCM induction in wild type cells required light (*Dionisio et al., 1989a*; *Dionisio et al., 1989b*; *Dionisio-Sese et al., 1990*; *Spalding and Ogren, 1982*; *Spencer et al., 1983*; *Tirumani et al., 2014*; *Villarejo et al., 1996*). Also implying that other factors, apart from $CO_2$, played a role in CCM induction, decreased $O_2$ tension and photorespiratory inhibitors, in low $CO_2$ conditions, also decreased carbonic anhydrase induction (*Ramazanov and Cardenas, 1992*; *Spalding and Ogren, 1982*; *Villarejo et al., 1996*).

*Bozzo et al., 2000* argued against the metabolite hypothesis in *Chlorella*, based on observations that: (1) photorespiratory inhibitors, which should result in an accumulation of photorespiration intermediates, failed to induce the CCM under high $CO_2$ and (2) the expression of transcripts for a subset of carbonic anhydrases increased under low $CO_2$ even in the dark (although to a lesser extent than in the light). Similar results have been found in several *Chlorella* species (*Matsuda and Colman, 1995*; *Shiraiwa and Miyachi, 1985*; *Umino et al., 1991*), suggesting that the induction of at least some CCM components can occur in the dark. However, it is unclear how relevant these results are, considering that the pyrenoid is not formed in the dark (*Kuchitsu et al., 1988*; *Lin and Carpenter, 1997*). It was also found in *Chlamydomonas* that changing $O_2$ levels (from 2% to 20%) did not affect growth, photosynthetic rate, or the induction of periplasmic carbonic anhydrase (Cah1) or glycolate dehydrogenase (Gdh) genes, over a wide range of $CO_2$ levels (*Vance and Spalding, 2005*). It is worth noting, though, that none of these previous experiments were conducted under true hyperoxia ($O_2$ levels above partial pressure of 21%), where we observe strong induction of the pyrenoid, and thus they do not exclude product signaling under our observed conditions.

There remains the possibility of multiple signals for the CCM. There is differential regulation of low $CO_2$ induced polypeptides in *Chlamydomonas*, with some only being induced in the light, while for others light is not necessary (*Villarejo et al., 1996*). Also, the observation that there are multiple acclimation states, with some mutants tolerant to very low $CO_2$ but not low $CO_2$, suggests the existence of multiple types of signaling (*Spalding et al., 2002*). Our findings that a key aspect of the CCM, the pyrenoid, can be induced, even under high $CO_2$ (i.e. with hyperoxia and $H_2O_2$) disproves, to our knowledge for the first time, that low $CO_2$ is a necessary condition for any aspect of CCM induction. Our results lead us to propose that $H_2O_2$, a by-product of photosynthesis, particularly under low $CO_2$ and high $O_2$, may fulfill the previous proposed 'metabolic' signal. Hydrogen peroxide is widely known to be a signal for a variety of stress related responses (*Blaby et al., 2015*; *Zalutskaya et al., 2019*), and has been found to alter the state of redox homeostasis in *Chlamydomonas* (*Pokora et al., 2018*). It has also been assigned roles in regulating a range of photosynthetic and associated processes in plants and algae (*Berens et al., 2019*; *Foyer and Noctor, 2009*), particularly those related to responses to $CO_2$ levels and the induction of photorespiration (*Foyer et al., 2009*). Interestingly, in higher plants, $H_2O_2$ has been suggested to play a role in the response to varying levels of $CO_2$ (*Foyer et al., 2009*). For example, *Sorghum* ($C_4$) plants grown under conditions with lower amounts of photorespiration (i.e. elevated $CO_2$) have decreased thickening of the bundle sheath cells (*Watling et al., 2000*), which, since they restrict the diffusion of $CO_2$ out of bundle sheath cells and thereby allow for efficient capture by rubisco, can be interpreted as structures analogous to the starch sheath of the pyrenoid.

Hydrogen peroxide has also been implicated in regulating cyclic electron flow (CEF) in vascular plants, both by inducing the expression of the thylakoid Complex I (or NDH) (*Casano et al., 2001*; *Gambarova, 2008*) and by activation of existing enzymes (*Strand et al., 2015*). It is not known, however, if $H_2O_2$ acts directly as a signaling agent, or indirectly, for example by altering the activities of assimilatory enzymes (*Strand et al., 2015*) possibly through redox balancing enzymes such as the peroxiredoxins (*Vaseghi et al., 2018*). CEF is thought also to play central role in providing the energy needed to power CCMs, including that in *Chlamydomonas* (*Lucker and Kramer, 2013*). Our findings indicating that $H_2O_2$ may be the signal for the synthesis of a central component of the CCM, the pyrenoid, suggests that a common molecular by-product of photorespiration can set off a coordinated response; inducing the formation of the pyrenoid and also the metabolic processes to power the pumping of bicarbonate into it.

It has been argued that mixotrophic conditions alter the relationship between the onset of photorespiration and the expression of the CCM (*Tirumani et al., 2019*), and that photorespiration, hydrogen peroxide detoxification, and acetate assimilation (i.e. the glyoxolate cycle) are all localized in the peroxisomal microbodies (*Lauersen et al., 2016*). In this regard, it is intriguing that ROS labeling under hyperoxia was strongly localized in CC-1009 but not CC-2342 (*Figure 9*), hinting that $H_2O_2$ produced in a specific subcellular location and process may play a role in the differential development of the pyrenoid in the two parent lines, as discussed below. Taken together, these data sets are consistent with control of pyrenoid morphology at multiple levels, perhaps similar to the processes that regulate the expression of LHCSR3, involved in photoprotective nonphotochemical quenching, which is regulated by light quality and $CO_2$ availability (*Maruyama et al., 2014*; *Semchonok et al., 2017*). Future studies can also investigate how hyperoxia plays a role in the gene regulatory network for antennae size, which has been shown to be affected by low $CO_2$ (*Blifernez-Klassen et al., 2021*).

## Possible linkages between $H_2O_2$ signaling, pyrenoid morphology and natural variations in responses to hyperoxia

By comparing genetically distinct isolates and their progeny, one can potentially explore possible mechanistic bases for responses to hyperoxia. We present here data from a limited set of progeny, which nevertheless reveals a segregation pattern which allows us to test certain future hypotheses. A more detailed analysis of a large number of progeny will be presented elsewhere. The most striking differences we observed between the lines were in the morphology of the pyrenoids (*Figure 2*), with the hyperoxia tolerant lines (CC-1009, c1_1, c1_2) showing thick, tightly sealed starch sheaths, while the sensitive lines (CC-2343, c1_3, c1_4) tending to have pores or gaps in the starch sheaths, suggesting that structural/functional differences in these sub-organelle compartments may play a role in the distinct responses to high $O_2$. That the miotic progeny with reduced biomass accumulation and fractured starch sheaths exhibited 2:2 segregation suggests that the phenotype variations were due to allelic differences in the nuclear genes. These differences appear to be most obvious during hyperoxia, and all lines showed disappearance of the pyrenoid structures under high $CO_2$/low light (*Appendix 1—figure 10*). Most interestingly, exogenous $H_2O_2$ led to synthesis of thick, tight pyrenoids in all lines, implying that the distinct morphologies is caused at least in part from differences in signaling, rather than structural components.

Given the possibility that $H_2O_2$ acts as a signal for pyrenoid biosynthesis, we tested for differences in its production under hyperoxia. Only in CC-1009 does $H_2O_2$ increase under hyperoxia (*Figure 8*). Furthermore, the localization of ROS production assessed by $H_2DCFDA$ fluorescence in confocal microscopy showed distinct localization patterns, with CC-1009 showing strongly localized dye fluorescence (*Figure 9B and D*), whereas CC-2343 showed diffuse staining throughout the cell (*Figure 9A and C*). Because the $H_2DCFDA$ is a general ROS indicator, it is not possible to unambiguously identify the specific reactive oxygen species, but one possible interpretation is that different localization patterns reflect the mechanism of ROS formation. The localized staining in CC-1009 is consistent with $H_2O_2$ produced in the peroxisome through photorespiration. By contrast, in CC-2343, the diffuse staining may reflect a range of different ROS, including but not limited to $H_2O_2$, $^1O_2$ and $O_2^{\bullet-}$ produced by excitation of the light reactions and other processes (*Osmond et al., 2000*).

We have several hypotheses regarding why the two lines may have differences in the signaling and formation of their pyrenoid starch sheaths. One is that there might be variations in the strains' utilization of the alternative photorespiration route that uses the glycolate dehydrogenase (GLYDH) enzyme, a route which does not result in hydrogen peroxide formation (*Janssen et al., 2014*).

Similarly, in the future we will investigate how the pyrenoid ameliorates the stresses of hyperoxia, with possibilities beyond photorespiration. Our rubisco assays (*Figure 1*) suggest that increased $O_2$ fixation or ROS production may lead to greater inhibition of rubisco in CC-2343 compared to CC-1009, possibly leading or concomitant to a general breakdown of the cell's machinery, as evidenced by the lower autotrophic grow rates of CC-2343 at high $O_2$ (*Figure 3*) and lower rates of oxygen evolution (*Figure 10*). Such a model is also consistent with the diffuse ROS staining observed in CC-2343, as the mismatch in light input and downstream assimilatory capacity could result in the accumulation of not just $H_2O_2$, but also $^1O_2$ and $O_2^{.-}$ (*Peng et al., 2016b*), forms of ROS that may reflect high levels of oxidative damage.

## Eco-physiological implications

For over a hundred years it has been known that *Chlamydomonas* strains show distinct pyrenoid structures (*Pasher, 1918*), although the physiological implications of these natural variations remain poorly understood. A few studies have noted structural differences in pyrenoid starch sheaths, and linked these differences to environmental $CO_2$ or organic carbon availability (*Morita et al., 1998*; *Morita et al., 1999*; *Nozaki et al., 1994*).

As discussed above, it is well established that the pyrenoid can allow algae to overcome critical limitations of low $CO_2$ levels often encountered in aqueous environments. However, under very high $CO_2$ levels, which are also encountered in certain environments, the sequestering of rubisco into the pyrenoid may impose rate limitations, or additional energy requirements, at the level of pumping of bicarbonate. Also, when rubisco is outside of the pyrenoid, it is thought that more of its surface area is exposed and its catalytic rate increases (*Badger et al., 1998*). A fragmented starch sheath may more easily allow migration in and out of the pyrenoid matrix. In two species of *Gonium*, the species with the more porous starch sheaths exhibited a higher ratio of rubisco migrating out of the pyrenoid in response to the addition of sodium acetate (*Nozaki et al., 1994*). Among closely related *Chlamydomonas* and *Chloromonas* strains, those with tight (which were termed 'typical') pyrenoids were able to accumulate higher levels of inorganic carbon when $CO_2$ was low compared to those with fragmented or porous (termed 'atypical') pyrenoid starch sheaths (*Morita et al., 1999*). On the other hand, *Chloromonas* species closely related to *Chlamydomonas* but lacking pyrenoids showed higher rates of max $O_2$ evolution when grown under elevated $CO_2$ (*Morita et al., 1998*), which could be attributed to the greater accessibility of rubisco to diffusible $CO_2$.

Some algae lack pyrenoids altogether and are found in environments expected to have high $CO_2$ and low or atmospheric oxygen levels. For example, *Coccomyxa*, an aerial grown lichen photobiont, completely lacks pyrenoids (*Palmqvist et al., 1994*; *Palmqvist et al., 1995*). Compared to that in *Chlamydomonas*, *Coccomyxa* prefers $CO_2$ as a substrate over $HCO_3^-$, similar to $C_3$ plant cells (*Palmqvist et al., 1994*; *Palmqvist et al., 1995*). It is important to note, though, that exposure to air allows for rapid diffusion of $O_2$: Even high rates of photosynthesis in *Coccomyxa* will not result in hyperoxia. In light of these studies, it seems fitting that CC-2343 and the progeny with porous pyrenoids grew better on a TAP plate exposed to air, and that rubisco most freely distributes through the chloroplast in *Chlamydomonas* when grown mixotrophically exposed to air, rather than aquatically (*Appendix 1—figure 7*).

By contrast, green algae can generate strongly hyperoxic conditions in the water specifically when inorganic carbon is plentiful. Our demonstration that pyrenoids are induced under these conditions suggests that they can function, in addition to overcoming slow assimilation when $CO_2$ is limiting, in preventing damage caused by high levels of the product $O_2$. Inducing the CCM should both increase the concentration of $CO_2$ above its $K_M$ at rubisco and outcompete $O_2$ at the rubisco active site. Higher $O_2$ levels (under hyperoxia) will require correspondingly higher local $CO_2$ levels, in turn requiring tighter diffusional barriers to the escape of $CO_2$ from the pyrenoid (*Wang et al., 2015*; *Yamano et al., 2015*). It is also possible that the tight starch sheaths will partially block $O_2$ from diffusing into the pyrenoid, and if the uptake of $O_2$ by rubisco is faster than its replacement by diffusion across the sheath, such a barrier could effectively decrease the $O_2$ levels in the matrix.

## Conclusions

The work presented above leads us to propose that, under combinations of light, high $O_2$ and/or low $CO_2$, the production of $H_2O_2$ becomes elevated, activating the formation of the pyrenoid and thickening of the starch sheath, leading to the classical response that allows cells to better discriminate between $CO_2$ and $O_2$ (*Aizawa and Miyachi, 1986*; *Badger et al., 1980*; *Borkhsenious et al., 1998*; *Manuel and Moroney, 1988*; *Ramazanov et al., 1994*; *Spalding et al., 1983*). We demonstrate that the pyrenoid, a key component of the algal CCM, can be induced under high $CO_2$, by hyperoxia or $H_2O_2$. Our results strengthen the 'metabolite signaling hypothesis,' (*Spalding, 2009*), which can explain the regulation of pyrenoid formation by multiple photosynthetic conditions, including $CO_2$, $O_2$, and its light dependence. Our results further suggest that differences in this signaling contribute, at least in part, to the observed natural varaition in pyrenoids (*Pasher, 1918*) as well as tolerances to hyperoxia. Several open questions remain, including whether a $H_2O_2$ signal works alone or in conjunction with a $CO_2$ signal for some aspects of the CCM, the precise nature and scope of the $H_2O_2$

response, the biochemical and genetic components involved, and whether more robust pyrenoid structures, by themselves, can improve growth under hyperoxia.

## Additional information

### Competing interests

Joseph Weissman: Joseph Weissman is affiliated with ExxonMobil. The author has no financial interests to declare. David M Kramer: Reviewing editor, *eLife*. The other authors declare that no competing interests exist.

### Funding

| Funder | Grant reference number | Author |
|---|---|---|
| U.S. Department of Energy | DE-FG02-91ER20021 | Peter Neofotis<br>Oliver L Tessmer<br>Jacob Bibik<br>Nicole Norris<br>Eric Pollner<br>Ben Lucker<br>Sarathi Wijetilleke<br>Alecia Withrow<br>Greg Mogos<br>David Hall<br>David M Kramer<br>Joshua Temple |
| ExxonMobil Research and Engineering Company | | Peter Neofotis<br>Oliver L Tessmer<br>Jacob Bibik<br>Nicole Norris<br>Eric Pollner<br>Ben Lucker<br>Sarathi Wijetilleke<br>Alecia Withrow<br>Greg Mogos<br>Melinda Frame<br>Joseph Weissman<br>David M Kramer<br>Joshua Temple |
| AgBioResearch, Michigan State University | | David M Kramer |

The funders had no role in study design, data collection and interpretation, or the decision to submit the work for publication.

### Author contributions

Peter Neofotis, Conceptualization, Data curation, Formal analysis, Investigation, Methodology, Supervision, Visualization, Writing – original draft, Writing – review and editing; Joshua Temple, Formal analysis, Investigation, Methodology, Software, Validation, Visualization, Writing – original draft, Writing – review and editing; Oliver L Tessmer, Data curation, Formal analysis, Software, Visualization; Jacob Bibik, Resources, Writing – review and editing; Nicole Norris, Barbara Sears, Greg Mogos, Investigation; Eric Pollner, Investigation, Writing – review and editing; Ben Lucker, Methodology, Resources, Writing – review and editing; Sarathi M Weraduwage, Formal analysis, Investigation, Methodology, Writing – original draft, Writing – review and editing; Alecia Withrow, Visualization; Melinda Frame, Formal analysis, Visualization, Writing – original draft, Writing – review and editing; David Hall, Investigation, Visualization, Writing – original draft, Writing – review and editing; Joseph Weissman, Funding acquisition, Investigation, Methodology, Writing – review and editing; David M Kramer, Conceptualization, Investigation, Project administration, Writing – original draft, Writing – review and editing

### Author ORCIDs

Peter Neofotis (iD) http://orcid.org/0000-0002-0360-9933
Joshua Temple (iD) http://orcid.org/0000-0002-9295-1422

David M Kramer ![ORCID] http://orcid.org/0000-0003-2181-6888

## Decision letter and Author response
Decision letter https://doi.org/10.7554/eLife.67565.sa1
Author response https://doi.org/10.7554/eLife.67565.sa2

---

# Additional files

## Supplementary files
- Transparent reporting form
- Supplementary file 1. Supporting and additional data and figures.

## Data availability
The raw images and data for all our figures can be found on our github site (https://github.com/protonzilla/Neofotis2021_Pyrenoid_Hyperoxia copy archived at https://archive.softwareheritage.org/swh:1:rev:7eb4a1b1118aa081ab140fd5ced951164f3ea66c).

The Transparent Reporting Images can also be found on the github site, (https://github.com/protonzilla/Neofotis2021_Pyrenoid_Hyperoxia/blob/main/PyrenoidPaper_TransparentReportingImages.pptx)

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

## Appendix 1

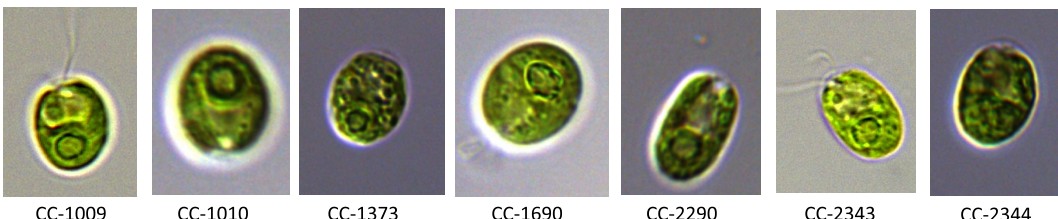

| CC-1009 | CC-1010 | CC-1373 | CC-1690 | CC-2290 | CC-2343 | CC-2344 |

**Appendix 1—figure 1.** Exemplary pictures of ecotypes which were examined in this study showing various pyrenoid formations, with CC-1009 and CC-1010 exhibiting "closed" types. Cultures were grown within flasks in minimal (2NBH) media under 50 µmoles $m^{-2}$ $s^{-1}$ of PAR.

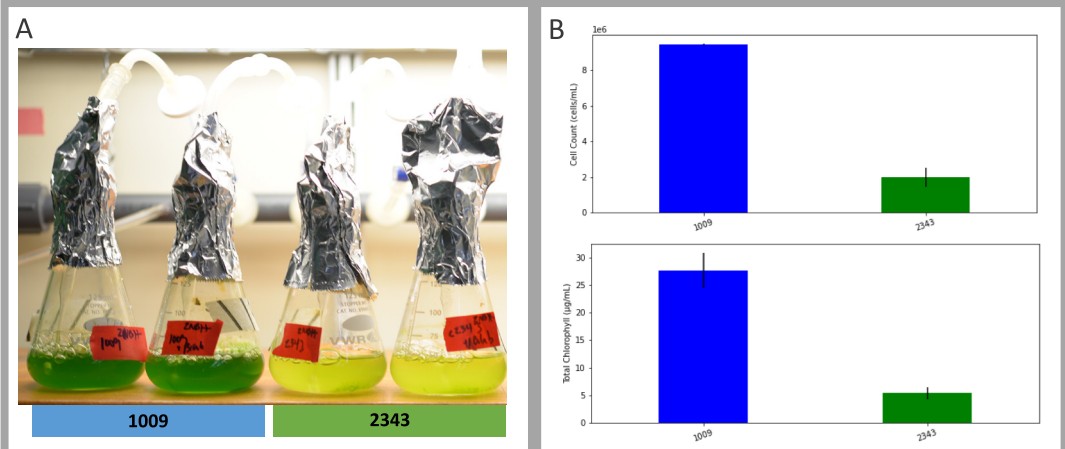

**Appendix 1—figure 2.** Panel (**A**) Photo of five days after 50 ml of fresh 2NBH media was inoculated to $1 \times 10^5$ cells/ml, and cultures were continuously bubbled with 5 % $CO_2$ and 95 % $O_2$. Panel (**B**) Graph of cell counts and chlorophyll content. Error bars represent the standard deviation of the two biological replicates.

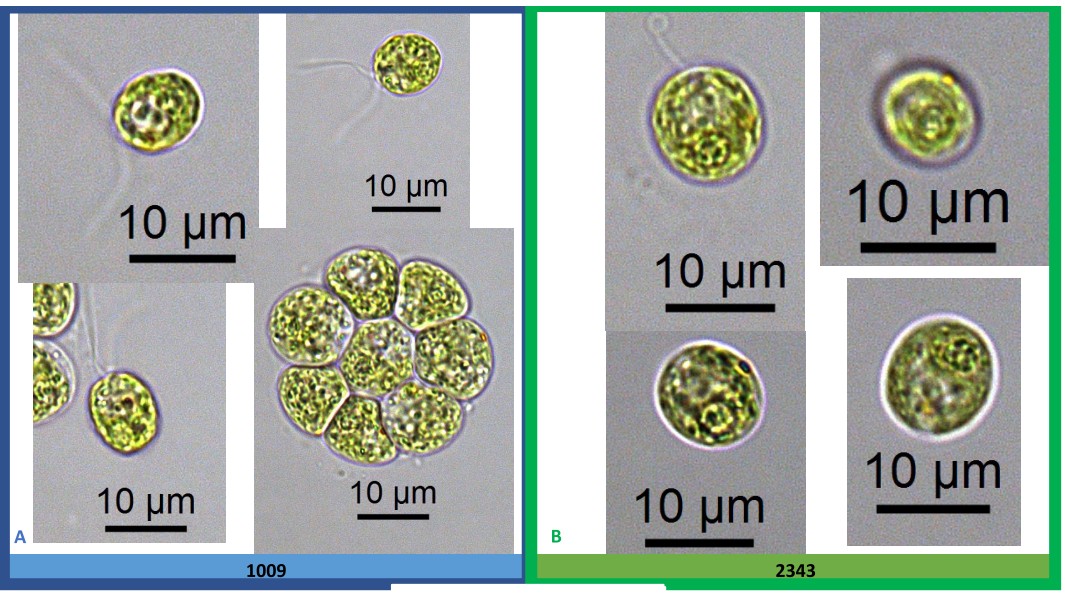

**Appendix 1—figure 3.** Exemplary light microscopy images of *Chlamydomonas* strains CC-1009 (Panel **A**) and CC-2343 (Panel **B**), while growing under saturating $CO_2$. Cells are growing with 5 % $CO_2$ with one minute on/one minute off sparging, and 14:10 hour (light:dark) sinusoidal illumination with peak light intensity of 2000 µmoles $m^{-2}$ $s^{-1}$, in minimal 2NBH media. Cells here were viewed at approximately noon, at 2000 µmoles $m^{-2}$ $s^{-1}$.

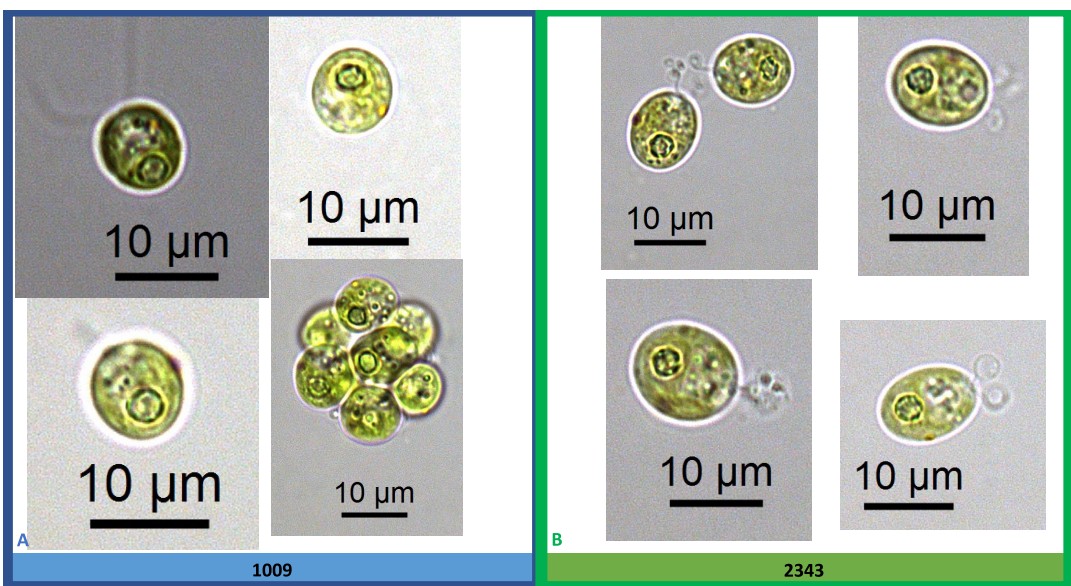

**Appendix 1—figure 4.** Exemplary light microscopy images of *Chlamydomonas* strains CC-1009 (Panel **A**) and CC-2343 (Panel **B**), after growing under saturating $CO_2$ and hyperoxia for over 6 hours. Pyrenoid starch sheaths were clearly visible. Cells are sparged with 5 % $CO_2$ and 95 % $O_2$, with one minute on/one minute off sparging, and 14:10 hour (light:dark) sinusoidal illumination with peak light intensity of 2000 µmoles $m^{-2}$ $s^{-1}$, in minimal 2NBH media. Prior to switching the gas to hyperoxia (i.e. 95 % $O_2$ and 5 % $CO_2$) cells had been grown in steady state conditions. Cells here were viewed around 1:00 pm, at ~1945 µmoles $m^{-2}$ $s^{-1}$.

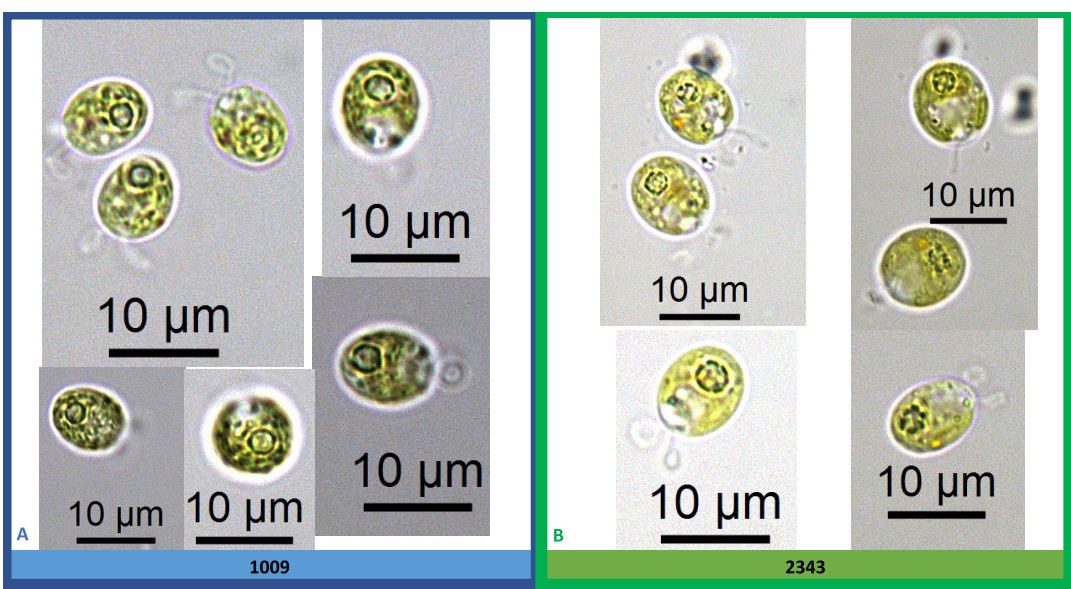

**Appendix 1—figure 5.** Exemplary light microscopy images of *Chlamydomonas* strains (CC-1009 & CC-2343), after growing under saturating $CO_2$ and hyperoxia for 31 hours. Differences were observed between the pyrenoids of the cells in Panel A and Panel B, with the cells in Panel A showing more robust, continuous, sealed pyrenoids. Cells are sparged with 5 % $CO_2$ and 95 % $O_2$, with one minute on/one minute off sparging, and 14:10 hour (light:dark) sinusoidal illumination with peak light intensity of 2000 µmoles $m^{-2}$ $s^{-1}$, in minimal 2NBH media. Prior to switching the gas to hyperoxia (i.e. 95 % $O_2$ and 5 % $CO_2$) cells had been grown in steady state conditions. Cells here were viewed at noon, at 2000 µmoles $m^{-2}$ $s^{-1}$.

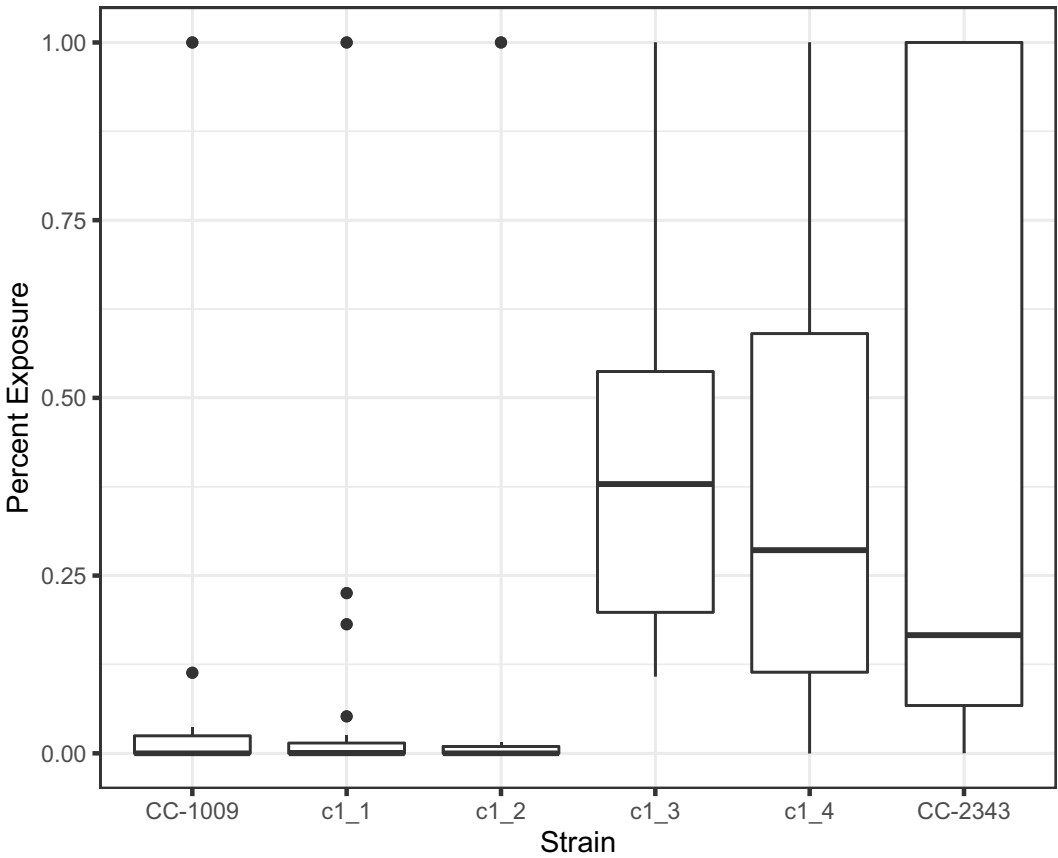

**Appendix 1—figure 6.** Measurement of percent of pyrenoid with gaps present in respective cell lines at 6 hours hyperoxia. Statistically significant differences ($P < 0.05$) were found when comparing any of the lines with continuous pyrenoids (CC-1009, c1_1, and c1_2) to the lines with porous pyrenoids (CC-2343, c1_3, c1_4). At least 24 randomly selected cells were analyzed of each strain.

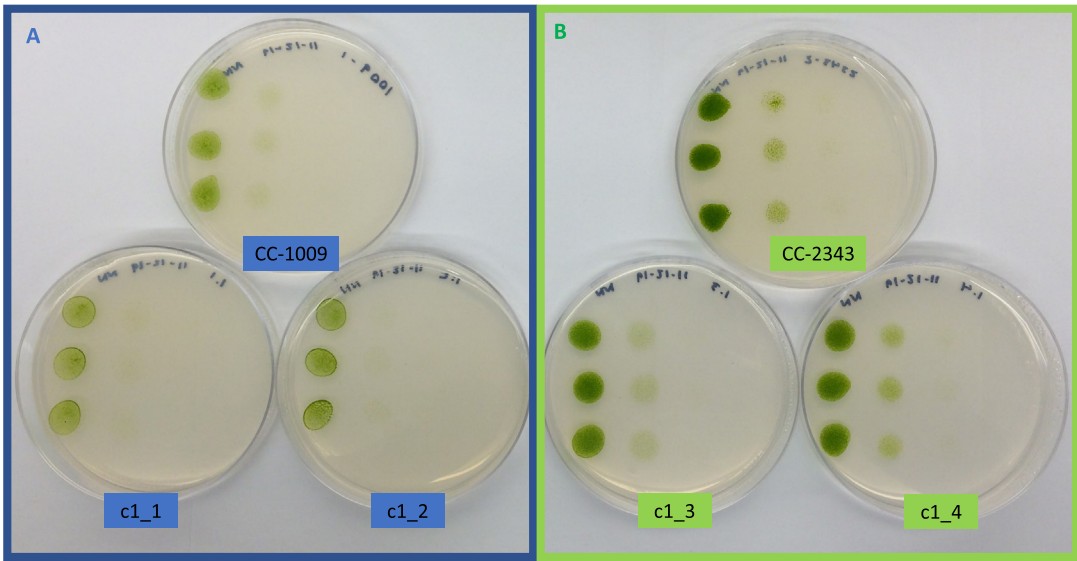

**Appendix 1—figure 7.** Growth of parents (CC-1009 and CC-2343) and F1 tetrad offspring following the serial dilution, by column, of a cultures on TAP agar plates. Rows are replicate dilutions. Photo was taken 3 days after plating. The oxygen intolerant lines (Panel B) all clearly grew better than the tolerant line (Panel A). See *Figure 3* for graph of oxygen tolerance.

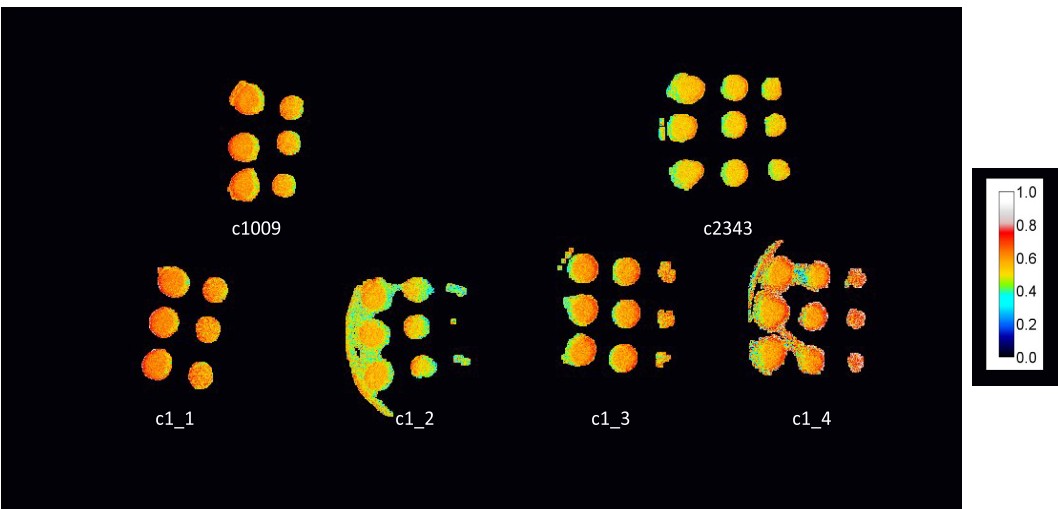

**Appendix 1—figure 8.** $\Phi_{II}$ values of spotted plates of parents and tetrad after one week, with c1_3, and c1_4 not showing any obvious higher level of electron transport, despite higher growth rates. Measurements shown here were under 50 µmoles m$^{-2}$ s$^{-1}$ PAR, though similar results were obtained up to 1500 µmoles m$^{-2}$ s$^{-2}$.

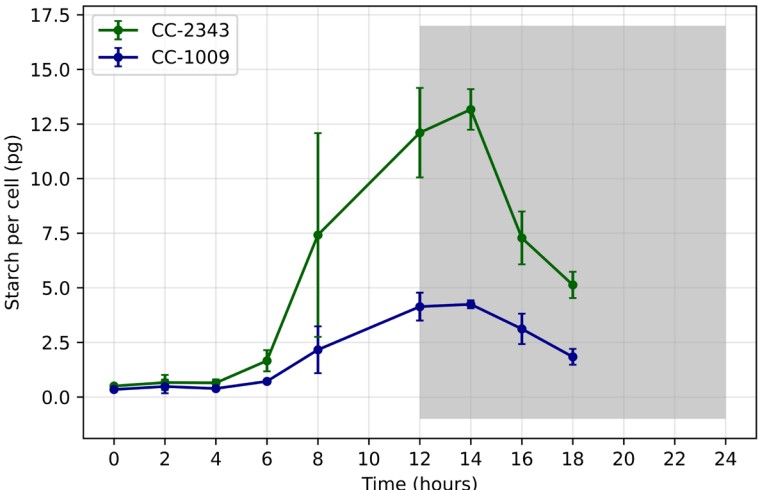

**Appendix 1—figure 9.** Total starch accumulated (pg/cell) in CC-1009 and CC-2343 cultures grown at steady state conditions, with 5 % $CO_2$ with 14:10 hour (light:dark) sinusoidal illumination with peak light intensity of 2000 µmoles m$^{-2}$ s$^{-1}$, in minimal 2NBH media.

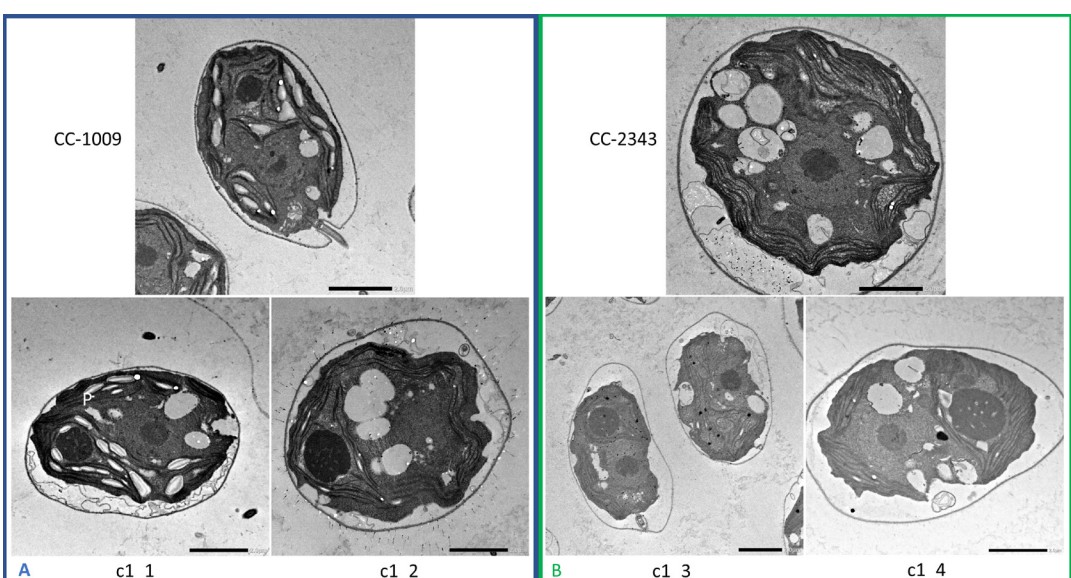

**Appendix 1—figure 10.** Representative TEM images of *Chlamydomonas* strains, the parents (CC-1009 & CC-2343), as well as their progeny c1_1, c1_2, c1_3, c1_4, having been sparged with $CO_2$ for 30 seconds every hour during the night. Under these conditions, both the strains in Panel **A** and **B** have not formed a pyrenoid starch sheath. Cells had been grown in steady state conditions in minimal 2NBH media, with 5 % $CO_2$ in air with 14:10 hour (light:dark) sinusoidal illumination with peak light intensity of 2000 µmoles $m^{-2}$ $s^{-1}$. Cells here were fixed two hours after dawn, at 7:00 am, when light levels were 825 µmoles $m^{-2}$ $s^{-1}$. Pyrenoids appeared absent or mostly unsheathed in these conditions. Scale bar = 2 µm.

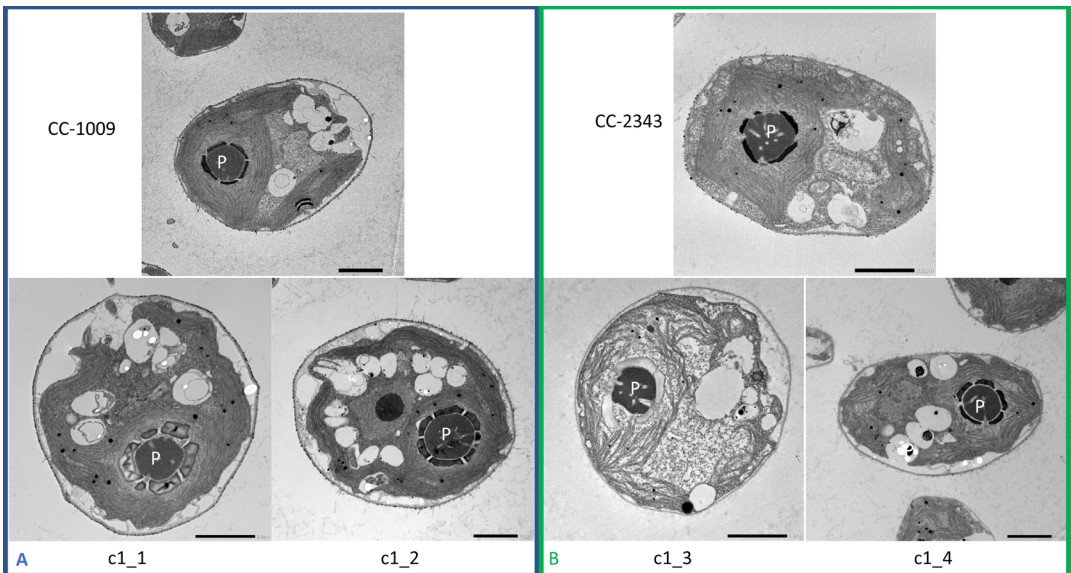

**Appendix 1—figure 11.** Representative TEM microscopy images of *Chlamydomonas* strains, the parents (CC-1009 & CC-2343), as well as their progeny c1_1, c1_2, c1_3, c1_4. Images taken 31 hours after the culture had been switched to from 5 % to low (i.e. ambient) levels of $CO_2$. Prior to switching the gas to ambient levels of $CO_2$, cells had been grown in steady state conditions, with 5 % $CO_2$ with 14:10 hour (light:dark) sinusoidal illumination with peak light intensity of 2000 µmoles $m^{-2}$ $s^{-1}$, in minimal 2NBH media. At this time point, the differences in the pyrenoids were not as apparent between the two groups (Panel **A** vs. **B**). Cells here were fixed at 11:00 am, at 1945 µmoles $m^{-2}$ $s^{-1}$. Pyrenoids are labeled with "P." Scale bar = 2 µm.

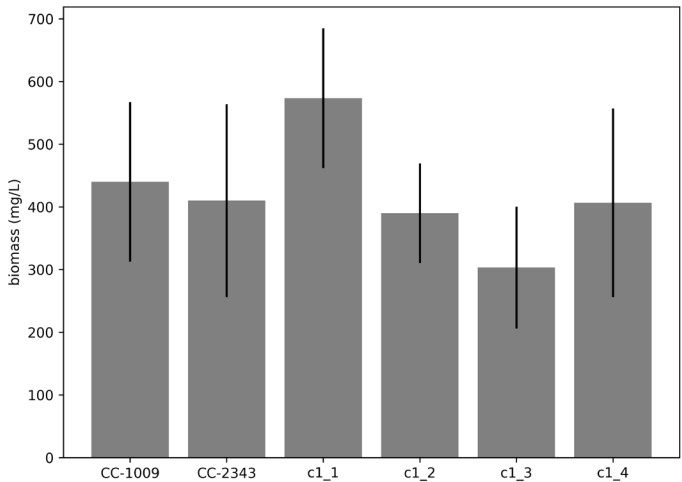

**Appendix 1—figure 12.** Biomass measurement of CC-1009, CC-2343, and tetrad progeny grown in flasks under 85 μmoles m$^{-2}$ s$^{-1}$ PAR and low $CO_2$ for 4 days, after each flask was inoculated with 1 × 10$^5$ cells/ml. Measurements were based on ash free dry weight of three different biological replicates for each strain. Error bars represent the standard deviation of three biological replicates. There was no statistical difference between the parents (CC-1009 and CC-2343) and any progeny. Though c1_1 did have higher productivity that two of the other progeny, all of the lines were able to grow under low $CO_2$ (i.e. none were high $CO_2$ requiring).

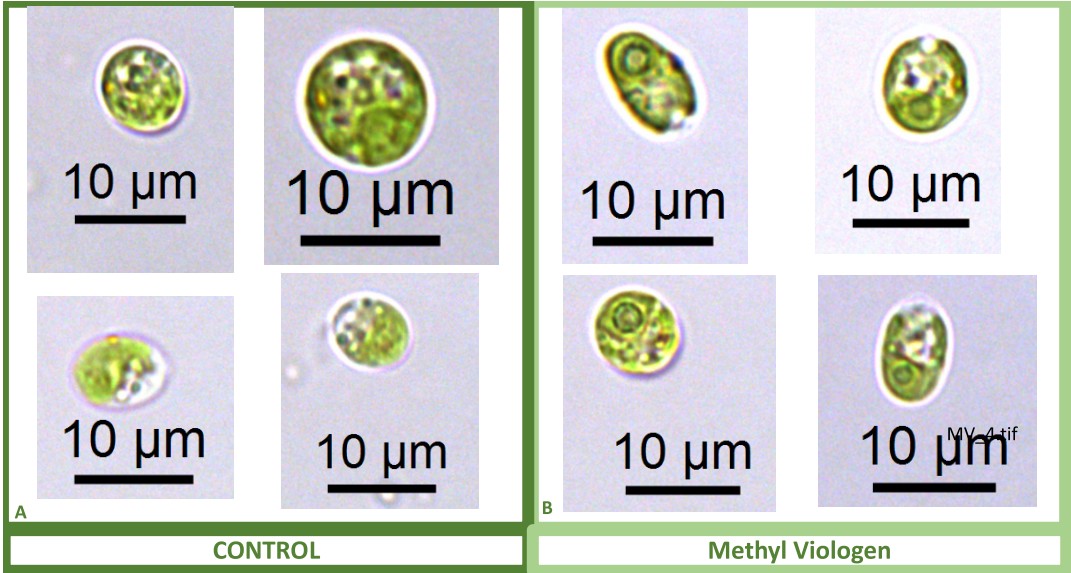

**Appendix 1—figure 13.** Representative light microscopy images of CC-2343 control (Panel A) and treated (Panel B) cells, two hours after our sinusoidal light had turned on, with 0.1 μM of methyl viologen, and then exposed to 6 hours of low light (~50 μmol m$^{-2}$ s$^{-1}$ PAR) with saturating (5 mM) bicarbonate in minimal 2NBH media.

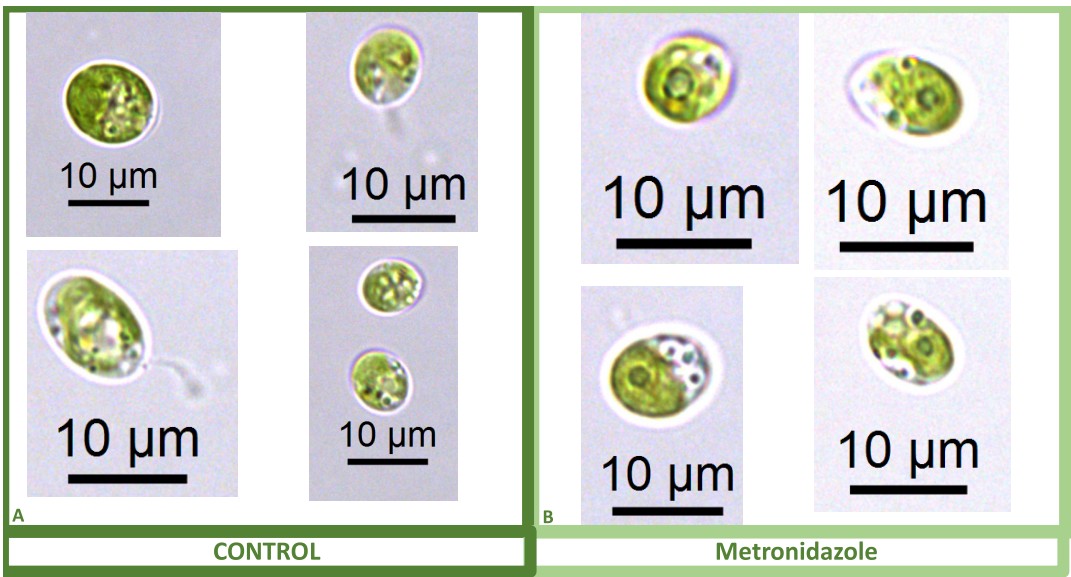

**Appendix 1—figure 14.** Representative light microscopy images of CC-2343 control (Panel A) and treated (Panel B) cells, two hours after our sinusoidal light had turned on, with 4 mM of metronidazole, and then exposed to 6 hours of low light (~50 µmol m$^{-2}$ s$^{-1}$ PAR) with saturating 5 mM bicarbonate in minimal 2NBH media.

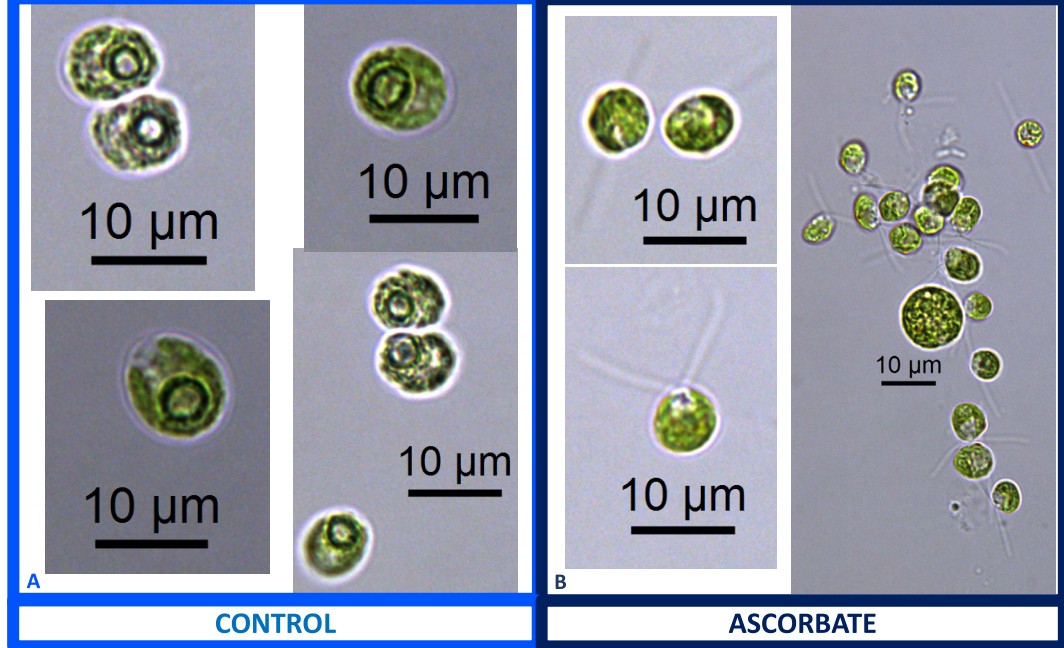

**Appendix 1—figure 15.** Representative light microscopy images of CC-1009 control (Panel A) and 10 mM ascorbate treated (Panel B) cells, grown for 24 hours while being bubbled with air (low $CO_2$) in minimal 2NBH media under ~85 µmol m$^{-2}$ s$^{-1}$ PAR.

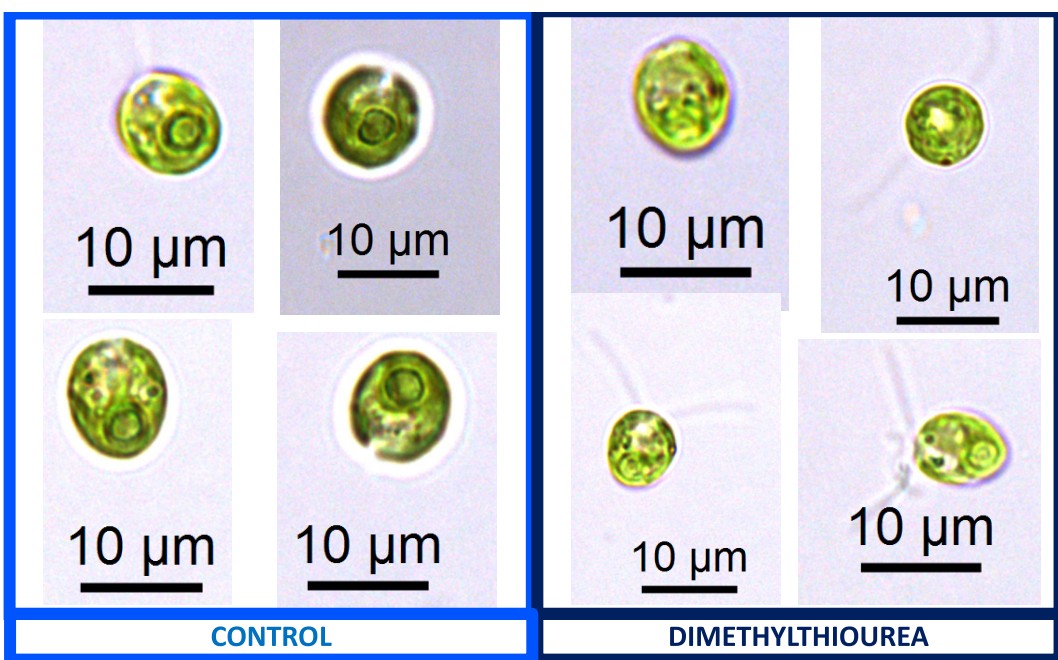

**Appendix 1—figure 16.** Representative light microscopy images of CC-1009 control (Panel A) and 15 mM dimethylthiourea treated (Panel B) cells, grown for 24 hours while being bubbled with air (low $CO_2$) in minimal 2NBH media under ~85 μmol m$^{-2}$ s$^{-1}$ PAR.

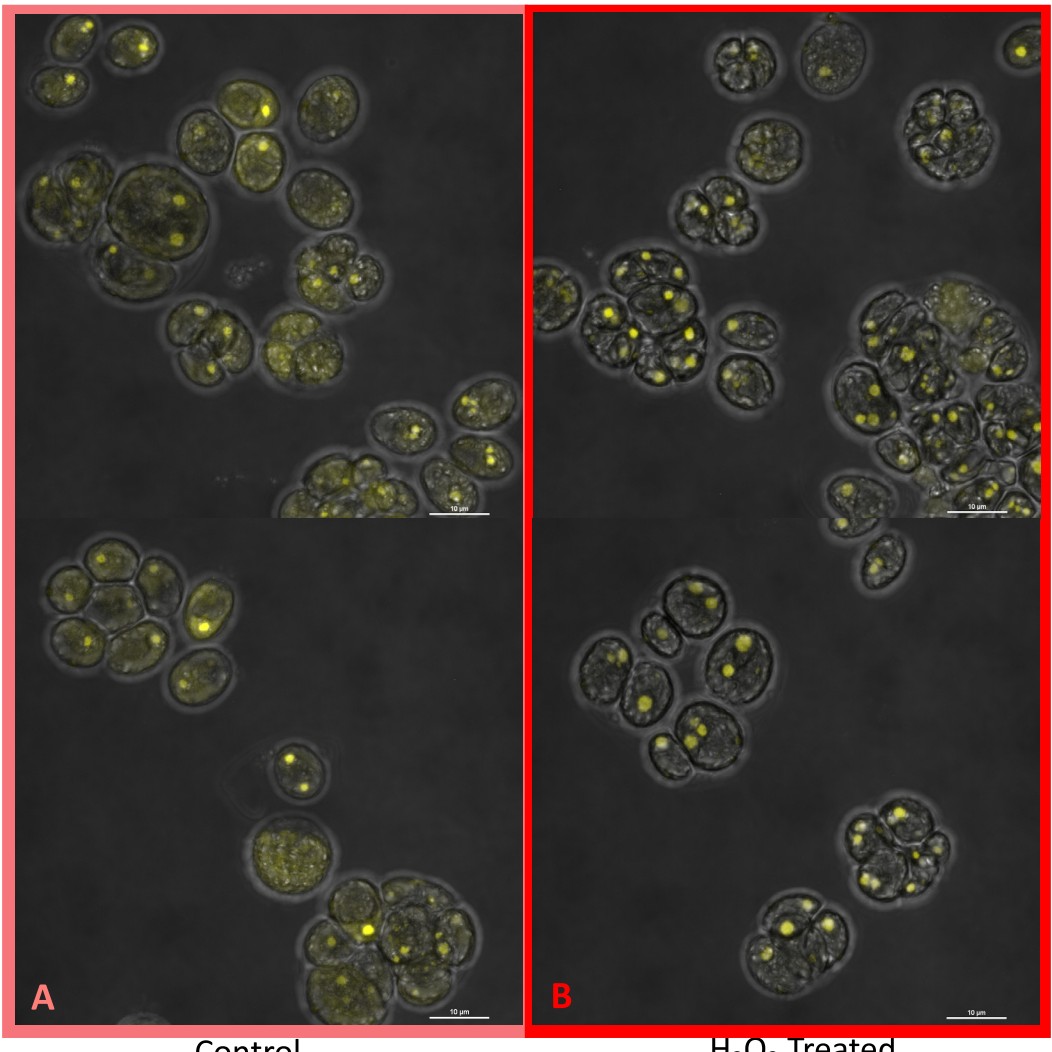

Control H₂O₂ Treated

**Appendix 1—figure 17.** Localization of rubisco, control cells (Panel A) and cells exposed to 100 μM $H_2O_2$ (Panel B), visualized by the Nikon A1 Confocal Laser Scanning Confocal Microscope confocal microscopy, in strain CC-5357, containing at RBCS1-Venus tag. Scale bar = 10 μm. These images, along with those with the Olympus (*Figure 6*), indicate a clear change in rubisco localization, rather the simply a change in rubisco amount. Since confocal microscopy is designed to acquire a very thin optical section through the thickness of a single cell, it can be difficult to use confocal fluorescence microscopy to accurately measure total protein content within the 3-dimensional volume of a single cell. Acquisition of several images through the thickness of the cell may actually over estimate or under estimate the total fluorescence intensity, depending on the Z-step increment. However, when comparing these images of Venus Fluorescent Protein-labeled rubisco within *Chlamydomonas* cells acquired using the Nikon A1 confocal microscope, the fluorescence intensity of the pyrenoid matrix of most cells under control conditions (5 mM bicarbonate, no $H_2O_2$ treatment) was similar to or dimmer than the fluorescence intensity of the pyrenoid matrix within cells treated for six hours with 100 μM $H_2O_2$. In contrast, the fluorescence intensity of the Venus Fluorescent Protein-labeled rubisco located outside of the pyrenoid matrix was measurably higher in cells under control conditions compared to $H_2O_2$-treated cells. If the decrease in Venus Fluorescent Protein-labeled rubisco fluorescence located outside of the pyrenoid matrix in $H_2O_2$-treated cells was due to an overall decrease in rubisco production within the cell and not due to delocalization of the protein from the pyrenoid to the cytoplasm of the cell, then fluorescence intensity of the pyrenoid within $H_2O_2$-treated cells should show a comparable decrease in fluorescence intensity, which is not seen. Fluorescence was detected using 0.5 % 514 nm diode laser intensity. Fluorescence emission was recorded through 565/70 nm band pass filter using a Gallium Arsenide Phosphide (GaASP) detector (High Voltage = 47, offset –7).

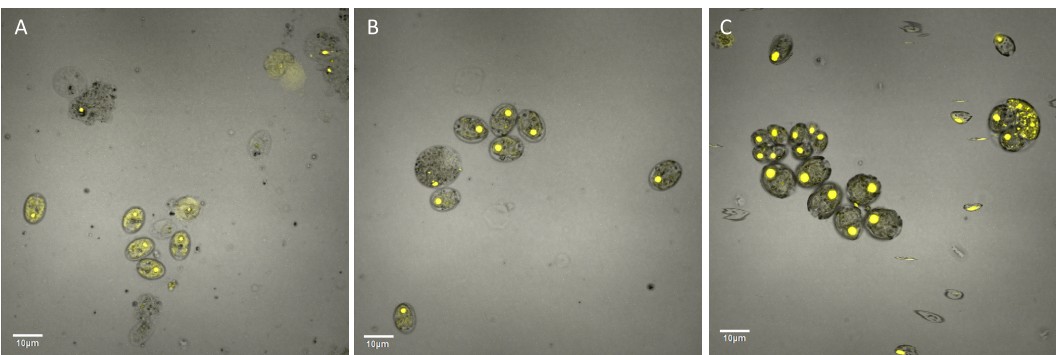

**Appendix 1—figure 18.** Localization of rubisco, control cells (Panel A), cells exposed to 0.1 µM methyl viologen (Panel B), and 8 mM metronidazole (Panel C) visualized by an Olympus FluoView 1,000 Confocal Laser Scanning Microscope, in strain CC-5357, containing a RBCS1-Venus tag. Scale bar = 10 µm.

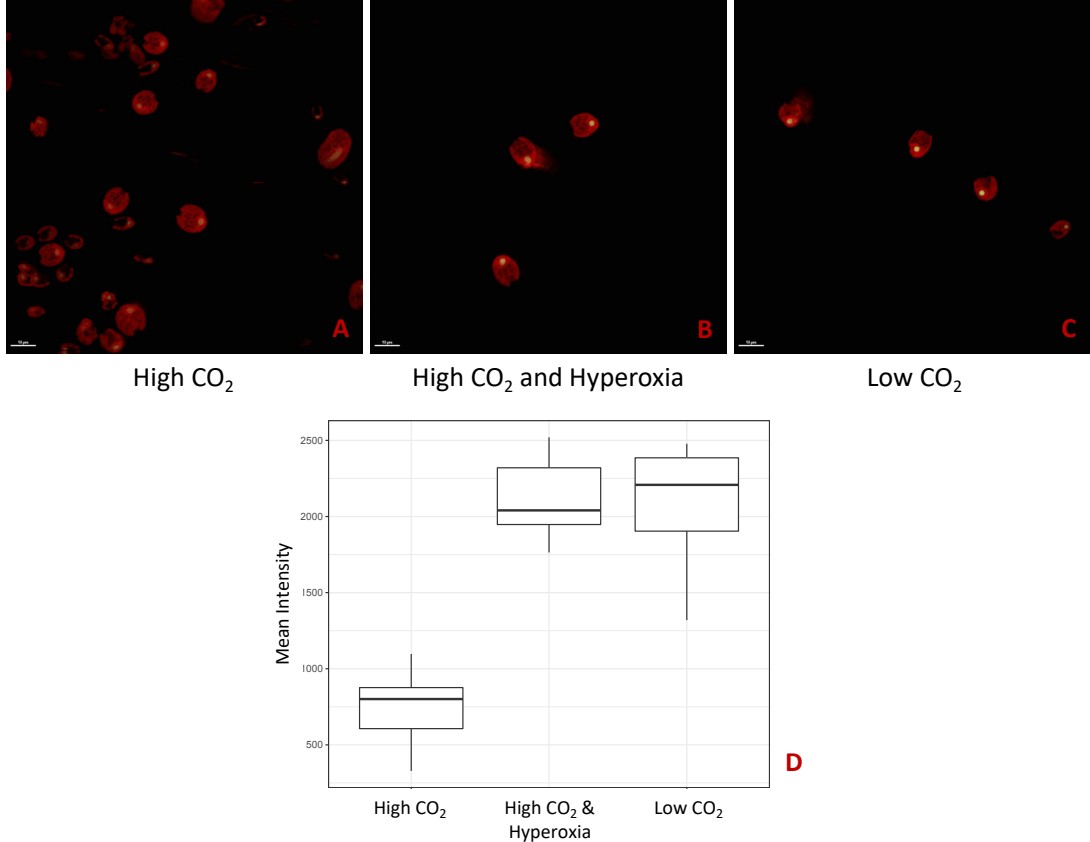

**Appendix 1—figure 19.** Strain CC-5357, containing a RbcS1-Venus, with Chlorophyll shown in red and RbcS1-Venus in yellow. All cells were grown with 14:10 hour (light:dark) sinusoidal illumination with peak light intensity of 2000 µmol m$^{-2}$ s$^{-1}$ PAR, in HS media. Cells were sparged for 60 seconds every 15 minutes with either 5% $CO_2$ (Panel **A**), 5% $CO_2$ and 95% Oxygen (Panel **B**), or ambient air (i.e. low $CO_2$) (Panel **C**). The average fluorescence intensity of the localized Venus Fluorescent Protein-labeled rubisco within the pyrenoids of *Chlamydomonas* cells was measured using the Nikon A1 Confocal Laser Scanning Microscope. Because the tagged rubisco was expressed with a PsaD (Photosystem I reaction center subunit II) promoter, the fluorescent signal may not be a conclusive proxy for rubisco amount and cannot be attributed to transcriptional responses to hyperoxia. However, the results are
*Appendix 1—figure 19 continued on next page*

*Appendix 1—figure 19 continued*

consistent with the fluorescent-tagged rubisco accumulating to a greater extent in the pyrenoids of cells exposed to hyperoxia, similar to what has been observed when cell are exposed to low $CO_2$. Panel D shows the mean intensity of the Venus-tagged rubisco of the three treatments. Panels A, B, and C show representative images used to quantify this mean intensity. Significant differences (P ≤ .0001) were found between the High $CO_2$ and Hyperoxia; High $CO_2$ and Low $CO_2$ treatments. No difference was found between Hyperoxia and Low $CO_2$. Scale bar = 10 µm.

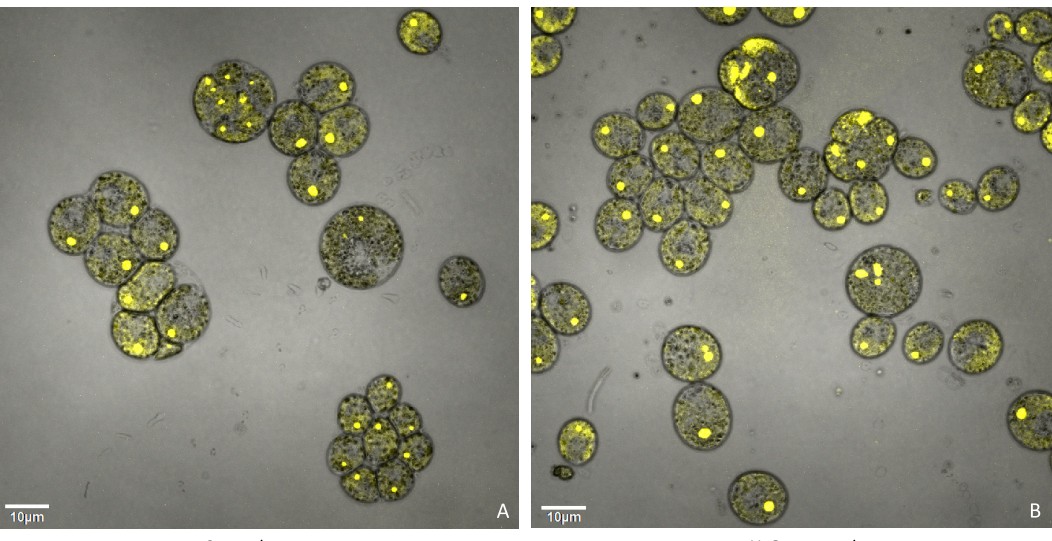

Control                                                  $H_2O_2$ treated

**Appendix 1—figure 20.** Confocal microscopy of liquid TAP-grown CC-5357, which has a Venus labeled RbcS1, treated without (Panel A) and with $H_2O_2$ (Panel B), showing no change in localization of rubisco. Cultures were harvested from photobioreactors in the morning (two hours after the start of illumination) and diluted by half with fresh TAP media – without (control) or with addition of 100 µM of $H_2O_2$. After ten hours in low light (~85 µmol photons $m^{-2}$ $s^{-1}$), cells were then viewed with the confocal microscope. Scale bar = 10 µm.

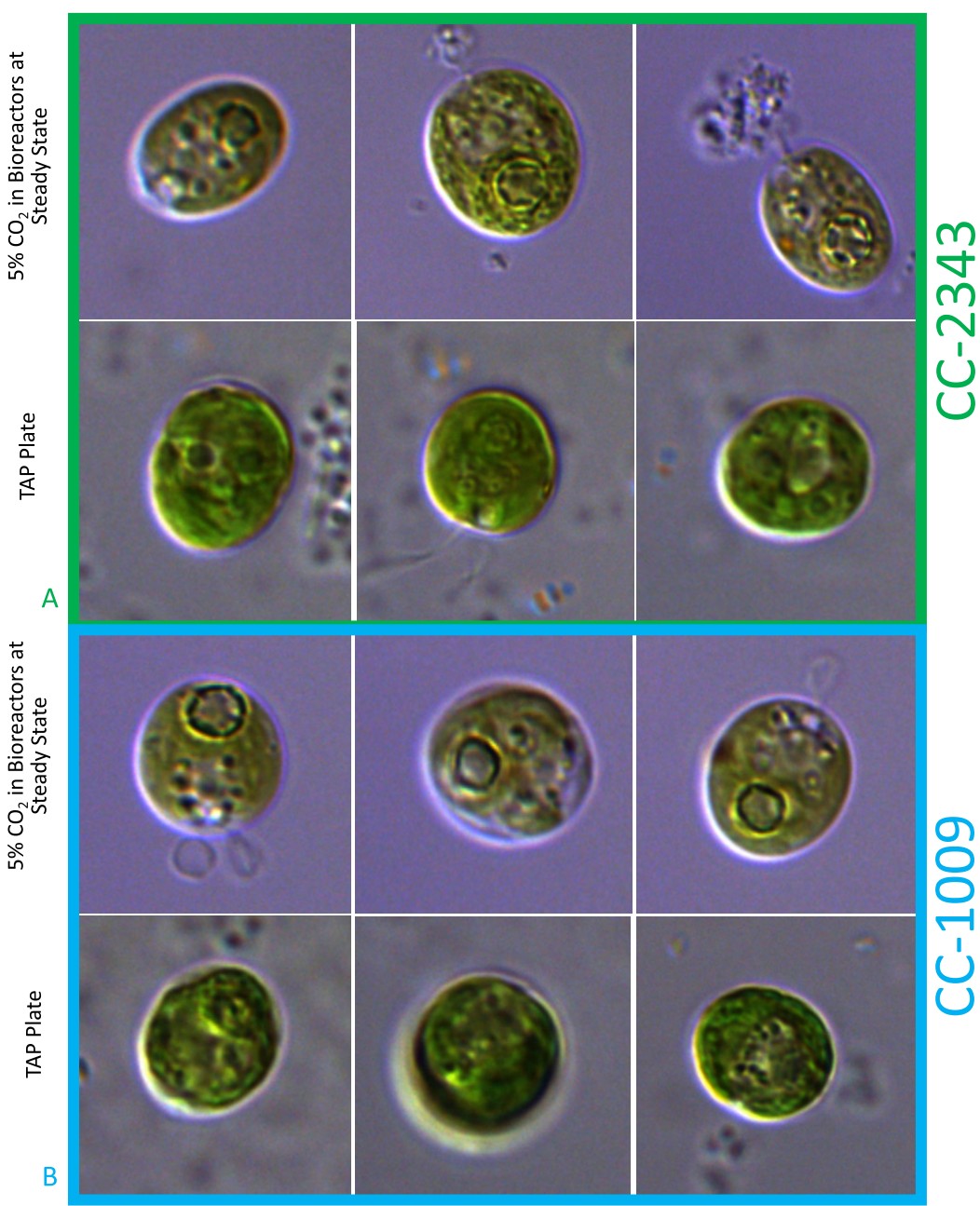

**Appendix 1—figure 21.** Comparison of cells grown under steady state in liquid media (with 5 % $CO_2$) versus cells grown on the surface of a TAP plate. Both CC-2343 (Panel A) and CC-1009 (Panel B) showed losses of the starch sheath when grown on a TAP plate.

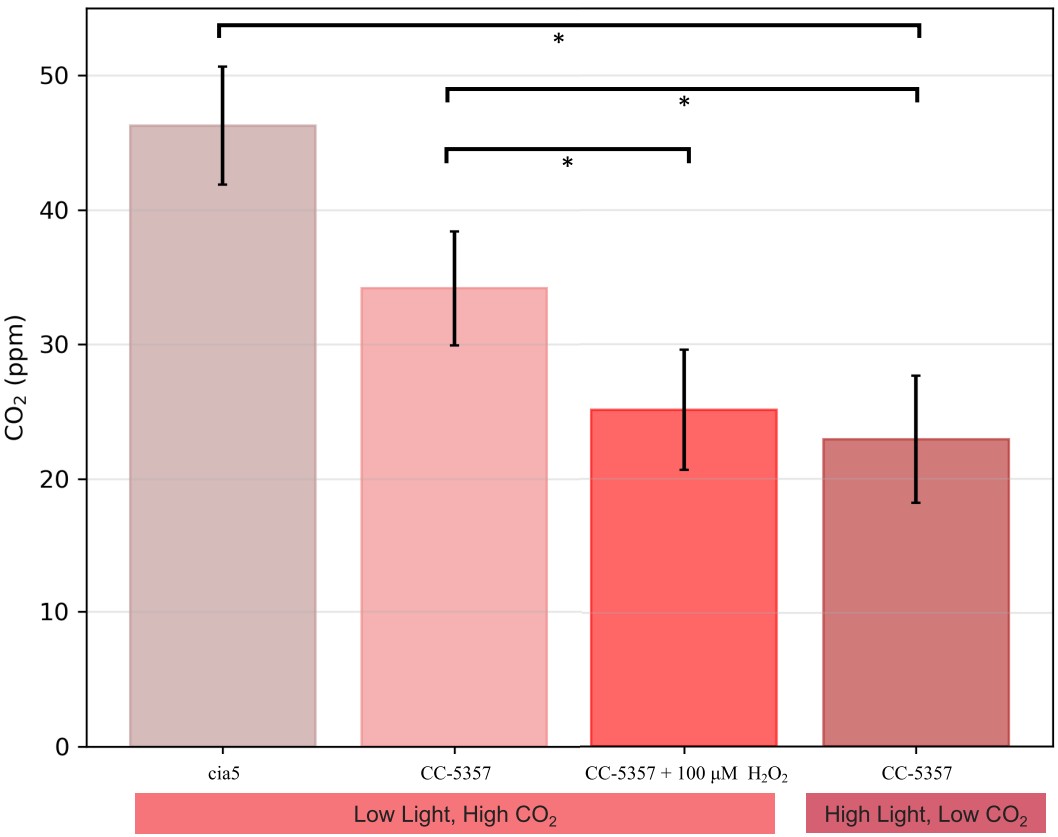

**Appendix 1—figure 22.** Ci compensation points, determined as in Methods, for a mutant defective in the CCM (*cia5*); CC-5357 cells grown under low light (10 µmol photons m$^{-2}$ s$^{-1}$ PAR) and high $CO_2$, where CCM activity is expected to be low; CC-5357 cells grown under high $CO_2$ and low light (10 µmol photons m$^{-2}$ s$^{-1}$ PAR) but pretreated for a minimum of 3 hours with 100 uM $H_2O_2$ to induce pyrenoid formation; and CC-5357 cells grown under high light (100 µmol photons m–2 s–1 PAR) and low $CO_2$, where we expect high CCM activity. Error bars represent standard deviation of at least five biological replicates. *P≤0.01.

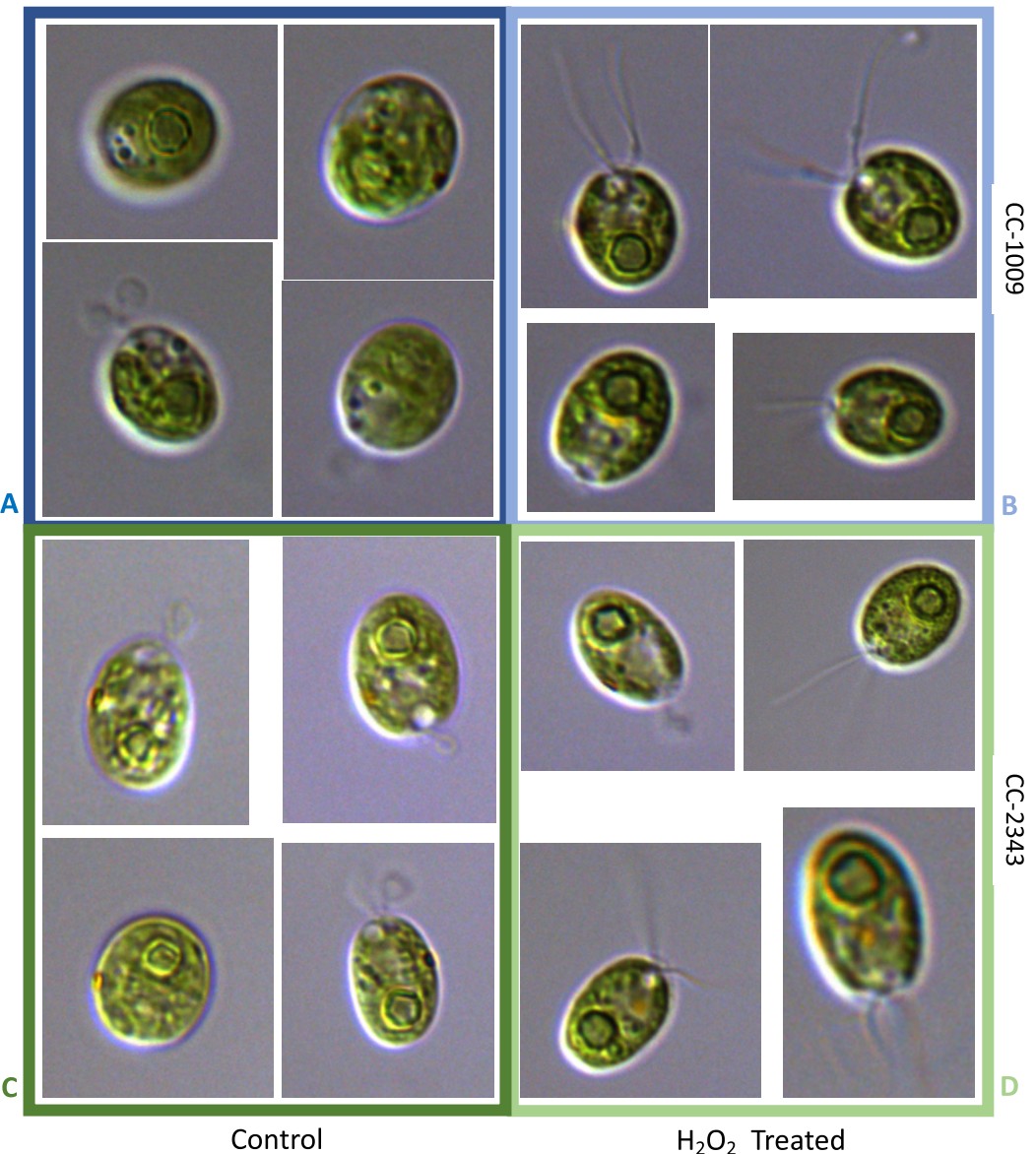

**Appendix 1—figure 23.** Representative light microscopy images of CC-1009 (Panels A & B) and CC-2343 (Panels C & D) control and cells treated, two hours after our sinusoidal light had turned on, with 100 µM of $H_2O_2$, and then exposed to 7 hours of low light (~50 µmoles photons $m^{-2}$ $s^{-1}$) with saturating 5 mM bicarbonate in minimal 2NBH media.

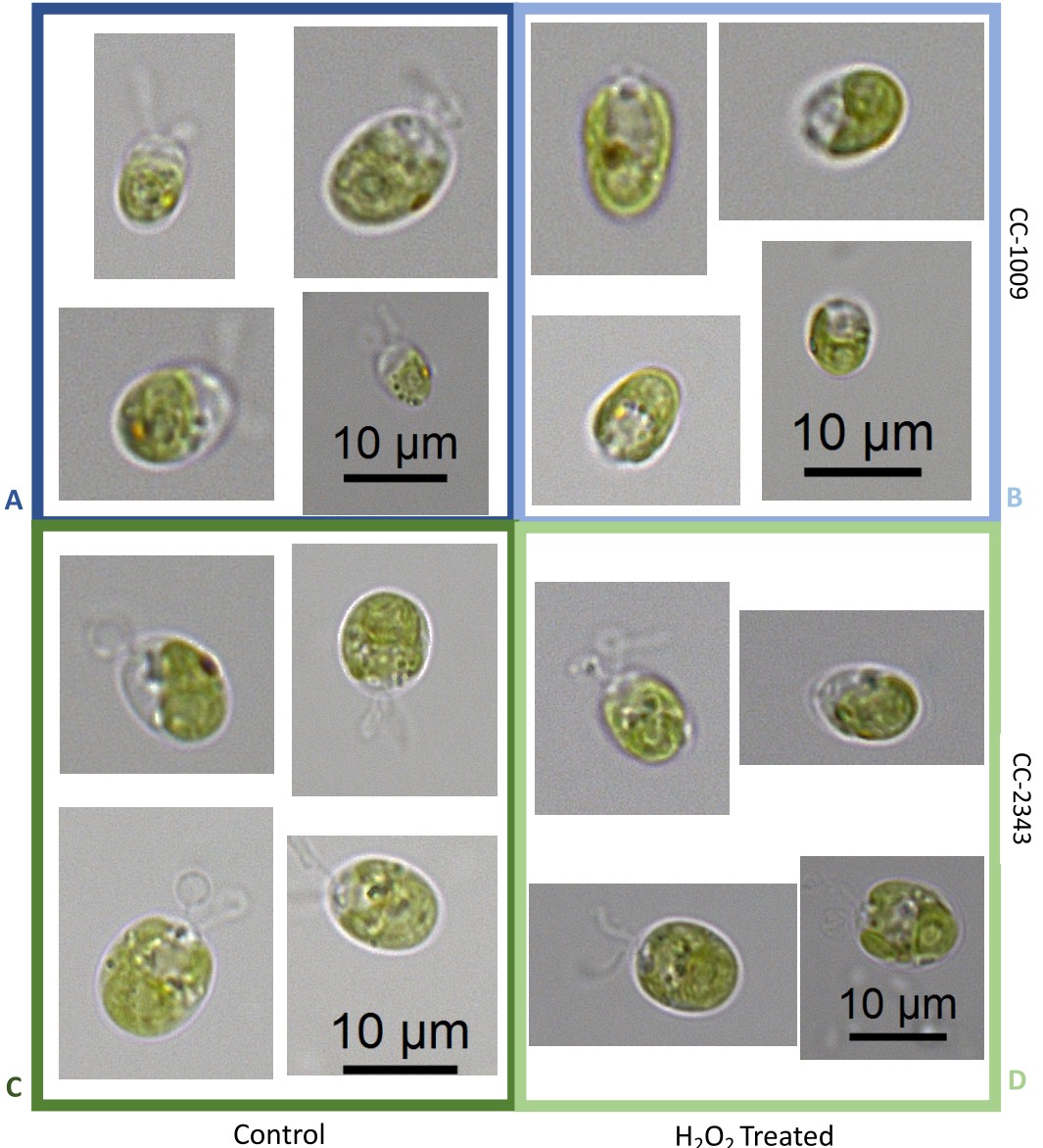

**Appendix 1—figure 24.** Representative light microscopy images of CC-1009 (Panels A & B) and CC-2343 (Panels C & D) control and cells treated, the light remaining off, with 100 μM of H₂O₂, and kept at 7 hours of dark with saturating 5 mM bicarbonate in minimal 2NBH media.

