## [Decision Letter]

**Decision letter after peer review:**

Thank you for submitting your article "The Induction of Pyrenoid Synthesis by Hyperoxia and its Implications for the Natural Diversity of Photosynthetic Responses in *Chlamydomonas*" for consideration by *eLife*. Your article has been reviewed by 3 peer reviewers, and the evaluation has been overseen by a Reviewing Editor and Jürgen Kleine-Vehn as the Senior Editor. The reviewers have opted to remain anonymous.

Essential revisions:

1) All reviewers agree that the presentation of the manuscript should be dramatically revised to improve readability. Generally, figures should be consistently organized, single panel figures should be combined into meaningful multipaneled figures, and the text should be refined. Please see specific suggestions made by each reviewer (particularly Reviewer #4) for detailed guidance.

2) Currently, the pyrenoid formation phenotypes under different conditions are descriptive and should be supported by quantification of multiple cells. Please add quantification of TEM images and appropriate statistical analyses. Additionally, please add further quantitative analysis of the Venus-RuBisCO line under additional conditions (see comments from Reviewer #1).

3) The finding that pyrenoid formation can be induced by hydrogen peroxide is exciting, but it remains unresolved whether CCM is also induced under these conditions, or whether, as Reviewer 1 suggests, the pyrenoid and starch sheath might act to exclude O2. This important question could be addressed by assessing the induction of CCM by measuring affinity for Ci under different conditions and for the different genotypes. If these experiments are not conducted, please revise the discussion to include the distinction between the induction of pyrenoid structure, which has been thoroughly studied in this work, and the induction of CCM, which has not yet been addressed by the experiments presented here.

4) Reviewer #3 made some excellent suggestions about using published datasets to generate hypotheses about the genetic basis of the responses documented in this paper. These analyses are not strictly necessary as part of a revised manuscript but are very constructive and pragmatic.

*Reviewer #1 (Recommendations for the authors):*

Presentation: The homogeneity of the different TEM Figures (and their legends) made it challenging to follow the manuscript. In every Figure it required some effort to find the specific growth conditions employed in the legend, and to independently interpret the implications. The first suggestion would be to ensure the Lead sentence of the Legend summarizes the main interpretation/finding communicated by the Figure. In addition it would help greatly if every Figure (and Supplementary Figure) showing TEM or light microscopy images of cells were supplemented with a standardized visual legend (possibly including some schematic cues) that clearly allowed rapid assessment of the growth conditions used (light, minimal media, hyperoxia, high CO2 etc.. ).

Points for improvement:

Clearly the findings shown leave many questions unaddressed, as is to be expected for a significant novel discovery. There is a key question that I am surprised was not addressed however. The induction of the CO2 concentrating mechanism is generally assessed by measuring the photosynthetic affinity for Ci. High CO2 acclimated cells have a much lower affinity than CCM induced cells. Since the work only addresses the pyrenoid, it would be extremely informative to understand whether the cells acclimated to hyperoxia (or H2O2) and 5% CO2 now also possess a high affinity for Ci. If not, it could support the notion that the pyrenoid and starch sheath here then really acts to exclude O2, rather than forming part of a powered CCM. It would also have profound energetic implications.

The introduction of the strain expressing fluorescently labelled Rubisco is critical to the manuscript. It provides powerful data, as shown in Figure 11b for instance, that pyrenoid formation (or Rubisco localization) is highly homogenous among large numbers of cells. This is in contrast to the majority of pyrenoid data in the manuscript that largely relies on representative images, and does not easily permit generation of quantitative data regarding pyrenoid formation. So it is somewhat surprising that this strain was not utilized a bit more extensively to generate quantitative data regarding the effect of the H2O2, hyperoxia, the speed of the response etc. (rather than the 6h and 31h timepoints that are shown now).

If such data exists, well-presented inclusion would greatly enhance the manuscript.

Detailed points:

1. P.13L16 – here it is stated that the conditions are not hyperoxia, but the Legend to Figure 6 (p.15) states sparging with 95%O2. I assume this is a mistake? Or does this refer to the different types of sparging? This needs to be much clearer for every Figure as pointed out above.

2. P14 Figure 5 legend – specify 5% CO2 in air.

3. P18: Here additional chemicals are utilized that induce internal H2O2. In addition, H2O2 scavengers are shown to prevent pyrenoid formation. The readout is the representative light microscopy images, where pyrenoid starch can be discerned. Were these experiments also performed for the fluorescent Rubisco strain? This would strengthen the data and the conclusions.

4. P22: Figure 11 (and 13)- the scale bars are invisible in these panels

5. P23: Figure 12 – why are there units on the y axis, if what is shown is normalized values?

6. P25: On this page especially, it is very noticeable in many instances that numbers are being reported to meaningless significant figures. E.g. what does 104.28 μm O2 min-1 mean? Does your instrument permit this type of accuracy? Do your replicates suggest that the numbers after the decimal point have any scientific meaning? Reasonable numbers of significant figures should be reported in the text to enhance readability. Numbers in biology by Ron Milo (available online) provides good guidance.

7. P27L17 – Kim and Portis 2004 relates to H2O2 production by Rubisco directly (misfire product of oxygenation). Oxidation of glycolate during photorespiration in the peroxisome is a separate issue. This is currently not clear in the description.

8. P41: Supplementary Material Table 2: If immunoblotting is maintained in the manuscript, then these antibodies need to be correctly described here with catalogue number etc.

9. Regarding Supplementary Figure 4: At this point the immunoblot finding does not really play a role in the manuscript/ negative finding. I understand here an unusual capillary method is utilized, it is very difficult to understand the outcome, and validity of the experiment. I can see the authors have utilized the method previously in Du Plant Cell 2018. At this point I would advise removing this data, or carefully presenting it, permitting it to be critically scrutinized (e.g. is it possible to assess equal loading- Ponceau equivalent etc..). There is great variability shown – is there any meaning?

10. On a related note, there are a number of proteins known to change in abundance when the CCM is induced. Were any of these probed (e.g. LCIB,C)? These could have potentially given more insights?

11. P51: Legend: how many micrographs were analyzed here for the morphometric analysis?

12. P55: Legend: greatly elaborate here what the specific conditions were. Maybe mention that this experiment is to explain/is consistent with the finding of Blaby et al. 2015 (no CCM induction on TAP medium.) (also in the text on page 20, at the moment the rationale of this experiment is not quite clear until we get to the discussion)

*Reviewer #3 (Recommendations for the authors):*

The manuscript presents exciting novel insights into the induction of the pyrenoid in *Chlamydomonas* and goes some way to probe a mechanistic explanation for this. Whilst a good job of probing the cellular response to hyperoxia is done, a genetic basis for these cellular changes is the obvious omittance that needs addressing to strengthen the manuscript. The recommendations below pertaining to this point are not necessarily requirements, but suggestions that the authors could further explore to try and identify the underlying genetic cause of the pyrenoid response to H2O2. Other recommendations refer to specific points of the existing methodologies that could be improved.

Suggestions for highlighting the genetic basis for pyrenoid response to hyperoxia:

Genomic analysis:

– Using data from the cited Jang and Ehrenreich 2012 paper, a genomic comparison between the strains used in the study could be completed to identify genetic candidates for that underpin the differences in strain response.

– If candidate genetic variants are found, simple comparisons between isolates with analogous genetic variations could provide a satisfactory starting point for the genetic explanation for the observed phenotypes.

Transcriptomic analysis:

– Transcriptomic comparison between strains upon H2O2 exposure or hyperoxia could give considerable insights into the underlying mechanism and also allow a time-course analysis of the induction process.

– A more targeted transcriptomic approach, following identification of candidate genes/pathways by other means (or based on your suggested pathways) could also provide satisfactory genetic evidence.

Mutant analysis:

– Phenotypic analysis of mutants in appropriate pathways (H202 production etc.) that show perturbed pyrenoid induction would provide convincing genetic evidence.

– Analysis of the huge dataset from Vilarrasa-Blasi et al., 2020 (BioRxiv), where pooled mutant collections were exposed to a wide range of chemical/growth conditions including H2O2, should allow identification of strong candidate genes in this process that could be further investigated.

Other general points:

Overall, the presentation of data in figures could be greatly improved.

– Several figures could be combined: Figures 1/2 can be combined and panelled, and should include non-treated cells as an introduction to the 'steady-state' morphology of these cells.

– Some figures should be swapped between the main text and supplementary information: Figures S3 and S11 should be incorporated in main text. Figure 4 should be moved to the supplementary information.

– Some figures are unclear: The methodology and labelling of S4 is unclear to the reader. Figure 10C labelling consistency with other figures. Light microscopy in figure S15 and S21 shows contrary evidence to in-text work.

Currently only light microscopy is used to assess pyrenoid phenotypes in a few instances, particularly where compound treatments are concerned. TEM images, and/or morphometric quantification of treatments potentially alongside quantification of pyrenoid localisation by confocal microscopy should be employed to thoroughly address the claims made in the text (pertaining to figures S12/13/14/15). The sporadic reuse of re-cropped images of the same cells (in figure S12/13), imaging of small cells (S14/15) and contradictory images (S15) should be addressed if these figures are kept also.

The authors could provide a better framing for the work in the place of significant recent literature in both the introduction and discussion. The introduction to the molecular details of pyrenoid assembly (P3, lines 1-9) fall short and should either be described in full or the reader referred to recent appropriate reviews of literature (Barrett et al., 2021; Wunder et al., 2020; Meyer, Goudet and Griffiths, 2020). Specific recent literature missing from the introduction and directly applicable to the study include Meyer et al., 2020, Sci. Adv. and Fei et al., 2021, bioRxiv. A better introduction to the current understanding of CCM/pyrenoid induction in *Chlamydomonas* could also be provided (see Fukuzawa et al., 2001, PNAS; Mitchell et al., 2014, Plant Physiology; Fang et al., 2012, Plant Cell). For the discussion, the authors should consider the physiological implications of an ROS-based signal for pyrenoid induction. The authors should also consider integrating recent literature on CCM induction upon limiting CO2 (Blifernez-Klassen et al., 2021, The Plant Cell) in *Chlamydomonas*, and how this hyperoxia response may fit into this network.

To ease comprehension for readers, and related to the above, it would be hugely beneficial for the authors to schematically summarise the findings of the study in the context of what is already known about pyrenoid-induced states in differing nutrient states.

*Reviewer #4 (Recommendations for the authors):*

These are comments in addition to observations and concerns expressed in my public review. The following comments and suggestions are meant to help enhance clarity, and even though I have proposed additional experimental work, I like to re-iterate that most of the experiments would exceed the scope of this work. It may be worthwhile though to use them as discussion points.

The two key findings, hypoxia induced pyrenoid formation and H2O2 signalling are very well demonstrated and represent exciting new insight into pyrenoid function. The manuscript addresses many interesting details but leaves also certain observation a little unexplained and confusing.

Since pyrenoid function is central to this research, the introduction would prepare the reader better if pyrenoid structure and function, as known mostly from studies of the CCM, was described in detail. Likewise, an overview of H2O2 signalling and ROS signalling, particularly in *Chlamydomonas*, would add helpful background information. In this context, a more detailed review of the CCM and pyrenoid structure and function in particular would help in particular to follow the rationale and conclusions of the physiological analyses presented.

I suggest to modify the hypotheses and leave out reference to identifying genetic loci since gene identification is not part of the work presented.

[Editors' note: further revisions were suggested prior to acceptance, as described below.]

Thank you for resubmitting your work entitled "The Induction of Pyrenoid Synthesis by Hyperoxia and its Implications for the Natural Diversity of Photosynthetic Responses in *Chlamydomonas*" for further consideration by *eLife*. Your revised article has been evaluated by Jürgen Kleine-Vehn (Senior Editor) and a Reviewing Editor.

The manuscript has been improved but there are some remaining issues that need to be addressed, as outlined below:

1) Overall, the text has improved, but reviewer 5 found some changes that should be made to the final version to improve clarity and concision of the title and text.

2) The figures are also improved, but will still benefit from more consistent and careful preparation. The third reviewer makes some excellent suggestions in the last paragraph of their review, particularly regarding image cropping and arrangement; the choice of images (inconsistent planes of section in TEM and inconsistent white balancing in light microscopy); lettering size, position, and font; scale bar size and position; the use and appearance of statistics; and the use of different colours across different figures.

*Reviewer #1 (Recommendations for the authors):*

My comments have been satisfactorily addressed in this revision. The manuscript is now much easier to follow. Combining the EM panels into one Figure 2 is especially helpful.

*Reviewer #4 (Recommendations for the authors):*

In my view, the authors have addressed all of the previous reviewer's points very satisfactorily.

The manuscript has been greatly improved with respect to data presentation. Additions to the introduction and discussion make the manuscript very easily accessible to expert and more general audiences.

I compliment the authors on their novel and, after revision, elegantly presented and impressive work.

*Reviewer #5 (Recommendations for the authors):*

The finding that hyperoxia, possibly through the signalling compound H2O2, can elicit a low-CO2 type acclimation in *Chlamydomonas*, even in the presence of large excess of CO2 concentrations is, truly, a remarkable result. It forces the CO2-concentrating mechanism community to reconsider part of the prevailing paradigm (consider O2/CO2 ratio rather than just CO2). It also opens exciting research avenues to identify the elicitors and signalling pathway related to CO2 (or O2/CO2) sensing. I have little doubts that this manuscript can attract a high citation. It is also timely as H2O2 sensors for *Chlamydomonas* have now been developed, which will help further test the H2O2 hypothesis (doi:10.1093/plcell/koab176).

The authors have made an excellent effort at improving the manuscript and data presentation, but I feel there is still room for improving both text and figures. Here are a number of suggestions the authors may want to consider, but also some questions which were not raised by previous reviewers.

Figures

Figure 1.A: Is this the best, unbiased way to normalize the data? It is not clear from Material and Methods and figure caption whether the data normalized the Rubisco concentration. Do CC-1009 and CC-2343 (who differ five-fold in terms of chlorophyll concentration) accumulate similar or different amounts of the CO2-fixing enzyme? The text suggests for Figure 19 in SI that total fluorescence signal is an acceptable proxy for Rubisco concentration. It isn't. See also comment on Figure 6.

Figure 1.B: The initial/final data is redundant and could be in SI. Why not plot the activity on the same graph (as a floating dot or something more elegant), using a second-Y scale?

Figure 4: are four rather than a single representative image really needed for a main figure? You have a neat median section for all four. A, top left; B, top right; C, bottom right; D, top left. These would be sufficient and help focus attention.

Figure 6: the use of the *Chlamydomonas* Culture Collection strain #5357 (RbcS1-Venus mt-) is a judicious choice but comes with limitations. Firstly, the parent strain is genetically different from CC-1009 and CC-2343, and no data is shown on its capacity to handle the high concentrations of H2O2 used in the experiment. Secondly, the tagged variant is co-expressed (under a strong promoter) with the native, untagged Rubisco (and therefore cannot serve as a quantification proxy). It is unclear whether the H2O2 treatment might affect stability of the tagged protein. Thirdly, and maybe more importantly, I would be good to augment the Supplementary Data Set with the Confocal Settings. The raw data on Github suggests that control ('Minus') signal is oversaturated, compared to the H2O2 treated cells. The laser intensity and signal collection settings should be the same for the two treatments. Also, images of the channel showing chlorophyll-autofluorescence should be shown (if available). It is difficult to gage from the images whether the treated cells are still alive. The bloated cells probably have lost their intracellular structures, except the pyrenoid who will resist proteolysis for a while should be excluded from the calculation (cells #1 and #3 for Plus^-1^; cell #1 for Plus-2). This would address any concerns that the near fourfold difference shown in Figure 6.C is not biased by imaging artefacts. Lastly, the bar char (Figure 6.C) would benefit from a more meaningful Y-scale, e.g. fraction of fluorescence signal outside the pyrenoid.

Figure 7 and L711-713: This data makes more sense if presented in the context of optimizing experimental conditions, building up to experiment in Figure 6, rather than on its own. Why not move this to supporting information? I would also err on the side of caution before stating "this is the first study, to our knowledge, to show that the most complete degree of rubisco delocalization occurs when an alga(e) is grown on a TAP agar plate exposed to air, rather than aquatically in liquid TAP". The phenomenon is well known to anyone who has worked with *Chlamydomonas* and looked under a microscope. There are probably numerous earlier publications that used EM, where the pyrenoid is either not visible or massively reduced. I understand the authors' desire to maximize priority on any data, but this data is not relevant to the overall narrative. It distracts from the highly original and remarkable findings of your main work.

Title

"Induction of Pyrenoid Synthesis", is rather broad and does not really capture the essence of your paper. In your rebuttal, you state "Because the main point of the paper is that hyperoxia and hydrogen peroxide induce starch sheath formation, we focused on measurements of the starch sheath." There are numerous instances of "pyrenoid induction" in the text, but the dynamics or Rubisco relocation or starch deposition are not really investigated. Let alone the tubules around which the matrix/starch sheath form. Why not consider a positive statement, for example: "*Chlamydomonas* grown in hyperoxia and elevated CO2 has a pyrenoid: Implications for the natural diversity of photosynthetic acclimation".

Material and methods, Experimental design

Algal growth conditions: 2000 micromols m-2 s^-1^ is 20x more than what Spreitzer and Mets (1981) used to assay light sensitive mutants, namely 2000 and 4000 Lux (cited in L322). Maybe add a note justifying such high intensities, to show that this was not a double stress experiment.

Experimental design (see also comment on Figure 1, above): in the opening sentence of results (L322-327), you explain the rational for choosing CC-1009 and CC-2343. It is, however, unclear to a non-specialist how wildtype like CC-1009 is, relative to (for example) the reference strain used for the genome or the widely used CLiP mutant library. Is CC-1009 particularly good at handling hyperoxia or is just as good as any of the WT used by the *Chlamydomonas* community?

---

## [Author Response]

Essential revisions:1) All reviewers agree that the presentation of the manuscript should be dramatically revised to improve readability. Generally, figures should be consistently organized, single panel figures should be combined into meaningful multipaneled figures, and the text should be refined. Please see specific suggestions made by each reviewer (particularly Reviewer #4) for detailed guidance.

Thank you for this suggestion. We felt that Reviewer 1 also made a very a constructive criticism, pointing out how the “homogeneity of the different TEM figures (and their legends) makes it challenging to follow the manuscript. In response, we have decided to combine the TEM images into one, as was suggested, more easily understandable and meaningful “multipaneled figure.”

2) Currently, the pyrenoid formation phenotypes under different conditions are descriptive and should be supported by quantification of multiple cells. Please add quantification of TEM images and appropriate statistical analyses. Additionally, please add further quantitative analysis of the Venus-RuBisCO line under additional conditions (see comments from Reviewer #1).

We have quantified the differences observed in the TEM images of the cells in our experiments related to H2O2’s induction of the pyrenoid using Image J. Because the main point of the paper is that hyperoxia and hydrogen peroxide induce starch sheath formation, we focused on measurements of the starch sheath.

The quantification indicates that hyperoxia and H2O2 induce increased pyrenoid formation and strengthens our conclusion that the signal for pyrenoid formation is not merely low CO_2_ (Figure 5). This result is also consistent with other data in the paper indicating that H2O2 raises the oxygen compensation point CC-1009 and CC-2343 (Figure 10), and decreases the CO_2_ compensation points (see Appendix 1 – Figure 22).

All of the strains showed some induction of the pyrenoid under hyperoxia, but there were significant differences between the strains in the degree to which they were “porous” or “sealed.” To quantify this attribute, we measured a parameter we term “percent exposure” that measures the percentage of the projected pyrenoid surfaces that were not enclosed by (in contact with) visible starch sheath. The quantification (See Appendix 1 Figure 6) confirms the qualitatively visible phenotype that the percentage of exposure was much higher in the sensitive lines, CC-2343 and progenies c1_3 and C1_4. We also provided at least 20 additional images for each strain, in addition to the ones provided in previous version and transparent reporting (See Voluntary Reporting Image Set 12).

Also, we have also now used the fluorescent labeled rubisco strain to compare the amount of rubisco in the pyrenoid when cells are grown under high CO_2_, hyperoxia, and low CO_2_. We found clear evidence that, even when sparged with saturating CO_2_, hyperoxia increased the rubisco content of the pyrenoids (Appendix 1-Figure 19), which has previously been associated with the formation of functional CCM in the pyrenoid (Freeman Rosenzweig et al., 2017; Mitchell et al., 2014).

3) The finding that pyrenoid formation can be induced by hydrogen peroxide is exciting, but it remains unresolved whether CCM is also induced under these conditions, or whether, as Reviewer 1 suggests, the pyrenoid and starch sheath might act to exclude O2. This important question could be addressed by assessing the induction of CCM by measuring affinity for Ci under different conditions and for the different genotypes. If these experiments are not conducted, please revise the discussion to include the distinction between the induction of pyrenoid structure, which has been thoroughly studied in this work, and the induction of CCM, which has not yet been addressed by the experiments presented here.

Thank you for this excellent suggestion. We note that, because of the interconversion of CO_2_ and HCO_3_^-^, the Ci compensation point could include contributions from both forms. Thus, the experiments were not trivial to perform, but we felt that the point was interesting and thus, we developed a method to estimate the degree to which the Ci in the medium is depleted by the combination of photosynthesis, photorespiration and respiration. We found that H2O2 induced a decrease in Ci compensation to levels similar to those achieved by low CO_2_. As controls we used both cells grown at low light and high CO_2_ and a mutant (cia5) that lacks functional CCM, both of which showed higher Ci compensation points. These results are included in the revised manuscript as Appendix 1-Figure 22.

4) Reviewer #3 made some excellent suggestions about using published datasets to generate hypotheses about the genetic basis of the responses documented in this paper. These analyses are not strictly necessary as part of a revised manuscript but are very constructive and pragmatic.

We appreciate the suggestion. We are pursuing this type of approach in a separate manuscript for two reasons: (1) We found that conditions of the published works were sufficiently different to make direct comparisons difficult. For example, some were taken in TAP-grown cells where CCM is probably not induced. (2) We have produced and obtained sequence data for the parents and the progeny as well as RNAseq results, but these go far beyond the scope of the current paper.

Reviewer #1 (Recommendations for the authors):Presentation: The homogeneity of the different TEM Figures (and their legends) made it challenging to follow the manuscript. In every Figure it required some effort to find the specific growth conditions employed in the legend, and to independently interpret the implications. The first suggestion would be to ensure the Lead sentence of the Legend summarizes the main interpretation/finding communicated by the Figure. In addition, it would help greatly if every Figure (and Supplementary Figure) showing TEM or light microscopy images of cells were supplemented with a standardized visual legend (possibly including some schematic cues) that clearly allowed rapid assessment of the growth conditions used (light, minimal media, hyperoxia, high CO2 etc..).

We thank the reviewer to the comment. We have condensed the TEM figures accordingly into one figure (Figure 2) in the main text to clarify the presentation.

Points for improvement:Clearly the findings shown leave many questions unaddressed, as is to be expected for a significant novel discovery. There is a key question that I am surprised was not addressed however. The induction of the CO2 concentrating mechanism is generally assessed by measuring the photosynthetic affinity for Ci. High CO2 acclimated cells have a much lower affinity than CCM induced cells. Since the work only addresses the pyrenoid, it would be extremely informative to understand whether the cells acclimated to hyperoxia (or H2O2) and 5% CO2 now also possess a high affinity for Ci. If not, it could support the notion that the pyrenoid and starch sheath here then really acts to exclude O2, rather than forming part of a powered CCM. It would also have profound energetic implications.

We thank the reviewer for this suggestion. We have found that cells treated with H2O2 indeed, even in the presence of high bicarbonate (5mM) possess a high affinity for Ci (see Appendix 1 – Figure 22). Please see our reply to Editor’s point #3.

The introduction of the strain expressing fluorescently labelled Rubisco is critical to the manuscript. It provides powerful data, as shown in Figure 11b for instance, that pyrenoid formation (or Rubisco localization) is highly homogenous among large numbers of cells. This is in contrast to the majority of pyrenoid data in the manuscript that largely relies on representative images, and does not easily permit generation of quantitative data regarding pyrenoid formation. So it is somewhat surprising that this strain was not utilized a bit more extensively to generate quantitative data regarding the effect of the H2O2, hyperoxia, the speed of the response etc. (rather than the 6h and 31h timepoints that are shown now).If such data exists, well-presented inclusion would greatly enhance the manuscript.

We agree and have added new data about how hyperoxia effects the rubisco content of the pyrenoid (Appendix 1 – Figure 19). While using the fluorescently labeled strain to measure the speed of the response via confocal microscopy is interesting, it would not directly impact our conclusions that high oxygen (even in the presence of high CO_2_) will lead to an increase in size and rubisco content of the pyrenoid, much the same as low CO_2_.

Detailed points:1. P.13L16 – here it is stated that the conditions are not hyperoxia, but the Legend to Figure 6 (p.15) states sparging with 95%O2. I assume this is a mistake? Or does this refer to the different types of sparging? This needs to be much clearer for every Figure as pointed out above.

We thank the reviewer for catching this error. We have now corrected and condensed the figure so that this should no longer be an issue.

2. P14 Figure 5 legend – specify 5% CO2 in air.

We have made this change. Figure Caption now reads:

Appendix 1-Figure 9: Representative TEM images of *Chlamydomonas* strains, the parents (CC-1009 and CC-2343), as well as their progeny c1_1, c1_2, c1_3, c1_4, having been sparged with CO_2_ for 30 seconds every hour during the night. Cells had been grown in steady state conditions, with 5% CO_2_ in air with 14:10 hour (light:dark) sinusoidal illumination with peak light intensity of 2000 μmoles m^-2^ s^-1^, in minimal 2NBH media. Cells here were fixed two hours after dawn, at 7:00 am, when light levels were 825 μmoles m^-2^ s^-1^. Pyrenoids appeared absent or mostly unsheathed in these conditions. Scale bar = 2 μm.

Please note this figure is being moved to Appendix 1.

3. P18: Here additional chemicals are utilized that induce internal H2O2. In addition, H2O2 scavengers are shown to prevent pyrenoid formation. The readout is the representative light microscopy images, where pyrenoid starch can be discerned. Were these experiments also performed for the fluorescent Rubisco strain? This would strengthen the data and the conclusions.

We thank the reviewer for the interesting suggestion and we had performed some experiments using methyl viologen and metronidazole using the confocal microscope. The initial results looked promising, and consistent with the induction of pyrenoid by H2O2 produced in the presence of these chemicals. However, we found that exposure to the confocal laser induced severe damage to cells treated with these chemicals, most likely because they produce large amounts of ROS production during illumination with the confocal laser. We thus concluded that such results would be difficult to interpret. However, we also now included in the paper an examination of how oxygen, even in the presence of high CO_2_, increased rubisco content of the pyrenoid. We found that, indeed, it did, and feel that this demonstrates that a signal apart from low CO_2_ can influence pyrenoid formation (Appendix 1 – Figure 19).

4. P22: Figure 11 (and 13)- the scale bars are invisible in these panels

We have enlarged the scale bars for these images. We have also changed the color of the image so that it yellow, to reflect that we are showing YFP.

5. P23: Figure 12 – why are there units on the y axis, if what is shown is normalized values?

We agree, and have deleted the units. Thank you for catching this.

6. P25: On this page especially it is very noticeable in many instances that numbers are being reported to meaningless significant figures. E.g. what does 104.28 μm O2 min-1 mean? Does your instrument permit this type of accuracy? Do your replicates suggest that the numbers after the decimal point have any scientific meaning? Reasonable numbers of significant figures should be reported in the text to enhance readability. Numbers in biology by Ron Milo (available online) provides good guidance.

We agree and have truncated the reported numbers to reasonable significant digits.

7. P27L17 – Kim and Portis 2004 relates to H2O2 production by Rubisco directly (misfire product of oxygenation). Oxidation of glycolate during photorespiration in the peroxisome is a separate issue. This is currently not clear in the description.

This is an excellent point. We have corrected the sentence to read

“…signal molecule (Foyer et al., 2009) and its production is increased under high light (Roach et al., 2015), high O_2_ as a misfired product of oxygenation (Kim and Portis, 2004) or low CO_2_ (Foyer et al., 2009). H2O2 is a product of the light reactions, through an alternative electron acceptor pathway such as the Mehler peroxidase reaction (MPR) or the water-water cycle, which is expected to be more active under conditions when light input exceeds the capacity of assimilation (Badger et al., 2000; Mehler, 1951; Strizh, 2008). H2O2 can also be produced as a by-product of photorespiration (Janssen et al., 2014).”

We also made a correction in the introduction, where we cite the Kim and Portis (2004) as follows:

“Oxygenation of rubisco can also result in the formation of rubisco inhibitors (Kim and Portis, 2004)” Instead of our previous statement that “This intermediate of photorespiration can also act as inhibitors of Rubisco” (Kim and Portis 2004).

8. P41: Supplementary Material Table 2: If immunoblotting is maintained in the manuscript, then these antibodies need to be correctly described here with catalogue number etc.

Based on the reviewer’s suggestion (below), the figure is not needed and was thus removed.

9. Regarding Supplementary Figure 4: At this point the immunoblot finding does not really play a role in the manuscript/ negative finding. I understand here an unusual capillary method is utilized, it is very difficult to understand the outcome, and validity of the experiment. I can see the authors have utilized the method previously in Du Plant Cell 2018. At this point I would advise removing this data, or carefully presenting it, permitting it to be critically scrutinized (e.g. is it possible to assess equal loading- Ponceau equivalent etc..). There is great variability shown – is there any meaning?

The meaning of the figure was just to show that, from what we could discern, no protein segregated in a 2:2 manner like the pyrenoid morphology. However, based on the reviewer’s suggestion, we have decided to remove the figure from the manuscript.

10. On a related note, there are a number of proteins known to change in abundance when the CCM is induced. Were any of these probed (e.g. LCIB,C)? These could have potentially given more insights?

We did not, but we are looking into probing these in a future study.

11. P51: Legend: how many micrographs were analyzed here for the morphometric analysis?

Details are presented in the legend. Approximately 30 cells were analyzed.

12. P55: Legend: greatly elaborate here what the specific conditions were. Maybe mention that this experiment is to explain/is consistent with the finding of Blaby et al. 2015 (no CCM induction on TAP medium..) (also in the text on page 20, at the moment the rationale of this experiment is not quite clear until we get to the discussion)

We have elaborated on methods and details in the Figure caption. The text now reads:

“Appendix 1 – Figure 20: Confocal microscopy of liquid TAP grown CC-5357, which has a Venus labeled RBCS1, treated without (Panel A) and with H2O2 (Panel B), showing no change in localization of rubisco. Cultures were harvested from photobioreactors in the morning (two hours after the start of illumination) and diluted by half with fresh TAP media – without (control) or with addition of 100 μM of H2O2. After ten hours in low light (~85 μmol photons m^-2^ s^-1^), cells were then viewed with the confocal.”

We have also added a line in the results, to better describe the system:

“By contrast, no significant changes in rubisco localization were observed when upon addition of 100 μM H2O2 to TAP-grown cells (Appendix 1-Figure 20), the media used in another study testing the effects of hydrogen peroxide on *Chlamydomonas* (Blaby et al., 2015), implying that the effect was dependent on the photosynthetic state of the cells and/or suppressed in the presence of this organic carbon source.”

Reviewer #3 (Recommendations for the authors):The manuscript presents exciting novel insights into the induction of the pyrenoid in Chlamydomonas and goes some way to probe a mechanistic explanation for this. Whilst a good job of probing the cellular response to hyperoxia is done, a genetic basis for these cellular changes is the obvious omittance that needs addressing to strengthen the manuscript. The recommendations below pertaining to this point are not necessarily requirements, but suggestions that the authors could further explore to try and identify the underlying genetic cause of the pyrenoid response to H2O2. Other recommendations refer to specific points of the existing methodologies that could be improved.Suggestions for highlighting the genetic basis for pyrenoid response to hyperoxia:Genomic analysis:– Using data from the cited Jang and Ehrenreich 2012 paper, a genomic comparison between the strains used in the study could be completed to identify genetic candidates for that underpin the differences in strain response.– If candidate genetic variants are found, simple comparisons between isolates with analogous genetic variations could provide a satisfactory starting point for the genetic explanation for the observed phenotypes.

As we discussed above, we now have genomic sequences for the parent and progeny lines, but give the large numbers of polymorphisms, it will be difficult to identify the causative loci at this stage. Thus, while we completely agree that this is a worthwhile pursuit, it is beyond the scope of the current paper.

Transcriptomic analysis:– Transcriptomic comparison between strains upon H2O2 exposure or hyperoxia could give considerable insights into the underlying mechanism and also allow a time-course analysis of the induction process.

This is an interesting suggestion and are pursuing this type of approach in a separate work, as discussed in the response to the Editors’ comments.

– A more targeted transcriptomic approach, following identification of candidate genes/pathways by other means (or based on your suggested pathways) could also provide satisfactory genetic evidence.

We agree that these are worthwhile research goals, but feel they are beyond the scope of this paper. We are working on it for a future study.

Mutant analysis:– Phenotypic analysis of mutants in appropriate pathways (H202 production etc.) that show perturbed pyrenoid induction would provide convincing genetic evidence.

While we agree that such work would be interesting, we feel it is best reserved for a future study. The mutants that we know of that have altered hydrogen peroxide/ROS can be very challenging to grow in media outside of TAP and the mutations likely have pleiotropic effects, and it is not clear to us how we would unambiguously attribute particular effect to H2O2 production in these mutants.

– Analysis of the huge dataset from Vilarrasa-Blasi et al., 2020 (BioRxiv), where pooled mutant collections were exposed to a wide range of chemical/growth conditions including H2O2, should allow identification of strong candidate genes in this process that could be further investigated.

Thank you for the suggestion, and we agree that this will be of interest for future directions.

Other general points:Overall, the presentation of data in figures could be greatly improved.– Several figures could be combined: Figures 1/2 can be combined and panelled, and should include non-treated cells as an introduction to the 'steady-state' morphology of these cells.

We agree and have combined Figures 1 and 2 also with Figures 6 and 7.

– Some figures should be swapped between the main text and supplementary information: Figures S3 and S11 should be incorporated in main text. Figure 4 should be moved to the supplementary information.

We have added Figure S3 to the main text now. Figure S11 has been replaced with our Image J analysis bar graph. Figure 4 has been moved to the supplementary information (now called Appendix 1).

– Some figures are unclear: The methodology and labelling of S4 is unclear to the reader.

We have removed this figure from the paper, in response to reviewer #1.

Figure 10C labelling consistency with other figures.

We have revised the labeling of the axis in the bar graph. We hope this makes the figure clearer.

Light microscopy in figure S15 and S21 shows contrary evidence to in-text work.

The point that the experiment was done in the dark, in order to demonstrate a light dependence of pyrenoid formation even with hydrogen peroxide. We have now clarified this in the text to avoid confusion.

“We found that in autotrophically-grown cells, exogenous addition of H2O2 in the presence of light strongly induced within approximately 6 hours the formation pyrenoid starch sheaths (Figure 4; Figure 5; Appendix 1- Figures 23), and caused rubisco to localize into the pyrenoid (Figure 6). The H2O2 did not induce the pyrenoid when the cells were kept in the dark (Appendix 1-Figure 24).”

Currently only light microscopy is used to assess pyrenoid phenotypes in a few instances, particularly where compound treatments are concerned. TEM images, and/or morphometric quantification of treatments potentially alongside quantification of pyrenoid localisation by confocal microscopy should be employed to thoroughly address the claims made in the text (pertaining to figures S12/13/14/15).

We have now included quantification of the TEM images related to the effects of hydrogen peroxide on the pyrenoid (See Figure 5) and the differences in pyrenoid starch sheaths of the parents and progeny after being exposed to hyperoxia (Appendix 1-Figure 6). With regard to rubisco localization, see response to reviewer 1 with regard to the use of the inhibitors:

We thank the reviewer for the interesting suggestion and we had performed some experiments using methyl viologen and metronidazole using the confocal microscope. The initial results looked promising, and consistent with the induction of pyrenoid by H2O2 produced in the presence of these chemicals. However, we found that exposure to the confocal laser induced severe damage to cells treated with these chemicals, most likely because they produce large amounts of ROS production during illumination with the confocal laser. We thus concluded that such results would be difficult to interpret. However, we also now included in the paper an examination of how oxygen, even in the presence of high CO_2_, increased rubisco content of the pyrenoid. We found that, indeed, it did, and feel that this demonstrates that a signal apart from low CO_2_ can influence pyrenoid formation (Appendix 1-Figure 19).

We did quantify the differences in rubisco amount in response to oxygen in Appendix 1-Figure 19 (see Panel D).

The sporadic reuse of re-cropped images of the same cells (in figure S12/13), imaging of small cells (S14/15) and contradictory images (S15) should be addressed if these figures are kept also.

We have fixed S12/13 (now Appendix 1-Figure 13; Appendix 1-Figure 14), and with regard to the use of small cells (i.e. Appendix 1-Figure 15), the image we assume is referred to is included because it shows many cells, none of which show a pyrenoid. The microscope image includes a scale bar, and the cells are not particularly small (as indicated when comparing the scale bar to the other images). What is just clear from this image of many cells is that none have apparent starch sheaths. Like all of our images, the raw file of this image is available (See Ascorbate2.tif in images for Appendix 1-Figure 15).

We are unsure what is contradictory about the image in S15 (what is now Appendix 1_Figure 16). The control cells do have apparent starch sheaths, while those treated with dimethylthiourea do not.

The authors could provide a better framing for the work in the place of significant recent literature in both the introduction and discussion. The introduction to the molecular details of pyrenoid assembly (P3, lines 1-9) fall short and should either be described in full or the reader referred to recent appropriate reviews of literature (Barrett et al., 2021; Wunder et al., 2020; Meyer, Goudet and Griffiths, 2020). Specific recent literature missing from the introduction and directly applicable to the study include Meyer et al., 2020, Sci. Adv. and Fei et al., 2021, bioRxiv. A better introduction to the current understanding of CCM/pyrenoid induction in Chlamydomonas could also be provided (see Fukuzawa et al., 2001, PNAS; Mitchell et al., 2014, Plant Physiology; Fang et al., 2012, Plant Cell). For the discussion, the authors should consider the physiological implications of an ROS-based signal for pyrenoid induction. The authors should also consider integrating recent literature on CCM induction upon limiting CO2 (Blifernez-Klassen et al., 2021, The Plant Cell) in Chlamydomonas, and how this hyperoxia response may fit into this network.

Thank you for recommending these stimulating articles:

The introduction now reads:

“The expression and function of green algal CCMs in eukaryotic algae is highly regulated; cells grown on or below air levels of CO_2_ (0.04%) develop active CCMs (Aizawa and Miyachi, 1986; Badger et al., 1980), whereas those grown with high CO_2_ levels lack them, and thus show low apparent affinities for CO_2_. [….] Although, it has recently been found that pyrenoid proteins share a sequence motif that is necessary to target proteins to the pyrenoid and binds to rubisco (Meyer et al., 2020b).”

And in the discussion, we now write:

“In this regard, it is intriguing that ROS labeling under hyperoxia was strongly localized in CC-1009 but not CC-2342 (Figure 13), hinting that H2O2 produced in a specific subcellular location and process may play a role in the differential development of the pyrenoid in the two parent lines, as discussed below. Taken together, these data sets are consistent with control of pyrenoid morphology at multiple levels, perhaps similar to the processes that regulate the expression of LHCSR3, involved in photoprotective nonphotochemical quenching, which is regulated by light quality and CO_2_ availability (Maruyama et al., 2014; Semchonok et al., 2017). Future studies can also investigate how hyperoxia plays a role in the gene regulatory network for antennae size, which has been shown to be affected by low CO_2_ (Blifernez-Klassen et al., 2021).”

To ease comprehension for readers, and related to the above, it would be hugely beneficial for the authors to schematically summarise the findings of the study in the context of what is already known about pyrenoid-induced states in differing nutrient states.

We now include a summary with bulleted points describing what is already known about pyrenoid-induced states under different conditions, with reference how our work fits in with it. This can be found in Appendix 2.

Reviewer #4 (Recommendations for the authors):These are comments in addition to observations and concerns expressed in my public review. The following comments and suggestions are meant to help enhance clarity, and even though I have proposed additional experimental work, I like to re-iterate that most of the experiments would exceed the scope of this work. It may be worthwhile though to use them as discussion points.

We thank the reviewer for the suggestion, and for the comment about exceeding the scope of this work. We are pursuing many of them for future studies.

The two key findings, hypoxia induced pyrenoid formation and H2O2 signalling are very well demonstrated and represent exciting new insight into pyrenoid function. The manuscript addresses many interesting details but leaves also certain observation a little unexplained and confusing.Since pyrenoid function is central to this research, the introduction would prepare the reader better if pyrenoid structure and function, as known mostly from studies of the CCM, was described in detail. Likewise, an overview of H2O2 signalling and ROS signalling, particularly in Chlamydomonas, would add helpful background information. In this context, a more detailed review of the CCM and pyrenoid structure and function in particular would help in particular to follow the rationale and conclusions of the physiological analyses presented.

We have now added a more detailed review of the CCM and pyrenoid structure in the introduction, as also discussed in our response to Reviewer 3.

I suggest to modify the hypotheses and leave out reference to identifying genetic loci since gene identification is not part of the work presented.

We have modified the hypothesis to remove references to identifying the genetic loci.

“In order to address these hypotheses, we examined two natural isolates of *Chlamydomonas* with varying tolerances to hyperoxia, and their progeny, with the goal of better understanding the physiological mechanisms that underly responses to hyperoxa. Understanding such traits can give insights into the mechanisms and tradeoffs of adaptations for specific environmental niches. By extension, such traits and tradeoffs have strong relevance to applications ranging from algae cultivation to bioengineering crops for increase productivity (Long et al., 2015).”

[Editors' note: further revisions were suggested prior to acceptance, as described below.]

The manuscript has been improved but there are some remaining issues that need to be addressed, as outlined below:1) Overall, the text has improved, but reviewer 5 found some changes that should be made to the final version to improve clarity and concision of the title and text.2) The figures are also improved, but will still benefit from more consistent and careful preparation. The third reviewer makes some excellent suggestions in the last paragraph of their review, particularly regarding image cropping and arrangement; the choice of images (inconsistent planes of section in TEM and inconsistent white balancing in light microscopy); lettering size, position, and font; scale bar size and position; the use and appearance of statistics; and the use of different colours across different figures.Reviewer #5 (Recommendations for the authors):The finding that hyperoxia, possibly through the signalling compound H2O2, can elicit a low-CO2 type acclimation in Chlamydomonas, even in the presence of large excess of CO2 concentrations is, truly, a remarkable result. It forces the CO2-concentrating mechanism community to reconsider part of the prevailing paradigm (consider O2/CO2 ratio rather than just CO2). It also opens exciting research avenues to identify the elicitors and signalling pathway related to CO2 (or O2/CO2) sensing. I have little doubts that this manuscript can attract a high citation. It is also timely as H2O2 sensors for Chlamydomonas have now been developed, which will help further test the H2O2 hypothesis (doi:10.1093/plcell/koab176).The authors have made an excellent effort at improving the manuscript and data presentation, but I feel there is still room for improving both text and figures. Here are a number of suggestions the authors may want to consider, but also some questions which were not raised by previous reviewers.FiguresFigure 1.A: Is this the best, unbiased way to normalize the data? It is not clear from Material and Methods and figure caption whether the data normalized the Rubisco concentration. Do CC-1009 and CC-2343 (who differ five-fold in terms of chlorophyll concentration) accumulate similar or different amounts of the CO2-fixing enzyme?

It is important to note that our main goal was to assess if the rates of carbon fixation were determined by the fraction activation of rubisco. This was expressed as the ratio of activities of as-isolated and fully activated forms of rubisco. This value will be unitless (self-normalized fraction) and will not depend on the procedure used to normalized the activities, because the units cancel out.

Nevertheless, we are grateful that the reviewer took the time to think about the method of normalization, which we agree is important for interpreting the total activity of rubisco. In this regard, we are comfortable with the normalization by chlorophyll and note that renormalizing to other parameters will not substantially affect the interpretation, as detailed in the following:

The differences in chlorophyll contents of the cultures were also quite similar, e.g. we found that CC-2343 had only about 17% higher chlorophyll than CC-1009 (see, e.g., measurement at 31 hours).

1. As we state in the methods, our assay was based on an established protocol for plants (Li et al., 2019; Roeske and Oleary 1985; Sharkey et al., 1986), with the slight adjustments to make the protocol suitable specific for *Chlamydomonas*. In plants, the normalization is usually done according to leaf area (Li et al., 2019). This makes sense because the larger the leaf, the more light it can intercept, and light (the ultimate substrate) is expressed in units of moles photons per second per area.

2. The photobioreactors differ from leaves in three important respects:

a. The exposed area was determined by the geometry of the photobioreactor and was thus constant between samples;

b. Light intensities were set to be equal between the samples;

c. The turbidostat function of the photobioreactor held the cells at approximately the same densities. The biomass density (g per unit volume) in the cultures were measured over time and were nearly identical (not significantly different) between the genotypes, with no significant difference between cultures at day 1 (31 hours hyperoxia). For transparency, we show biomass measurements (ash free dry weight) for CC1009 and CC2343 across our hyperoxia treatment. Note that the turbidostat will maintain approximately the same cell density even when growth rates differ.

**Author response image 1. sa2fig1:** Variations in biomass (ash free dry weight) measured in turbidostat cultures in ePBR units.

Overall, we note that changing the normalization to exposed area, or total biomass would not alter the overall interpretation, i.e., that “although we cannot ascribe the differences in photosynthetic phenotypes solely to rubisco deactivation, these results do suggest that the CO_2_/O_2_ concentrations or metabolic environments near rubisco are different under hyperoxia in the two lines”.

The text suggests for Figure 19 in SI that total fluorescence signal is an acceptable proxy for Rubisco concentration. It isn't. See also comment on Figure 6.

We appreciate the point and agree that the total fluorescence is not a quantitative indicator of rubisco concentration, but that the results are qualitatively consistent with our interpretation. Also, though the increase in rubisco content may not be due to increase in mRNA, there remains post translational mechanisms that could be at play. We clarified this in the text by changing this section:

“Most importantly, we found very clear evidence that even when sparged with saturating CO_2_, hyperoxia increased the rubisco content of the pyrenoids (Appendix 1-Figure 19), asserting that there is some control of rubisco in the pyrenoid in response to oxygen.”.

To instead say:

“We also found evidence that even when sparged with saturating CO_2_, hyperoxia may result in an apparent increase in the aggregation of rubisco in the pyrenoids (Appendix 1-Figure 19), indicating that oxygen has some control of this aggregation.”

We have also modified the figure caption accordingly:

“Appendix 1-Figure 19: Strain CC-5357, containing a RBCS1-Venus. All cells were grown with 14:10 hour (light:dark) sinusoidal illumination with peak light intensity of 2000 μmol m^-2^ s^-1^ PAR, in HS media. Cells were sparged for 60 seconds every 15 minutes with either 5% CO_2_ (Panel A), 5% CO_2_ and 95% Oxygen (Panel B), or ambient air (Panel C). The average fluorescence intensity of the localized Venus Fluorescent Protein-labeled rubisco within the pyrenoids of *Chlamydomonas* cells was measured using the Nikon A1 Confocal Laser Scanning Microscope. Because the tagged rubisco was expressed with a PsaD (Photosystem I reaction center subunit II) promoter, the fluorescent signal may not be a conclusive proxy for rubisco amount and cannot be attributed to transcriptional responses to hyperoxia. However, the results are consistent with the fluorescently-tagged rubisco accumulating to a greater extent in the pyrenoids of cells exposed to hyperoxia, similar to what has been observed when cell are exposed to low CO_2_. Panel D shows the mean intensity of the Venus-tagged rubisco of the three treatments. Significant differences (P <.0001) were found between the High CO_2_ and Hyperoxia; High CO_2_ and Low CO_2_ treatments. No difference was found between Hyperoxia and Low CO_2_. Scale bar = 2 μm.”

Figure 1.B: The initial/final data is redundant and could be in SI. Why not plot the activity on the same graph (as a floating dot or something more elegant), using a second-Y scale?

We tried several ways to plot these results, but in the end determined that this was the least confusing, albeit perhaps less “elegant.” In response to the reviewer’s comment, we have moved this particular data to the SI.

Figure 4: are four rather than a single representative image really needed for a main figure? You have a neat median section for all four. A, top left; B, top right; C, bottom right; D, top left. These would be sufficient and help focus attention.

We thank the reviewer for suggesting that a single image would be sufficient. We attempted to rework this figure as suggested, but concluded that the single figure lost some of the details that we feel are important for understanding the consistencies and commonalities in the images. We thus prefer to keep the composite image.

Figure 6: the use of the Chlamydomonas Culture Collection strain #5357 (RbcS1-Venus mt-) is a judicious choice but comes with limitations. Firstly, the parent strain is genetically different from CC-1009 and CC-2343, and no data is shown on its capacity to handle the high concentrations of H2O2 used in the experiment. Secondly, the tagged variant is co-expressed (under a strong promoter) with the native, untagged Rubisco (and therefore cannot serve as a quantification proxy). It is unclear whether the H2O2 treatment might affect stability of the tagged protein. Thirdly, and maybe more importantly, I would be good to augment the Supplementary Data Set with the Confocal Settings. The raw data on Github suggests that control ('Minus') signal is oversaturated, compared to the H2O2 treated cells. The laser intensity and signal collection settings should be the same for the two treatments. Also, images of the channel showing chlorophyll-autofluorescence should be shown (if available). It is difficult to gage from the images whether the treated cells are still alive. The bloated cells probably have lost their intracellular structures, except the pyrenoid who will resist proteolysis for a while should be excluded from the calculation (cells #1 and #3 for Plus^-1^; cell #1 for Plus-2). This would address any concerns that the near fourfold difference shown in Figure 6.C is not biased by imaging artefacts. Lastly, the bar char (Figure 6.C) would benefit from a more meaningful Y-scale, e.g. fraction of fluorescence signal outside the pyrenoid.

We do not believe that 100 μm of hydrogen peroxide is high, and previous studies (Blaby 2015) used as much as 1mM. More importantly, we show data indicating that CC-5357 grows well under these conditions, i.e., there is no indication that it is deleterious (SI Figure 22). Indeed, CC-5357 appears to have a greater capacity to take up CO_2_ (a lower CO_2_ compensation point) when treated with this hydrogen peroxide when compared to the non-treated sample. We also noticed when conducting our light microscopy of CC-5357 that the addition of 100 μm H2O2 did not appear to inhibit cell motility, though we do not present those results.

With regard to the confocal settings, we can assure the reviewer that we used the same confocal settings for both the H2O2 treated and non-treated cells (For the images acquired on Figure 6 on the Olympus FV1000, fluorescence was excited using 3% Argon gas laser intensity. Fluorescence emission was recorded through 530-630 nm band pass filter using a Photomultiplier detector with a high voltage 831). The reviewer is correct, though, in that there may be oversaturation in the images. It was partially because of the concern for oversaturation, though, that we repeated the procedure with the Nikon. The Nikon is MSU’s premier confocal instrument, with great sensitivity, and getting time on it is very difficult at the MSU’s Center for Advanced Microscopy, so we began with the Olympus. When we repeated the experiment and examined the cells with the Nikon, we got the similar results (see Appendix 1 – Figure 17). We have now also included the settings we used when we used the Nikon instrument in the figure caption Fluorescence was detected using 0.5% 514 nm diode laser intensity. Fluorescence emission was recorded through 565/70 nm band pass filter using a Gallium Arsenide Phosphide (GaASP) detector (High Voltage = 47, offset -7). Lastly, we repeated the test one more time on the Olympus and, again, observed similar results (See Transparent Reporting Figure 11). The settings we used for the images in Transparent Reporting Figure 11 were nearly the same as that what we used with Figure 6, we just raised the voltage from 831 to 850. We have added these settings to the figure caption for Transparent Reporting Figure 11. But to be clear, whenever we had a session with a confocal, we took images of both control and treated cells with the same settings!

We have redone Figure 6, as requested, excluding the bloated cells and improving the overall appearance of the figure with a box plot. The effect of hydrogen peroxide on the “fraction of fluorescence signal outside of the pyrenoid” was highly significant (P<.001), and we thank the reviewer for their suggestion.

Figure 7 and L711-713: This data makes more sense if presented in the context of optimizing experimental conditions, building up to experiment in Figure 6, rather than on its own. Why not move this to supporting information? I would also err on the side of caution before stating "this is the first study, to our knowledge, to show that the most complete degree of rubisco delocalization occurs when an alga(e) is grown on a TAP agar plate exposed to air, rather than aquatically in liquid TAP". The phenomenon is well known to anyone who has worked with Chlamydomonas and looked under a microscope. There are probably numerous earlier publications that used EM, where the pyrenoid is either not visible or massively reduced. I understand the authors' desire to maximize priority on any data, but this data is not relevant to the overall narrative. It distracts from the highly original and remarkable findings of your main work.

We find the statement that “the phenomenon is well known to anyone who has worked with *Chlamydomonas* and looked under a microscope,” both comforting (in that it offers some confirmation of our observations) and perplexing, because we have not been able to find a description of the phenomenon in the literature.

Please note that we found the lack of reports of this phenomenon surprising and made some effort to track down any previous references. Further, we already used caution in stating that “to our knowledge…” the phenomenon has not been reported.

We emphasize that the phenomenon--regardless of whether is it well known to anyone withing with *Chlamydomonas* (expect perhaps us!)—is both non-trivial and important for interpreting our and others’ results. Specifically, the observation is important to our study because it suggests that factors other than simply CO_2_ levels control the pyrenoid and CCM. We would like to keep Figure 7 in the main part of the paper, because we do feel like it is important result.

Nevertheless, we have revised the text to remove the implied novelty of the observation, so that we now write,

“Also consistent with the cited previous work, low-light with mixotrophic conditions resulted in rubisco delocalization. We further show that the most complete degree of rubisco delocalization occurs when algae is grown on a TAP agar plate exposed to air, rather than aquatically in liquid TAP (Figure 7).”

Title"Induction of Pyrenoid Synthesis", is rather broad and does not really capture the essence of your paper. In your rebuttal, you state "Because the main point of the paper is that hyperoxia and hydrogen peroxide induce starch sheath formation, we focused on measurements of the starch sheath." There are numerous instances of "pyrenoid induction" in the text, but the dynamics or Rubisco relocation or starch deposition are not really investigated. Let alone the tubules around which the matrix/starch sheath form. Why not consider a positive statement, for example: "Chlamydomonas grown in hyperoxia and elevated CO2 has a pyrenoid: Implications for the natural diversity of photosynthetic acclimation".

We will remove the word “synthesis” from the title based on the reviewers comments, but the word induction is necessary to convey that hyperoxia (and H2O2) induces the pyrenoid. We show that hyperoxia is condition where the pyrenoid forms and that H2O2 can also induce the pyrenoid, even under high CO_2_. Based on the reviewer’s comments, we may change the comment to the following:

The Induction of a Pyrenoid by Hyperoxia: Implications for the Natural Diversity of Photosynthetic Responses in *Chlamydomonas*

Material and methods, Experimental designAlgal growth conditions: 2000 micromols m-2 s^-1^ is 20x more than what Spreitzer and Mets (1981) used to assay light sensitive mutants, namely 2000 and 4000 Lux (cited in L322). Maybe add a note justifying such high intensities, to show that this was not a double stress experiment.

Very good point. We have added this: “Spreitzer and Mets (1981) found that rubisco activity-deficient mutants exhibited chlorotic phenotypes similar to those observed with CC-2343 under hyperoxia. We conjectured that rubisco inhibition may be playing a role in CC-2343’s intolerance to hyperoxia. To be clear, while Spreitzer and Mets (1981) were screening for mutants that were highly sensitive to even low light (~90 μmoles m^-2^ s^-1^ PAR), we grew our wild type strains of *Chlamydomonas* under hyperoxia with diurnal sinusoidal light with peak light intensities of 2000 μmoles m^-2^ s^-1^ PAR. We found that, when sparged with 5% CO_2_, CC-1009 and CC-2343 grow very well at such light intensities.”

Experimental design (see also comment on Figure 1, above): in the opening sentence of results (L322-327), you explain the rational for choosing CC-1009 and CC-2343. It is, however, unclear to a non-specialist how wildtype like CC-1009 is, relative to (for example) the reference strain used for the genome or the widely used CLiP mutant library. Is CC-1009 particularly good at handling hyperoxia or is just as good as any of the WT used by the Chlamydomonas community?

Thank you for the point. We did test a number of strains for sensitivity to hyperoxia and cite the work of Hall (Ph.D. Thesis of former student in the Kramer lab) in our results.

In an initial screen of sequenced *Chlamydomonas* isolates (Jang and Ehrenreich, 2012), we found two with contrasting tolerances to hyperoxia, with strain CC-1009 relatively tolerant to hyperoxia, continuing to grow, albeit at a suppressed rate, when exposed to 95% oxygen and 5% CO_2_, while CC-2343 showed severely suppressed growth and eventual chlorosis or photobleaching in our ePBRs (Hall, 2017).

This work is available online so it should be accessible. Another paper (Lucker et al.) describing these characteristics is under review, and we will cite it when accepted. We tested CC-1009 with several different non-lab strains (CC-1009 was chosen because it was closely related to the widely used lab strains).

Also, in the Methods we state that “Strain CC-1009 (mt-) is a wild type strain tracing back to the 1945 collection by G.M. Smith, collected from Amherst MA, but has been a separate line from the c137c (CC-124 and CC-125) and Sagar (CC-1690) since about 1950 (Pröschold et al., 2005).”

However, we did not clarify that c137 is the reference strain. Thus, we have modified the text as follows,

“Strain CC-1009 (mt-) is a wild type strain tracing back to the 1945 collection by G.M. Smith, collected from Amherst MA, but has been a separate line from the sequenced and widely regarded reference strain c137 (CC-124 and CC-125) and Sagar (CC-1690) since about 1950 (Pröschold et al., 2005).”